# Provable and Practical Online Learning Rate Adaptation
# with Hypergradient Descent

**Ya-Chi Chu** [* 1]  **Wenzhi Gao** [* 2]  **Yinyu Ye** [2 3 4]  **Madeleine Udell** [2 4]

## Abstract

This paper investigates the convergence properties of the hypergradient descent method (HDM), a 25-year-old heuristic originally proposed for adaptive stepsize selection in stochastic first-order methods (Almeida et al., 1999; Baydin et al., 2018). We provide the first rigorous convergence analysis of HDM using the online learning framework of Gao et al. (2024) and apply this analysis to develop a new state-of-the-art adaptive gradient methods with empirical and theoretical support. Notably, HDM automatically identifies the optimal stepsize for the local optimization landscape and achieves local superlinear convergence. Our analysis explains the instability of HDM reported in the literature and proposes efficient strategies to address it. We also develop two HDM variants with heavy-ball and Nesterov momentum. Experiments on deterministic convex problems show HDM with heavy-ball momentum (HDM-HB) exhibits robust performance and significantly outperforms other adaptive first-order methods. Moreover, HDM-HB often matches the performance of L-BFGS, an efficient and practical quasi-Newton method, using less memory and cheaper iterations.

## 1. Introduction

We consider smooth convex optimization problem

$$\underset{x \in \mathbb{R}^n}{\text{minimize}} \ f(x),$$

---
[*]Equal contribution [1]Department of Mathematics, Stanford University [2]ICME, Stanford University [3]Shanghai Jiao Tong University and Shanghai Institute for Mathematics and Interdisciplinary Sciences [4]Management Science and Engineering, Stanford University. Correspondence to: Ya-Chi Chu <ycchu97@stanford.edu>, Wenzhi Gao <gwz@stanford.edu>.

*Proceedings of the 42^{nd} International Conference on Machine Learning*, Vancouver, Canada. PMLR 267, 2025. Copyright 2025 by the author(s).

where $f : \mathbb{R}^n \to \mathbb{R}$ is convex and $L$-smooth with $f(x^\star) := \min_x f(x) > -\infty$. Theoretically, gradient descent

$$x^{k+1} = x^k - \alpha_k \nabla f(x^k)$$

with constant stepsize $\alpha_k \equiv 1/L$ is guaranteed to converge. However, the choice of stepsize $\alpha_k$ strongly affects the performance of gradient descent in practice (Defazio et al., 2024; Agarwal et al., 2020). Various stepsize selection strategies have been proposed to improve the convergence of gradient descent. Examples include line-search (Armijo, 1966), Polyak stepsize (Polyak, 1987), stepsize scheduling (Li et al., 2021; Wang and Yuan, 2023), hypergradient descent (Almeida et al., 1999; Rubio, 2017; Baydin et al., 2018) and the well-known adaptive stepsizes (Orabona and Pál, 2016; Duchi et al., 2011; Kingma, 2014; Malitsky and Mishchenko, 2020; 2024).

This paper focuses on the *hypergradient descent method* (HDM), which was initially proposed by Almeida et al. (1999) as a heuristic for stochastic optimization. It was later tested by Baydin et al. (2018) on modern machine learning problems and exhibited promising performance. In HDM, the stepsize $\alpha_k$ is adjusted by another gradient descent update:

$$\alpha_{k+1} = \alpha_k - \tilde{\eta}_k \frac{\mathrm{d}}{\mathrm{d}\alpha}[f(x^k - \alpha \nabla f(x^k))]\big|_{\alpha = \alpha_k}$$
$$= \alpha_k - \eta_k \frac{-\langle \nabla f(x^{k+1}), \nabla f(x^k) \rangle}{\|\nabla f(x^k)\|^2},$$

where the hypergradient stepsize $\tilde{\eta}_k$ is often set to be $\tilde{\eta}_k = \frac{\eta_k}{\|\nabla f(x^k)\|^2}$ for $\eta_k > 0$ to make the update invariant of the scaling of $f$. Recently, Gao et al. (2024) generalized HDM to update a matrix stepsize (preconditioner) $P_k \in \mathbb{R}^{n \times n}$ in *preconditioned* gradient descent through the iteration

$$x^{k+1} = x^k - P_k \nabla f(x^k), \tag{1}$$
$$P_{k+1} = P_k - \eta_k \frac{-\nabla f(x^{k+1})\nabla f(x^k)^\top}{\|\nabla f(x^k)\|^2}, \tag{2}$$

where (2) follows from

$$\nabla_P[f(x^k - P \nabla f(x^k))]\big|_{P=P_k} = -\nabla f(x^{k+1})\nabla f(x^k)^\top.$$

We call the update (1)-(2) *vanilla* HDM throughout the paper. In practice, matrix stepsize $P_k$ is often set to be diagonal and the update (2) simplifies to

$$P_{k+1} = P_k - \eta_k \frac{-\mathrm{diag}(\nabla f(x^{k+1}) \circ \nabla f(x^k))}{\|\nabla f(x^k)\|^2},$$

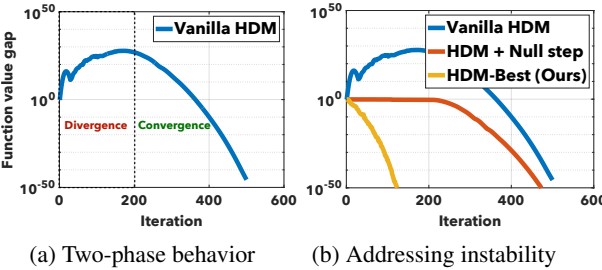

(a) Two-phase behavior     (b) Addressing instability

Figure 1: The convergence behavior of different `HDM` variants on a toy quadratic optimization problem. Figure 1a: two-phase convergence behavior of vanilla `HDM`. Figure 1b: effect of null step and our best variant `HDM-Best`.

where $\circ$ is the entry-wise product and $\mathrm{diag}(d)$ is the diagonal matrix with $d \in \mathbb{R}^n$ on the diagonal.

While vanilla `HDM` has been used heuristically in various applications (Chandra et al., 2022; Wang et al., 2023; Ozkara et al., 2024; Baydin et al., 2018), it can be unstable if the hypergradient stepsize $\eta_k$ is not carefully tuned (Kunstner et al., 2024; Chandra et al., 2022; Rubio, 2017). Figure 1a shows $f(x^k)$ can spike as high as $10^{30}$ in the early iterations of vanilla `HDM`, which would lead users to abandon the algorithm. Surprisingly, our analysis reveals that this behavior of `HDM` is not true divergence; instead, it can be understood as the warm-up phase of an online learning procedure, and is followed by fast convergence (Figure 1a). Moreover, we show in both theory and practice that the explosion of $f(x^k)$ can be circumvented by taking a *null step*, which skips the update whenever the new iterate fails to decrease the objective value, i.e., $f(x^k - P_k \nabla f(x^k)) \geq f(x^k)$. The null steps flatten the objective value curve in the warm-up phase of `HDM` but cannot shorten the warm-up (Figure 1b).

Our analysis exploits the online learning framework by Gao et al. (2024), in which the authors observe that the $P$-update (2) in vanilla `HDM` can be viewed as online gradient descent with respect to the online surrogate loss

$$h_x(P) := \frac{f(x - P\nabla f(x)) - f(x)}{\|\nabla f(x)\|^2}. \tag{3}$$

The function $h_x(P)$, called *hypergradient feedback* in this paper, is a function of preconditioner $P$ and is well-defined for all non-stationary $x$. To see that (2) aligns with the online gradient descent update, notice $\nabla h_{x^k}(P_k) = -\frac{\nabla f(x^{k+1})\nabla f(x^k)^\top}{\|\nabla f(x^k)\|^2}$ so the update (2) sets

$$P_{k+1} = P_k - \eta_k \nabla h_{x^k}(P_k).$$

To ensure boundedness of $P_k$, $P$-update in Gao et al. (2024) and in this paper is projected onto a bounded closed convex candidate set $\mathcal{P}$:

$$P_{k+1} = \Pi_{\mathcal{P}}[P_k - \eta_k \nabla h_{x^k}(P_k)]$$

However, we do *not* require $\mathcal{P}$ to be a subset of positive (semi)definite cone and $\mathcal{P}$ is often taken to be a simple subset such as a ball or a box.

---

**Algorithm 1** Hypergradient Descent Method (`HDM`)

**input** initial point $x^1, P_1 \in \mathcal{P}$ (not necessarily PSD)
**for** $k = 1, 2,...$ **do**
$$x^{k+1} = \underset{x \in \{x^k, x^k - P_k\nabla f(x^k)\}}{\arg\min} f(x)$$
$$P_{k+1} = \Pi_{\mathcal{P}}[P_k - \eta_k \nabla h_{x^k}(P_k)]$$
**end**

---

Vanilla `HDM` + null steps (Algorithm 1) was first considered by Gao et al. (2024) and guaranteed to converge globally. However, their analysis is not sufficient to explain the practical behavior of `HDM` and provides no advice for how to design a practically efficient `HDM`. In this paper, we dive deeper into the convergence behavior of `HDM` (Algorithm 1), establishing sharper global convergence guarantees and conducting a local convergence analysis. Our findings offer new insights into (vanilla) `HDM` and serve as a foundation to design more efficient and practical variants of `HDM`. The contributions of this paper include:

- We provide the first rigorous convergence analysis for `HDM`, including both global and local convergence guarantees (Section 3) that show `HDM` can adapt to the local optimization landscape. Our analysis provides several new insights into how `HDM` adapts to optimization landscapes (Section 3.1), why vanilla `HDM` is unstable in practice (Section 3.2), and the connection between `HDM` and quasi-Newton methods (Section 3.3).

- We develop and analyze two improved variants of `HDM`: `HDM` + heavy-ball momentum (`HDM-HB` in Section 4.1), which has the same convergence rate as `HDM` but is faster than `HDM` in practice; and `HDM` + Nesterov momentum (`HDM-AGD` in Section 4.2), which is faster in theory and intermediate between `HDM` and `HDM-HB` in practice.

- We develop a practically efficient variant `HDM-Best` (Figure 1b), which updates $x^k$ by preconditioned gradient descent with heavy-ball momentum and jointly updates $P_k$ and momentum parameter by `AdaGrad`. Our `HDM-Best` outperforms most adaptive first-order methods and performs on par with `L-BFGS` (with memory size 5 or 10) using *less memory* (memory size 1) (Section 5).

### 1.1. Related Literature

**Adaptive First-order Methods.** Notable adaptive first-order methods include `AdaGrad` (Duchi et al., 2011; McMahan and Streeter, 2010), `Adam` (Kingma, 2014; Zhang et al., 2024), parameter-free stepsizes (Orabona and

Pál, 2016; Defazio et al., 2024), and online learning guided stepsizes (Zhuang et al., 2019). Most of these techniques originate in the online learning community, and they typically achieve both strong empirical convergence and online learning regret guarantees.

**Hypergradient Descent.** Hypergradient descent dates back to Almeida et al. (1999), which was first proposed as a heuristic to accelerate stochastic gradient descent. Similar concepts were also explored in Sutton (1992); Schraudolph (1999); Jacobs (1988); Mahmood et al. (2012), while those works employed slightly different algorithmic updates. Later, Baydin et al. (2018) rediscovered the `HDM` and named it "hypergradient descent"; Baydin et al. (2018) also extended `HDM` to other first-order methods with extensive experimental validation of its practical efficacy. Recent studies (Jie et al., 2022; Chandra et al., 2022; Ozkara et al., 2024) further empirically enhanced `HDM` for broader applicability, reporting promising numerical results.

Despite these empirical successes, a rigorous theoretical understanding of `HDM` has emerged only recently. Rubio (2017) showed that `HDM` converges on convex quadratic functions and established several analytic properties. Subsequently, Kunstner et al. (2024) demonstrated that when using a diagonal preconditioner, hypergradient can be employed to generate cutting planes in the preconditioner space, achieving an $\mathcal{O}(\sqrt{n}\kappa^\star \log(1/\varepsilon))$ complexity on smooth strongly convex functions. Here, $\kappa^\star$ is the condition number associated with the optimal diagonal preconditioner. The idea to update the preconditioner using the ellipsoid method also appeared in Monteiro et al. (2004) for solving linear systems, albeit with a very different motivation. More recently, Gao et al. (2024) showed that `HDM` can be viewed as online gradient descent applied to some surrogate loss function and that `HDM` has strong trajectory-based convergence guarantees.

### 1.2. Notations

We denote Euclidean norm by $\|\cdot\|$ and Euclidean inner product by $\langle \cdot, \cdot \rangle$. The upper and lower case letters $A, a$ respectively denote matrices and scalars. Denote the Frobenius norm by $\|A\|_F := \sqrt{\sum_{ij} a_{ij}^2}$. Define $[\cdot]_+ :=$ $\max\{\cdot, 0\}$. We use $\Pi_\mathcal{C}[\cdot]$ to denote the orthogonal projection onto a closed convex set $\mathcal{C}$ and use $\operatorname{dist}(x, \mathcal{C}) :=$ $\|x - \Pi_\mathcal{C}[x]\|$ to denote the distance between a point $x$ and a closed convex set $\mathcal{C}$. Denote the optimal set of $f$ by $\mathcal{X}^\star = \{x : f(x) = f(x^\star)\}$; and the $\alpha$-sublevel set of $f$ by $\mathcal{L}_\alpha := \{x : f(x) \le \alpha\}$. The condition number of an $L$-smooth and $\mu$-strongly convex function is $\kappa := L/\mu$. For consistency of notation, a *stepsize* $P$ in this paper always refers to a matrix applied in the gradient update. Define the set of scalar stepsizes $\mathcal{S} := \{P = \alpha I : \alpha \in \mathbb{R}\}$ and diagonal stepsizes $\mathcal{D} := \{P = \operatorname{diag}(d) : d \in \mathbb{R}^n\}$, for which

the hypergradient feedback (3) simplifies to

$$h_x(\alpha) := \frac{f(x - \alpha \nabla f(x)) - f(x)}{\|\nabla f(x)\|^2} \quad \text{if } \mathcal{P} = \mathcal{S};$$
$$h_x(d) := \frac{f(x - d \circ \nabla f(x)) - f(x)}{\|\nabla f(x)\|^2} \quad \text{if } \mathcal{P} = \mathcal{D}.$$

## 2. Background: `HDM` and Online Learning

This section establishes the connection between `HDM` and online learning through the framework in Gao et al. (2024). We refer to the following assumptions in the paper.

**A1:** $f(x)$ is $L$-smooth and convex.

**A2:** $f(x)$ is $\mu$-strongly convex with $\mu > 0$.

**A3:** Closed convex set $\mathcal{P}$ satisfies $0 \in \mathcal{P}, L^{-1}I \in \mathcal{P}$ and $\operatorname{diam}(\mathcal{P}) := \min_{P, Q \in \mathcal{P}} \|P - Q\|_F \le D < \infty$.

### 2.1. Descent Lemma and Hypergradient Feedback

Hypergradient feedback (3) is motivated by descent lemma:

$$f(x - \tfrac{1}{L}\nabla f(x)) - f(x) \le -\tfrac{1}{2L}\|\nabla f(x)\|^2.$$

The descent lemma states that, under the constant stepsize $P_k \equiv \frac{1}{L}I$, the function value progress of a gradient step is at least proportional to $\|\nabla f(x)\|^2$ with ratio $-1/(2L)$. When an (effective) preconditioner $P_k$ is used, the *effective* smoothness constant decreases, and thus the ratio $h_x(P) = \frac{f(x - P\nabla f(x)) - f(x)}{\|\nabla f(x)\|^2}$ is expected to become smaller than $-1/(2L)$, yielding faster convergence. Hence, the ratio $h_x(P)$ is a suitable feedback to measure the quality of a preconditioner. `HDM` uses this feedback to learn a good preconditioner with online gradient descent. The hypergradient feedback $h_x(P)$ has the following properties.

**Lemma 2.1** (Extension of Proposition 6.1 in Gao et al. (2024)). *For any $x \notin \mathcal{X}^\star$.*

- *Under **A1**, $h_x(P)$ is convex and $L$-smooth and $h_x(\frac{1}{L}I) \le -\frac{1}{2L}$. Moreover, if **A2** holds and $\mathcal{P} \subseteq \mathcal{S}$, then $h_x(\alpha)$ is $\mu$-strongly convex.*

- *Under **A1** and **A3**, $h_x(P)$ is $(LD + 1)$-Lipschitz. Moreover, if **A2** holds and $\mathcal{P} \subseteq \mathcal{D}$, then $h_x(d)$ is $\frac{\mu}{(1+LD)^2}$-exponential concave (Hazan et al., 2007).*

### 2.2. Online Learning Guarantees

Using convexity and Lipschitz continuity of $h_x(P)$, analysis in online learning (Orabona, 2019; Hazan et al., 2016) guarantees sublinear regret for online gradient descent.

**Lemma 2.2** (Sublinear regret (Gao et al., 2024)). *Under **A1** and **A3**, online gradient descent*

$$P_{k+1} = \Pi_\mathcal{P}[P_k - \eta_k \nabla h_{x^k}(P_k)] \tag{4}$$

with stepsize $\eta_k \equiv \frac{D}{2(LD+1)\sqrt{K}}$ or $\eta_k = \frac{D}{2(LD+1)\sqrt{k}}$ generates $\{P_k\}$ such that

$$\sum_{k=1}^{K} h_{x^k}(P_k) - \min_{P \in \mathcal{P}} \sum_{k=1}^{K} h_{x^k}(P)$$
$$\leq \rho_K := 8D(LD+1)\sqrt{K}. \quad (5)$$

If strong convexity **A2** is further assumed and $P_k \in \mathcal{S}$, a different choice of hypergradient stepsize $\eta_k$ in (4) improves the regret to $\log K$.

**Lemma 2.3** (Logarithmic regret). *Instate **A1** to **A3** and suppose $\mathcal{P} \subseteq \mathcal{S}$. Then online gradient descent (4) with $\eta_k = 1/(k\mu)$ generates $\{P_k\}$ such that $\sum_{k=1}^{K} h_{x^k}(P_k) - \min_{P \in \mathcal{P}} \sum_{k=1}^{K} h_{x^k}(P) \leq \frac{(LD+1)^2}{2} \log K$.*

Given exponential-concavity of $h_x$ in Lemma 2.1, it is possible to apply online learning algorithms such as the online Newton method (Hazan et al., 2007).

*Remark* 1. The diameter $D$ of candidate stepsize set $\mathcal{P}$ is measured in Frobenius norm (**A3**) and can incur dimension dependence in the regret bound when $P_k$ are diagonal or full matrices. This introduces a trade-off between online regret and the best possible cumulative feedback $\min_{P \in \mathcal{P}} \sum_{k=1}^{K} h_{x^k}(P)$: full matrices may achieve smaller feedback but have larger dimension factor in the regret bound. Diagonal stepsize often strikes a good balance in practice.

### 2.3. Hypergradient Reduction and HDM

One major contribution of Gao et al. (2024) is a reduction that relates the minimization of cumulative hypergradient feedback $\sum_{k=1}^{K} h_{x^k}(P_k)$ to the function value gap:

$$\varphi(x^k) := f(x^k) - f(x^\star). \quad (6)$$

We provide a sharper version of this reduction.

**Lemma 2.4** (Sharper version of Lemma 6.1 in Gao et al. (2024)). *Denote the negative average feedback by*

$$\gamma_K(\{P_k\}) := -\frac{1}{K} \sum_{k=1}^{K} h_{x^k}(P_k).$$

*Under **A1**, the iterates generated by Algorithm 1 satisfy*

$$\varphi(x^{K+1}) \leq \min\left\{ \frac{\Delta^2}{K[\gamma_K(\{P_k\})]_+}, \varphi(x^1) \right\},$$

*where $\Delta = \max_{x \in \mathcal{L}_{f(x^1)}} \min_{x^\star \in \mathcal{X}^\star} \|x - x^\star\|$. Further, under **A1** and **A2**,*

$$\varphi(x^{K+1}) \leq \varphi(x^1)(1 - 2\mu [\gamma_K(\{P_k\})]_+)^K.$$

According to Lemma 2.4, the negative average feedback $\gamma_K(\{P_k\})$ determines the rate for sublinear/linear convergence of Algorithm 1: larger $\gamma_K(\{P_k\})$ implies faster con-

vergence. Given the objective $\gamma_K(\{P_k\})$, HDM applies online gradient descent to generate a sequence of preconditioners $\{P_k\}$ that guarantee the following lower bound:

$$\gamma_K(\{P_k\}) \geq \max_{P \in \mathcal{P}} \frac{1}{K} \sum_{k=1}^{K} -h_{x^k}(P) + o(1), \quad (7)$$

which follows from the sublinear regret $\rho_K = o(K)$ in Lemma 2.2 and Lemma 2.3, implying $\frac{\rho_K}{K} = o(1)$.

## 3. The Convergence Behavior of HDM

This section presents our main convergence results on HDM and consequent insights. All the analyses are based on the online learning framework established in Section 2.

### 3.1. HDM Adapts to the Local Landscape

Our first convergence result follows by combining Lemma 2.4 and Lemma 2.2:

**Theorem 3.1** (Static adaptivity). *Instate $\varphi(x^k)$ in (6). Under **A1** and (**A1** + **A2**) respectively, Algorithm 1 with $\eta_k \equiv \frac{D}{2(LD+1)\sqrt{K}}$ or $\eta_k = \frac{D}{2(LD+1)\sqrt{k}}$ satisfies*

$$\varphi(x^{K+1}) \leq \min\left\{ \frac{\Delta^2}{K[\gamma_K^\star - \frac{\rho_K}{K}]_+}, \varphi(x^1) \right\}; \quad \textbf{(A1)}$$

$$\varphi(x^{K+1}) \leq \varphi(x^1)(1 - 2\mu[\gamma_K^\star - \frac{\rho_K}{K}]_+)^K, \quad \textbf{(A1 + A2)}$$

*where $\Delta$ is the same as defined in Lemma 2.4, $\rho_K$ is defined in (5), and $\gamma_K^\star := -\min_{P \in \mathcal{P}} \frac{1}{K} \sum_{k=1}^{K} h_{x^k}(P)$. In particular, the relations hold for all $K \geq 1$ when $\eta_k = \frac{D}{2(LD+1)\sqrt{k}}$.*

Theorem 3.1 has two implications: 1) Since $\gamma_K^\star \geq -\frac{1}{K} \sum_{k=1}^{K} h_{x^k}(\frac{1}{L}I) \geq \frac{1}{2L}$ (by descent lemma) and $\frac{\rho_K}{K} = o(1)$, both upper bounds in Theorem 3.1 converge to 0 when $K$ goes to infinity, guaranteeing global convergence of HDM. 2) More importantly, $\gamma_K^\star$ reflects the possibly improved convergence rate of HDM through the adaptive $P$-update, which depends on the local optimization landscape. To see this, when $K$ is large and $\frac{\rho_K}{K}$ is negligible, the convergence of HDM is competitive with preconditioned gradient descent (1) with any *static* preconditioner. In particular, the optimal $P_K^\star := \arg\min_{P \in \mathcal{P}} \frac{1}{K} \sum_{k=1}^{K} h_{x^k}(P)$ achieves the rate $\frac{\Delta^2}{K\gamma_K^\star}$. Note that $\gamma_K^\star$ (or $P_K^\star$) depends only on the past trajectory $\{x^k\}_{k \leq K}$; and thus if the algorithm visits a local region with a smaller smoothness constant than the global constant $L$, one can expect $\gamma_K^\star \gg \frac{1}{2L}$. In summary, HDM *adapts to the local optimization landscape* and leads to faster convergence than standard gradient descent.

We borrow a standard dynamic regret argument in online convex optimization literature (Hazan et al., 2016) to provide an even stronger notion of adaptivity of HDM:

**Theorem 3.2** (Dynamic adaptivity). *Instate (6). Under **A1** and (**A1** + **A2**) respectively, Algorithm 1 with stepsize $\eta_k \equiv$*

$\frac{D}{2(LD+1)\sqrt{K}}$ *satisfies*

$$\varphi(x^{K+1}) \leq \min\left\{ \frac{\Delta^2}{K[\delta_K^\star - \frac{\rho_K}{K}]_+}, \varphi(x^1) \right\}; \qquad \textbf{(A1)}$$

$$\varphi(x^{K+1}) \leq \varphi(x^1)(1 - 2\mu[\delta_K^\star - \frac{\rho_K}{K}]_+)^K, \quad \textbf{(A1 + A2)}$$

*where $\Delta$ is the same as defined in Lemma 2.4, $\rho_K$ is defined in* (5),

$$\delta_K^\star := - \min_{\{\hat{P}_k \in \mathcal{P}\}} \left\{ \frac{1}{K} \sum_{k=1}^{K} h_{x^k}(\hat{P}_k) \right. $$
$$\left. + \frac{2(LD+1)\mathsf{PL}(\{\hat{P}_k\})}{\sqrt{K}} \right\}, \qquad (8)$$

*and* $\mathsf{PL}(\{\hat{P}_k\}) := \sum_{k=1}^{K-1} \|\hat{P}_{k+1} - \hat{P}_k\|_F$.

Theorem 3.1 and Theorem 3.2 differ in the constants $\gamma_K^\star$ and $\delta_K^\star$, as the minimum in (8) searches over different optimal preconditioners for different $h_{x^k}$. Theorem 3.2 shows that, even if the sequence $\{x^k\}$ traverses different regions of the landscape, HDM automatically chooses $\hat{P}_k$ to adapt to the local region, at the price of an additional regret term $\mathsf{PL}(\{\hat{P}_k\})$. Note that the upper bounds in Theorem 3.2 holds for *any* benchmark path $\{\hat{P}_k\} \subset \mathcal{P}$. In particular, HDM is guaranteed to asymptotically achieve the performance of the optimal path that maximizes the algorithm progress. Adaptivity of HDM undergirds its good empirical performance.

**Behavior of HDM and Static/Dynamic Adaptivity.** We observe that the behavior of HDM can be divided into two stages: (1) when the iterates $x^k$ are far from the optimum $x^\star$, the change in landscape is more drastic and we expect dynamic adaptivity to capture the convergence behavior; (2) when $x^k$ is near the optimum $x^\star$, $f(x)$ locally behaves like a quadratic and $P_k$ remains more stable, and thus static adaptivity describes the convergence behavior.

### 3.2. Online Regret and Instability

Though adaptive $P$-update underpins the strong performance of HDM, vanilla HDM is observed unstable in practice. This section identifies the source of instabilty in vanilla HDM (Figure 1) based on our analysis. We also propose two simple yet effective strategies to address the instability.

**Divergence Behavior due to Regret.** Recall from Theorem 3.1 that the optimality gap at $x^{K+1}$ is bounded by $\frac{\Delta^2}{K[\gamma_K^\star - \frac{\rho_K}{K}]_+}$. This rate can be better than that of gradient descent when $K$ is large and $\gamma_K^\star \gg \frac{1}{2L}$, but the analysis provides no guarantee on earlier iterates $\{x^k\}_{k \leq K}$. In particular, the convergence rate makes sense only if $\gamma_K^\star > \frac{\rho_K}{K}$. That is, the progress $\sum_{k=1}^{K} h_{x^k}(P_k)$ accumulated by the online gradient descent outweighs its regret $\rho_K$. In other words, online gradient descent takes

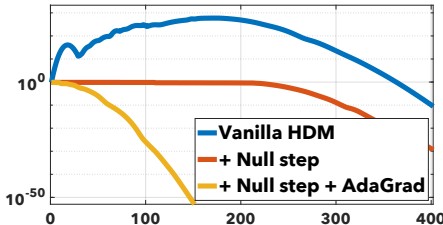

Figure 2: Addressing instability of HDM

time to learn a good preconditioner, and the regret accumulated during this warm-up phase causes HDM to behave as if it is diverging until the progress $\sum_{k=1}^{K} h_{x^k}(P_k)$ outpaces the regret $\rho_K$. Since $\rho_K$ grows sublinearly with the iteration count $K$, HDM will eventually converge. However, the objective value will usually explode and possibly be terminated by the user before convergence begins. Consequently, the two-phase convergence behavior (Figure 1a) is rarely observed.

**Addressing Instability.** While our analysis guarantees HDM will converge eventually, an algorithm that diverges up to $10^{30}$ before converging is not practical. We propose two simple but effective fixes based on our analysis:

- *Null step.* The $x$-update is skipped if new iterate increases objective value (see the first line of Algorithm 1). The null step ensures a monotonic decrease as HDM learns a good preconditioner, although it requires an additional function value oracle call at each iteration. Even on iterations when $x^k$ is not updated, the preconditioner $P_k$ is updated using online gradient descent, so the algorithm is still making progress. In Figure 2, the null steps flatten the objective value curve in the divergence phase.

- *Advanced Learning Algorithms.* Better online learning algorithms with lower regret shorten the divergence phase. Figure 2 shows a significant speedup when the online gradient descent in Algorithm 1 is replaced by AdaGrad. In our experiments, AdaGrad often improves the robustness of HDM since it does not require pre-specifying algorithm parameters that depend on the total iteration count $K$ and provides convergence guarantees for the earlier iterates $\{x^k\}_{k \leq K}$ (Appendix B.6).

### 3.3. Local Superlinear Convergence

Figure 1 shows HDM converges faster than the (linearly convergent) first-order methods. In fact, HDM exhibits local superlinear convergence on strongly convex objectives (Theorem 3.3 below). This subsection assume a strongly convex objective (**A2**) and Lipschitz Hessian (**A4**):

**A4:** $f(x)$ has $H$-Lipschitz Hessian.

**Strongly Convex Quadratics.** We develop intuition by considering a strongly convex quadratic. For $f(x) = \frac{1}{2}\langle x, Ax\rangle - \langle b, x\rangle$, we have $x^\star = x - [\nabla^2 f(x^\star)]^{-1}\nabla f(x)$. In other words, $P^\star = [\nabla^2 f(x^\star)]^{-1}$ is a universal minimizer of $h_x(P)$ that drives any non-optimal point $x \notin \mathcal{X}^\star$ to the optimum $x^\star$ in one step. When $[\nabla^2 f(x^\star)]^{-1} \in \mathcal{P}$, Theorem 3.1 guarantees the performance of HDM is competitive to $[\nabla^2 f(x^\star)]^{-1}$. Therefore, we expect the descent curve to decrease more and more sharply, giving superlinear convergence (Figure 1a).

**Local Superlinear Convergence.** For general functions satisfying **A4**, $f(x)$ behaves like a quadratic near $x^\star$:

$$f(x) \approx f(x^\star) + \tfrac{1}{2}\langle x - x^\star, \nabla^2 f(x^\star)(x - x^\star)\rangle.$$

Therefore, local superlinear convergence is expected for HDM near $x^\star$. Theorem 3.3 formalizes this intuition.

**Theorem 3.3** (Local superlinear convergence). *Instate $\varphi(x^k)$ in (6). Suppose $[\nabla^2 f(x^\star)]^{-1} \in \mathcal{P}$ and assume **A1** to **A4**. Then Algorithm 1 with $\eta_k = \frac{D}{2(LD+1)\sqrt{k}}$ has local superlinear convergence:*

$$\varphi(x^{K+1}) \le \varphi(x^1)\big(\tfrac{H^2\kappa^2}{4\mu^2 K}\textstyle\sum_{k=1}^K \|x^k - x^\star\|^2 + \tfrac{2L\rho_K}{K}\big)^K, \quad (9)$$

*where $\rho_K$ is the regret bound defined in (5).*

Theorem 3.3 justifies our observation of superlinear convergence in Figure 1a: for strongly convex quadratics, the Hessian Lipschitz constant is zero ($H = 0$) and (9) guarantees the superlinear convergence at rate $\mathcal{O}((\frac{\rho_K}{K})^K) = \mathcal{O}((\frac{1}{\sqrt{K}})^K)$. For general strongly convex objectives, global linear convergence (Theorem 3.1) implies $\lim_{K\to\infty} \frac{1}{K}\sum_{k=1}^K \|x^k - x^\star\|^2 = 0$ when $\eta_k = \mathcal{O}(1/\sqrt{k})$. So eventually the first term in (9) vanishes, giving superlinear convergence. This superlinear convergence behavior demonstrates that HDM can perform significantly better than standard adaptive first-order methods and line search. HDM represents a new family of first-order methods that achieves superlinear convergence on strongly convex objectives, following the celebrated quasi-Newton (QN) family (Nocedal and Wright, 1999; Fletcher, 2000). Table 1 summarizes the superlinear convergence rates of HDM and other QN methods.

Table 1: Recent Superlinear convergence rates

| Algorithm | Rate |
|---|---|
| Greedy QN (Rodomanov and Nesterov, 2021) | $\mathcal{O}(e^{-\frac{1}{2}K^2})$ |
| Broyden family (Rodomanov and Nesterov, 2022) | $\mathcal{O}(e^{-\frac{1}{2}K\log K})$ |
| Online-learning guided QN (Jiang et al., 2023) | $\mathcal{O}(e^{-\frac{1}{2}K\log K})$ |
| BFGS with line-search (Jin et al., 2024a;b) | $\mathcal{O}(e^{-K\log K})$ |
| HDM (This paper) | $\mathcal{O}(e^{-\frac{1}{2}K\log K})$ |

**HDM Learns the Hessian at the Optimum.** In fact, $\{P_k\}$ in HDM will converge to $[\nabla^2 f(x^\star)]^{-1}$ under an assumption similar to one studied in the quasi-Newton literature (Conn et al., 1991; Nocedal and Wright, 1999). Lemma 3.1 quantifies the effect of learning the preconditioner through the distance $\|P_k - [\nabla^2 f(x^\star)]^{-1}\|_F$.

**Lemma 3.1.** *Under the same assumptions as Theorem 3.3, Algorithm 1 generates $\{P_k\}$ such that*

$$\|P_{k+1} - [\nabla^2 f(x^\star)]^{-1}\|_F^2$$
$$\le \|P_k - [\nabla^2 f(x^\star)]^{-1}\|_F^2$$
$$\quad - \tfrac{\mu(\eta - L\eta^2)}{2}\big\|(P_k - [\nabla^2 f(x^\star)]^{-1})\tfrac{\nabla f(x^k)}{\|\nabla f(x^k)\|}\big\|^2$$
$$\quad + (2\eta - L\eta^2)\tfrac{H^2\kappa}{4\mu^3}\|x^k - x^\star\|^2. \quad (10)$$

Relation (10) consists of three terms: the distance $\|P_k - [\nabla^2 f(x^\star)]^{-1}\|_F^2$; a decrement in the distance; and an error term that converges to zero as $x^k \to x^\star$. The decrement is determined by the magnitude of $\big\|(P_k - [\nabla^2 f(x^\star)]^{-1})\tfrac{\nabla f(x^k)}{\|\nabla f(x^k)\|}\big\|^2$, which measures the difference between the operators $P_k$ and $[\nabla^2 f(x^\star)]^{-1}$ in the (unit) gradient direction $\frac{\nabla f(x^k)}{\|\nabla f(x^k)\|}$. To ensure fast convergence, it suffices for $P_k\nabla f(x^k)$ and $[\nabla^2 f(x^\star)]^{-1}\nabla f(x^k)$ to remain sufficiently close. If the set $\big\{\frac{\nabla f(x^k)}{\|\nabla f(x^k)\|}\big\}$ spans the entire space over the iterations, $P_k$ and $[\nabla^2 f(x^\star)]^{-1}$ should align in all directions, leading to convergence of $\{P_k\}$.

**Theorem 3.4** (Convergence of the preconditioner). *Instate the same assumptions as in Lemma 3.1 and let $\eta_k \equiv \eta \in (0, \frac{1}{2L(LD+1)^2\kappa}]$ in online gradient descent (4). Suppose the gradient directions $\big\{\frac{\nabla f(x^k)}{\|\nabla f(x^k)\|}\big\}$ are uniformly independent* [1]. *Then $\lim_{k\to\infty} \|P_k - [\nabla^2 f(x^\star)]^{-1}\| = 0$.*

The convergence of stepsize in HDM was observed experimentally by Baydin et al. (2018) for a scalar stepsize. ($\mathcal{P} \subseteq \mathcal{S}$). Our result theoretically justifies this observation.

**HDM and Quasi-Newton Methods.** Our results identify a similarity between HDM and quasi-Newton methods. Both learn the inverse Hessian operator $g \mapsto [\nabla^2 f(x^\star)]^{-1}g$ as the algorithm progresses, but through different properties of the operator. The quasi-Newton methods use the secant equation $x - y \approx [\nabla^2 f(x^\star)]^{-1}(\nabla f(x) - \nabla f(y))$ for $x, y$ close to $x^\star$ and enforce this equation, replacing the inverse Hessian by $P_k$, to guide learning (Jiang et al., 2023). In contrast, HDM learns an optimal preconditioner for the function. Since the function is locally quadratic, this optimal preconditioner is the inverse Hessian. HDM uses the hypergradient feedback $h_x(P)$ to directly measure

---

[1] The formal definition of a uniformly independent sequence is given in Appendix C.5, which is adapted from quasi-Newton literature (Conn et al., 1991; Nocedal and Wright, 1999)

the quality of the preconditioner and can search for an optimal preconditioner in a given closed convex set $\mathcal{P}$, whereas quasi-Newton methods use the secant equation as an indirect proxy. Both approaches require a safeguard to prevent divergence in the warm-up phase, which is achieved by line-search in quasi-Newton and null step in HDM.

# 4. HDM with Momentum

In this section, we develop two variants of HDM, with heavy-ball momentum (Polyak, 1964) and with Nesterov momentum (Nesterov, 1983).

## 4.1. Heavy-ball Momentum

Heavy-ball method is a practical acceleration technique:

$$x^{k+1} = x^k - P_k \nabla f(x^k) + B_k(x^k - x^{k-1}). \quad (11)$$

The momentum parameter $B_k$ is typically chosen as a scalar $B_k = \beta_k I$ with $\beta_k > 0$. HDM can learn a matrix momentum $B_k \in \mathcal{B} \subseteq \mathbb{R}^{n \times n}$ with convergence guarantees (Theorem 4.1) when $\mathcal{B}$ satisfies this assumption:

**A5:** Closed convex set $\mathcal{B}$ satisfies $\frac{1}{2}I \in \mathcal{B}$, $\text{diam}(\mathcal{B}) \leq D$.

HDM can *jointly* learn the pair $(P_k, B_k)$ using the modified feedback function

$$h_{x,x^-}(P,B) := \frac{\psi(x^+(P,B),x) - \psi(x,x^-)}{\|\nabla f(x)\|^2 + \frac{\tau}{2}\|x - x^-\|^2}, \quad (12)$$

where $\psi$ is the potential function for heavy-ball momentum defined by $\psi(x,x^-) := f(x) + \frac{\omega}{2}\|x - x^-\|^2$ (Danilova et al., 2020); $x^+(P,B) := x - P\nabla f(x) + B(x - x^-)$ updates $x$; and $\omega > 0$ and $\tau > 0$ are constants. Algorithm 2 presents the resulting method, HDM-HB, which uses HDM, heavy-ball momentum, and a null step to ensure decrease of the potential function $\psi$. Figure 3a compares non-adaptive heavy-ball ($P_k \equiv \alpha I, B_k \equiv \beta I$) against HDM-HB with full-matrix/diagonal preconditioner and scalar momentum.

---

**Algorithm 2** HDM with heavy-ball momentum (HDM-HB)

**input** initial point $x^0 = x^1, \eta_p, \eta_b > 0, P_1, B_1$
**for** $k = 1, 2,...$ **do**
$\quad x^{k+1/2} = x^k - P_k \nabla f(x^k) + B_k(x^k - x^{k-1})$
$\quad P_{k+1} = \Pi_{\mathcal{P}}[P_k - \eta_p \nabla_P h_{x^k,x^{k-1}}(P_k, B_k)]$
$\quad B_{k+1} = \Pi_{\mathcal{B}}[B_k - \eta_b \nabla_B h_{x^k,x^{k-1}}(P_k, B_k)]$
$\quad (x^{k+1}, x^k) = \underset{(x^+,x)\in\{(x^k,x^{k-1}),(x^{k+1/2},x^k)\}}{\arg\min} \psi(x^+, x)$
**end**
**output** $x^{K+1}$

---

**Theorem 4.1** (Convergence of HDM-HB). *Instate $\varphi(x^k)$ in* (6). *Under* **A1**, **A3** *and* **A5**, *Algorithm 2 with* $\eta_b, \eta_p = \mathcal{O}(1/\sqrt{K})$ *or* $\mathcal{O}(1/\sqrt{k})$ *satisfies*

$$\varphi(x^{K+1}) \leq \frac{\varphi(x^1)}{KV[\gamma_K^\star - \frac{\rho_K}{K}]_+ + 1},$$

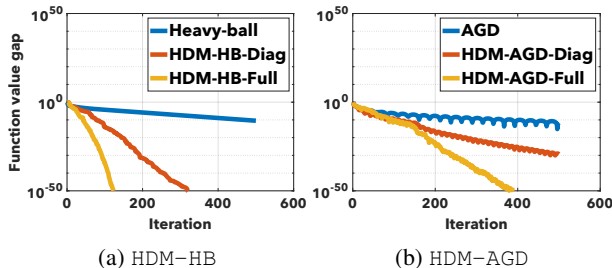

(a) HDM-HB      (b) HDM-AGD

Figure 3: The convergence behavior of HDM-HB and HDM-AGD on a toy quadratic problem. Figure 3a: HDM-HB. Figure 3b: HDM with Nesterov momentum.

*where* $\gamma_K^\star := -\min_{(P,B)\in\mathcal{P}\times\mathcal{B}} \frac{1}{K}\sum_{k=1}^K h_{x^k,x^{k-1}}(P,B)$ *depends on the iteration trajectory* $\{x^k\}_{k\leq K}$; $\rho_K = \mathcal{O}(\sqrt{K})$ *is the regret with respect to feedback* (12); $V := \min\{\frac{\varphi(x^1)}{4\Delta^2}, \frac{\tau}{4\omega}\}$; $\Delta$ *is defined in Lemma 2.4.*

## 4.2. Nesterov Momentum

HDM can also improve accelerated gradient descent AGD (a.k.a. Nesterov momentum): for a given sequence $\{A_k\}$,

$$\begin{aligned} y^k &= x^k + (1 - \tfrac{A_k}{A_{k+1}})(z^k - x^k) \\ x^{k+1} &= y^k - \tfrac{1}{L}\nabla f(y^k) \\ z^{k+1} &= z^k + \tfrac{A_{k+1} - A_k}{L}\nabla f(y^k). \end{aligned} \quad (13)$$

HDM can learn a preconditioner $P_k$ that replaces $\frac{1}{L}$ to accelerate the gradient step (13) in AGD. We call the resulting algorithm HDM-AGD. It achieves $\mathcal{O}(1/K^2 + \rho_K/K^3)$ convergence rate and an empirical speedup (Figure 3b). Further discussion of HDM-AGD appears in Appendix D.2.

**Theorem 4.2** (Informal). *Instate $\varphi(x^k)$ in* (6). *Under* **A1**, **A3**, *HDM-AGD with proper initialization gives* $\varphi(x^{K+1}) \leq \mathcal{O}(\frac{\lambda_K^\star}{K^2} + \frac{\rho_K}{K^3})$, *where* $\lambda_K^\star \leq 2L$ *depends on* $\gamma_K^\star$.

# 5. Experiments

This section conducts numerical experiments to validate the empirical performance of hypergradient descent. We compare HDM-Best (see Section 5.1 below) with different adaptive optimization algorithms.

## 5.1. Efficient and Practical Variant: HDM-Best

This section highlights the major components of our most competitive variant HDM-Best. The algorithm and a more detailed explanation are available in Appendix A. The implementation is available at https://github.com/udellgroup/hypergrad.

**Diagonal Preconditioner and Heavy-ball Momentum.** HDM-Best updates $x$ by (11) with diagonal preconditioner

$\mathcal{P} \subseteq \mathcal{D}$ and scalar momentum $\mathcal{B} = \{\beta I : \beta \in \mathbb{R}\}$. This choice balances practical efficiency and implementation complexity. Boundedness of $\mathcal{P}$ does not greatly impact the performance, while the bound on $\mathcal{B}$ can significantly change algorithm behavior. Two empirically robust ranges for $\mathcal{B}$ are $[0, 0.9995]$ and $[-0.9995, 0.9995]$.

**`AdaGrad` for Online Learning.** `HDM-Best` uses `AdaGrad` to shorten the warm-up phase for learning of $(P_k, \beta_k)$ (see Section 3.2). `AdaGrad` usually yields faster convergence of `HDM` than online gradient descent at the cost of additional memory of size $n$.

### 5.2. Dataset and Testing Problems

We test `HDM-Best` on deterministic convex problems. We adopt two convex optimization tasks in machine learning: support vector machine (Lee and Mangasarian, 2001) and logistic regression (Hastie, 2009). The testing datasets are obtained from `LIBSVM` (Chang and Lin, 2011).

### 5.3. Experiment Setup

**Algorithm Benchmark.** We benchmark the following algorithms. More details appear in the appendix Table 3.

- `GD`. Vanilla gradient descent.
- `GD-HB`. Gradient descent with heavy-ball momentum.
- `AGD-CVX`. The smooth convex version of accelerated gradient descent (Nesterov momentum).
- `AGD-SCVX`. The smooth strongly convex version of accelerated gradient descent.
- `Adam`. Adaptive momentum estimation.
- `AdaGrad`. Adaptive (sub)gradient method.
- `BFGS`. BFGS from `scipy`.
- `L-BFGS-Mk`. L-BFGS with memory size k in `scipy`.
- Practical variant `HDM-Best` uses as memory 7 vectors of size $n$, comparable to memory for `L-BFGS-M1`.

**Algorithm Configuration.** See Appendix A for details.

- For `HDM-Best`, we search for the optimal $\eta_p$ within $\{0.1/L, 1/L, 10/L, 100/L\}$ and $\eta_b \in \{1, 3, 5, 10, 100\}$.
- Stepsize in `GD`, `GD-HB`, `AGD-CVX`, and `AGD-SCVX` are all set to $1/L$.
- The momentum parameter in `GD-HB` is chosen within the set $\{0.1, 0.5, 0.9, 0.99\}$.
- The `Adam` stepsize is chosen within the set $\{1/L, 10^{-3}, 10^{-2}, 10^{-1}, 1, 10\}$. $\beta_1 = 0.9, \beta_2 = 0.999$.
- The `AdaGrad` stepsize is chosen within the set $\{1/L, 10^{-3}, 10^{-2}, 10^{-1}, 1, 10\}$.
- `BFGS`, `L-BFGS-Mk` use default parameters in `scipy`.

**Testing Configurations.**

1) *Maximum oracle access.* We allow a maximum of 1000 gradient oracles for each algorithm.
2) *Initial point.* All the algorithms are initialized from the same starting point generated from normal distribution $\mathcal{N}(0, I_n)$ and normalized to have unit length.
3) *Stopping criterion.* Algorithms stop if $\|\nabla f\|_\infty \leq 10^{-4}$.

Table 2: Number of solved problems for each algorithm.

| Algorithm/Problem | SVM (33 ↑) | Logistic Regression (33 ↑) |
|---|---|---|
| GD | 5 | 2 |
| GD-HB | 9 | 7 |
| AGD-CVX | 8 | 3 |
| AGD-SCVX | 7 | 6 |
| Adam | 26 | 11 |
| AdaGrad | 9 | 8 |
| L-BFGS-M1 | 13 | 11 |
| L-BFGS-M3 | 20 | 14 |
| L-BFGS-M5 | 26 | 16 |
| L-BFGS-M10 | 31 | 18 |
| BFGS | 32 | 26 |
| HDM-Best | 32 | 21 |

For each algorithm, we record the number of successfully solved instances ($\|\nabla f\|_\infty \leq 10^{-4}$ within 1000 gradient oracles). Table 2 summarizes the detailed statistics. The number of instances solved by `HDM-Best` is comparable to that of `L-BFGS-M10`.

**Support Vector Machine.** Figure 4 shows the function value gap and gradient norm plots on sample test instances on support vector machine problems. The optimal value for each instance is obtained by running `BFGS` until $\|\nabla f\|_\infty \leq 10^{-4}$. We see that the practical variant of `HDM-Best` achieves a significant speedup over other adaptive first-order methods. In particular, `HDM-Best` often matches `L-BFGS-M5` and `L-BFGS-M10`, while its memory usage is closer to `L-BFGS-M1`. Notably, `Adam` also achieves competitive performance in several instances.

**Logistic Regression.** In logistic regression (Figure 5), `HDM-Best` still compares well with `L-BFGS-M5` and is significantly faster than other adaptive first-order methods.

Overall, `HDM-Best` demonstrates superior performance on deterministic convex problems and is comparable with the mature `L-BFGS` family. We believe that further development of `HDM` will fully unleash its potential for a broad range of optimization tasks.

## 6. Conclusion

This paper addresses the long-standing challenge of establishing convergence of the hypergradient descent heuristic.

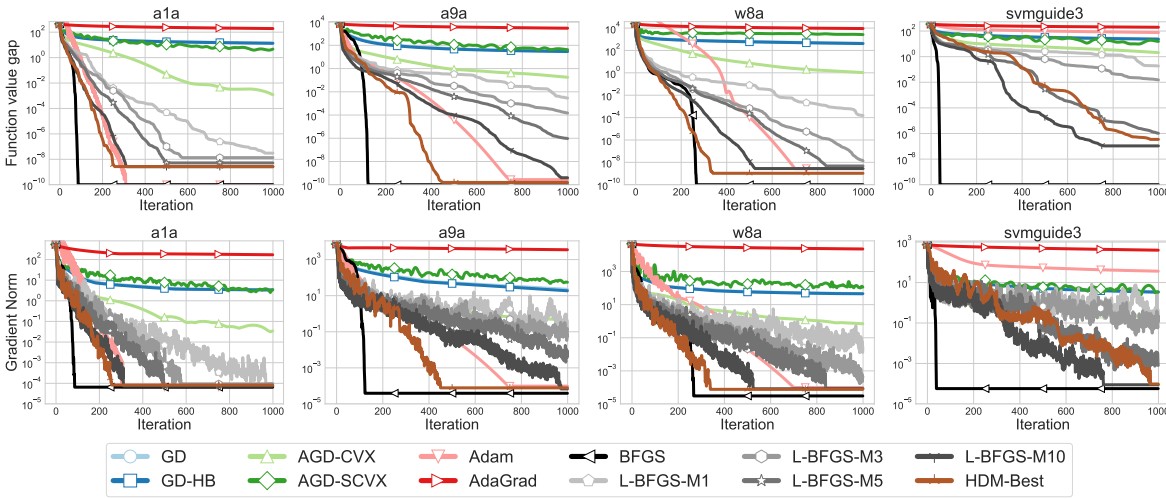

Figure 4: Experiments on support vector-machine problem. First row: function value gap. Second row: gradient norm

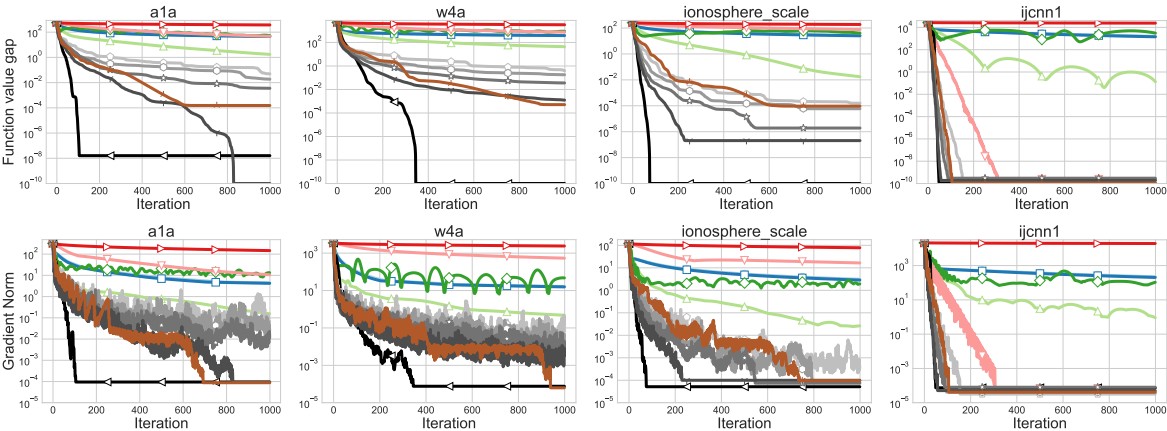

Figure 5: Experiments on logistic regression problems. First row: function value gap. Second row: gradient norm

We provide the first rigorous theoretical foundation for hypergradient descent and introduce a novel online learning perspective that extends to other first-order methods with adaptive hyperparameter updates. Our theoretical advances support effective and scalable enhancements that allow the (first-order) `HDM` to achieve superlinear convergence with guarantees that resemble quasi-Newton methods. Building on these results, we propose `HDM-Best`, an efficient variant of `HDM` that performs competitively with the widely used `L-BFGS` method on convex problems. This empirical success positions `HDM` as a compelling alternative for modern machine learning. Extending the theory of `HDM` to stochastic and nonconvex optimization is a crucial next step to understanding its potential to speed up the training of large-scale models.

## Acknowledgments

MU, YC, and WG gratefully acknowledge support from the National Science Foundation (NSF) Award IIS-2233762, the Office of Naval Research (ONR) Awards N000142212825, N000142412306, and N000142312203, the Alfred P. Sloan Foundation, and from IBM Research as a founding member of Stanford Institute for Human-centered Artificial Intelligence (HAI).

## Impact Statement

This paper addresses the convergence behavior of the hypergradient descent heuristic. There are many potential societal consequences of our work, none of which we feel must be specifically highlighted here.

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

# Appendix

## Table of Contents

**Structure of the Appendix.** The appendix is organized as follows. In Appendix A, we introduce a practical variant of hypergradient descent and explain its implementation details. Appendix E provides additional experiment details on the tested problems. Appendix B to Appendix D provide proofs of the main results in the paper.

# A. `HDM` in Practice

This section introduces `HDM-Best`, our recommended practical hypergradient descent method. This variant is adapted from `HDM-HB`, with simplifications to reduce the implementation complexity. The algorithm is given in Algorithm 3.

---

**Algorithm 3** `HDM-Best`

---

**input** starting point $x^0 = x^1$, $\mathcal{P} = \mathbb{S}_+^n \cap \mathcal{D}$, $\mathcal{B} = [0, 0.9995]$, initial diagonal preconditioner $P_1 \in \mathbb{S}_+^n \cap \mathcal{D}$, initial scalar momentum parameter $\beta_1 = 0.95$, `AdaGrad` stepsize $\eta_p, \eta_b > 0$, `AdaGrad` diagonal matrix $U_1 = 0$, `AdaGrad` momentum scalar $v_1 = 0, \tau > 0$

**for** $k = 1, 2,...$ **do**

$$x^{k+1/2} = x^k - P_k \nabla f(x^k) + \beta_k(x^k - x^{k-1})$$

$$\nabla_P h_{x^k, x^{k-1}}(P_k, \beta_k) = \frac{\text{diag}(\nabla f(x^{k+1/2}) \circ \nabla f(x^k))}{\|\nabla f(x^k)\|^2 + \frac{\tau}{2}\|x^k - x^{k-1}\|^2} \quad \text{\# Element-wise product}$$

$$\nabla_\beta h_{x^k, x^{k-1}}(P_k, \beta_k) = \frac{\langle \nabla f(x^{k+1/2}), x^k - x^{k-1} \rangle}{\|\nabla f(x^k)\|^2 + \frac{\tau}{2}\|x^k - x^{k-1}\|^2} \quad \text{\# Inner product}$$

$$U_{k+1} = U_k + \nabla_P h_{x^k, x^{k-1}}(P_k, \beta_k) \circ \nabla_P h_{x^k, x^{k-1}}(P_k, \beta_k) \quad \text{\# Diagonal matrix}$$

$$v_{k+1} = v_k + \nabla_\beta h_{x^k, x^{k-1}}(P_k, \beta_k) \cdot \nabla_\beta h_{x^k, x^{k-1}}(P_k, \beta_k) \quad \text{\# Scalar matrix}$$

$$P_{k+1} = \Pi_{\mathbb{R}_+^n \cap \mathcal{D}}[P_k - \eta_p U_{k+1}^{-1/2} \nabla_P h_{x^k, x^{k-1}}(P_k, \beta_k)] \quad \text{\# Diagonal matrix}$$

$$\beta_{k+1} = \Pi_{[0, 0.9995]}[\beta_k - \eta_b v_{k+1}^{-1/2} \nabla_\beta h_{x^k, x^{k-1}}(P_k, \beta_k)]$$

$$x^{k+1} = \underset{x \in \{x^k, x^{k+1/2}\}}{\arg\min} f(x).$$

**end**

**output** $x^{K+1}$

---

We make several remarks about Algorithm 3.

- *Choice of online learning algorithm.* Unless $f(x)$ is quadratic, adaptive online learning algorithms such as `AdaGrad` often significantly outperform online gradient descent with constant stepsize. Note that `AdaGrad` introduces additional memory of size $n$ to store the diagonal online learning preconditioner $U$.

- *Sensitivity of parameters.* The two stepsize parameters in `AdaGrad` are the most important algorithm parameters: $\eta_p, \eta_b$. According to the experiments, $\eta_p$ should be set proportional to $1/L$, the smoothness constant, while an aggressive choice of $\eta_b \in \{1, 10, 100\}$ often yields fast convergence. A local estimator of the smoothness constant $L$ can significantly enhance algorithm performance.

- *Heavy-ball feedback and null step.* In practice, it is observed that dropping the $\frac{\omega}{2}\|x^+(P, B) - x\|^2$ in the numerator of heavy-ball feedback (12) often does not affect algorithm performance. Therefore, in Algorithm 3 the hypergradient with respect to $\frac{\omega}{2}\|x^+(P, B) - x\|^2$ is ignored. On the other hand, the $\frac{\tau}{2}\|x^+(P, B) - x\|^2$ term in the denominator smoothes the update of $\beta_k$ and can strongly affect convergence. The parameter $\tau$ should be taken to be proportional to $L^2$ according to the discussions in Appendix D.1. The null step is taken with respect to the function value $f(x)$ instead of the heavy-ball potential function.

- *Memory usage.* The memory usage of `HDM-Best`, measured in terms of number of vectors of length $n$ is $7n$: 1) three vectors store primal iterates $x^-, x, x^+$. 2) Two vectors store past and buffer gradients $\nabla f(x), \nabla f(x^+)$. 3) A vector stores the diagonal preconditioner $P_k$. 4) A vector stores the `AdaGrad` stepsize matrix $U$.

- *Computational cost.* The major additional computation cost arises from computing hypergradient $\nabla h$, which involves one element-wise product and one inner product for vectors of size $n$. In addition, `HDM-Best` needs to maintain a diagonal matrix for `AdaGrad`. The overall additional computational cost is several $\mathcal{O}(n)$ operations.

# B. Proof of Results in Section 2

## B.1. Auxiliary Results

**Lemma B.1** (Sublinear static regret). *Given a family of convex and $\gamma$-Lipschitz losses $\{h_k\}$, online gradient descent $P_{k+1} = \Pi_{\mathcal{P}}[P_k - \eta_k \nabla h_k(P_k)]$ with stepsize $\eta_k \equiv \frac{D}{2\gamma\sqrt{K}}$ or $\eta_k = \frac{D}{2\gamma\sqrt{k}}$ generates $\{P_k\}$ such that*

$$\sum_{k=1}^{K} h_k(P_k) - \sum_{k=1}^{K} h_k(P) \leq 2\gamma D\sqrt{K}$$

*for any $P \in \mathcal{P}$.*

The result follows from a standard dynamic regret analysis from online convex optimization literature, and we adapt the proof for our analysis. For any $P \in \mathcal{P}$, we deduce

$$
\begin{aligned}
\|P_{k+1} - P\|_F^2 &= \|\Pi_{\mathcal{P}}[P_k - \eta_k \nabla h_k(P_k)] - P\|_F^2 \\
&\leq \|P_k - P - \eta_k \nabla h_k(P_k)\|_F^2 \quad (14) \\
&\leq \|P_k - P\|_F^2 - 2\eta_k \langle \nabla h_k(P_k), P_k - P \rangle + \eta_k^2 \|\nabla h_k(P_k)\|_F^2 \\
&\leq \|P_k - P\|_F^2 - 2\eta_k [h_k(P_k) - h_k(P)] + \eta_k^2 \gamma^2, \quad (15)
\end{aligned}
$$

where (14) uses non-expansiveness of orthogonal projection; (15) applies convexity and $\gamma$-Lipschitz continuity of $h_k$. Re-arranging, we get

$$h_k(P_k) - h_k(P) \leq \frac{1}{2\eta_k}[\|P_k - P\|_F^2 - \|P_{k+1} - P\|_F^2] + \frac{\eta_k}{2}\gamma^2.$$

Suppose $\eta_k$ is non-increasing, and we have

$$
\begin{aligned}
\sum_{k=1}^{K} h_k(P_k) - h_k(P) &\leq \sum_{k=1}^{K}[\frac{1}{2\eta_k}\|P_k - P\|_F^2 - \frac{1}{2\eta_k}\|P_{k+1} - P\|_F^2] + \frac{\gamma^2}{2}\sum_{k=1}^{K}\eta_k \\
&\leq \frac{1}{2\eta_1}\|P_1 - P\|_F^2 + \frac{1}{2}\sum_{k=1}^{K-1}(\frac{1}{\eta_{k+1}} - \frac{1}{\eta_k})\|P_{k+1} - P\|_F^2 + \frac{\gamma^2}{2}\sum_{k=1}^{K}\eta_k \\
&\leq \frac{1}{2\eta_1}D^2 + \frac{1}{2}\sum_{k=1}^{K-1}(\frac{1}{\eta_{k+1}} - \frac{1}{2\eta_k})D^2 + \frac{\gamma^2}{2}\sum_{k=1}^{K}\eta_k \\
&\leq \frac{D^2}{2\eta_K} + \gamma^2 \sum_{k=1}^{K} \frac{\eta_k}{2} \leq 2\gamma D\sqrt{K}
\end{aligned}
$$

since $\sum_{k=1}^{K} \frac{1}{\sqrt{k}} \leq 2\sqrt{K}$. This completes the proof.

**Lemma B.2** (Sublinear dynamic regret (Hazan et al., 2016)). *Given a family of convex and $\gamma$-Lipschitz losses $\{h_k\}$, online gradient descent $P_{k+1} = \Pi_{\mathcal{P}}[P_k - \eta \nabla h_k(P_k)]$ with constant stepsize $\eta = \frac{D}{2\gamma\sqrt{K}}$ generates stepsizes $\{P_k\}$ such that*

$$\sum_{k=1}^{K} h_k(P_k) - h_k(\hat{P}_k) \leq \frac{7\gamma D}{\sqrt{K}} + \frac{2\gamma}{\eta}\mathsf{PL}(\{\hat{P}_k\}). \quad (16)$$

*where $\{\hat{P}_k\}, \hat{P}_k \in \mathcal{P}$ are arbitrarily chosen competitor stepsizes and $\mathsf{PL}(\{\hat{P}_k\}) := \sum_{k=1}^{K-1} \|\hat{P}_k - \hat{P}_{k+1}\|_F$ is the path length of the competitors.*

*Proof.* Starting from (15), we let $P = \hat{P}_k$ and re-arrange to get

$$
\begin{aligned}
h_k(P_k) - h_k(\hat{P}_k) &\leq \frac{1}{2\eta}[\|P_k - \hat{P}_k\|_F^2 - \|P_{k+1} - \hat{P}_k\|_F^2] + \frac{\eta}{2}\gamma^2 \\
&= \frac{1}{2\eta}[\|P_k\|_F^2 - \|P_{k+1}\|_F^2 + 2\langle \hat{P}_k, P_{k+1} - P_k \rangle] + \frac{\eta}{2}\gamma^2
\end{aligned}
$$

Telescoping, we get

$$
\begin{aligned}
\sum_{k=1}^{K} h_k(P_k) - h_k(\hat{P}_k) &\leq \frac{\|P_1\|_F^2}{2\eta} + \frac{\eta}{2}\gamma^2 K + \frac{1}{\eta}\sum_{k=1}^{K}\langle \hat{P}_k, P_{k+1} - P_k \rangle \\
&= \frac{\|P_1\|_F^2}{2\eta} + \frac{\eta}{2}\gamma^2 K + \frac{1}{\eta}\sum_{k=1}^{K-1}\langle P_{k+1}, \hat{P}_k - \hat{P}_{k+1} \rangle + \frac{1}{\eta}\langle \hat{P}_K, P_{K+1} \rangle - \frac{1}{\eta}\langle \hat{P}_1, P_1 \rangle \quad (17) \\
&\leq \frac{3D^2}{\eta} + \frac{\eta}{2}\gamma^2 K + \frac{D}{\eta}\sum_{k=1}^{K-1}\|\hat{P}_k - \hat{P}_{k+1}\|_F \quad (18) \\
&= \frac{3D^2}{\eta} + \frac{\eta}{2}\gamma^2 K + \frac{D}{\eta}\mathsf{PL}(\{\hat{P}_k\}),
\end{aligned}
$$

where (17) is by re-arrangement: $\sum_{k=1}^{K}\langle \hat{P}_k, P_{k+1} - P_k \rangle = \sum_{k=1}^{K-1}\langle P_{k+1}, \hat{P}_k - \hat{P}_{k+1} \rangle + \langle \hat{P}_K, P_{K+1} \rangle - \langle \hat{P}_1, P_1 \rangle$; (19) applies $\text{diam}(\mathcal{P}) < D$ and Cauchy-Schwarz $\langle P_{k+1}, \hat{P}_k - \hat{P}_{k+1} \rangle \leq \|P_{k+1}\|_F \|\hat{P}_k - \hat{P}_{k+1}\|_F \leq D\|\hat{P}_k - \hat{P}_{k+1}\|_F$. Plugging in $\eta = \frac{D}{2\gamma\sqrt{K}}$ completes the proof. $\square$

**Lemma B.3** (Logarithmic static regret (Orabona, 2019)). *Given a family of $\mu$-strongly convex and $\gamma$-Lipschitz losses $\{h_k\}$, online gradient descent $P_{k+1} = \Pi_{\mathcal{P}}[P_k - \eta_k \nabla h_k(P_k)]$ with stepsize $\eta_k = 1/(\mu k)$ generates a sequence of scaling matrices $\{P_k\}$ such that $\sum_{k=1}^{K} h_k(P_k) - h_k(P) \leq \frac{1}{2}\gamma^2 \log K$.*

*Proof.* Using strong convexity, we have $h_k(P) \geq h_k(P_k) + \langle \nabla h_k(P_k), P - P_k \rangle + \frac{\mu}{2}\|P - P_k\|_F^2$ and

$$
\begin{aligned}
\|P_{k+1} - P\|_F^2 &\leq \|P_k - P\|_F^2 - 2\eta_k \langle \nabla h_k(P_k), P_k - P \rangle + \eta_k^2 \gamma^2 \\
&\leq \|P_k - P\|_F^2 - 2\eta_k[h_k(P_k) - h_k(P)] + \eta_k^2 \gamma^2 - \mu\eta_k\|P - P_k\|_F^2 \\
&= \tfrac{k-1}{k}\|P_k - P\|_F^2 - \tfrac{2}{k\mu}[h_k(P_k) - h_k(P)] + \tfrac{\gamma^2}{k^2\mu^2},
\end{aligned}
\tag{19}
$$

where (19) plugs in $\eta_k = 1/(\mu k)$. Re-arranging the terms,

$$
h_k(P_k) - h_k(P) \leq \tfrac{\mu}{2}[(k-1)\|P_k - P\|_F^2 - k\|P_{k+1} - P\|_F^2] + \tfrac{\gamma^2}{2k\mu}
$$

and telescoping gives $\sum_{k=1}^{K} h_k(P_k) - h_k(P) \leq \sum_{k=1}^{K} \frac{\gamma^2}{2k\mu} \leq \frac{\gamma^2}{2\mu}(\log K + 1)$, which completes the proof. $\qquad \square$

## B.2. Proof of Lemma 2.1

Consider the first property. Convexity and smoothness follow directly from (Gao et al., 2024). To verify strong convexity, note that for $h_x(\alpha) = \frac{f(x - \alpha\nabla f(x)) - f(x^\star)}{\|\nabla f(x)\|^2}$

$$
h_x''(\alpha) = \frac{\mathrm{d}}{\mathrm{d}\alpha}\left[ \frac{\langle \nabla f(x - \alpha\nabla f(x)), \nabla f(x) \rangle}{\|\nabla f(x)\|^2} \right] = \left\langle \frac{\nabla f(x)}{\|\nabla f(x)\|}, \nabla^2 f(x)\frac{\nabla f(x)}{\|\nabla f(x)\|} \right\rangle \geq \mu
$$

since $\nabla^2 f(x) \succeq \mu I$ and $x \notin \mathcal{X}^\star$. This completes the proof of the first property.

Next, we consider the second property. Lipschitz continuity also follows from (Gao et al., 2024). To verify exp-concavity, recall that a twice-differentiable function $h$ is $\beta$-exp-concave if $\nabla^2 h(x) \succeq \beta\nabla h(x)\nabla h(x)^\top$ for some $\beta \geq 0$. By definition of $\mathcal{D}$,

$$
\nabla h_x(P) = -\frac{\nabla f(x) \circ \nabla f(x - P\nabla f(x))}{\|\nabla f(x)\|^2} = -\frac{\mathrm{diag}(\nabla f(x))\nabla f(x - P\nabla f(x))}{\|\nabla f(x)\|^2}
$$

and $\nabla^2 h_x(P) = \frac{\mathrm{diag}(\nabla f(x))\nabla^2 f(x - P\nabla f(x))\mathrm{diag}(\nabla f(x))}{\|\nabla f(x)\|^2}$. Using $\nabla^2 f(x - P\nabla f(x)) \succeq \mu I$, we deduce that

$$
\begin{aligned}
&\nabla^2 h_x(P) - \beta\nabla h_x(P)\nabla h_x(P)^\top \\
&= \frac{\mathrm{diag}(\nabla f(x))\nabla^2 f(x - P\nabla f(x))\mathrm{diag}(\nabla f(x))}{\|\nabla f(x)\|^2} - \beta\frac{\mathrm{diag}(\nabla f(x))\nabla f(x - P\nabla f(x))\nabla f(x - P\nabla f(x))^\top \mathrm{diag}(\nabla f(x))}{\|\nabla f(x)\|^4} \\
&= \mathrm{diag}\big(\tfrac{\nabla f(x)}{\|\nabla f(x)\|}\big)\Big[\nabla^2 f(x - P\nabla f(x)) - \beta\tfrac{\nabla f(x - P\nabla f(x))}{\|\nabla f(x)\|}\tfrac{\nabla f(x - P\nabla f(x))^\top}{\|\nabla f(x)\|}\Big]\mathrm{diag}\big(\tfrac{\nabla f(x)}{\|\nabla f(x)\|}\big) \\
&\succeq \mathrm{diag}\big(\tfrac{\nabla f(x)}{\|\nabla f(x)\|}\big)\Big[\mu I - \beta\tfrac{\nabla f(x - P\nabla f(x))}{\|\nabla f(x)\|}\tfrac{\nabla f(x - P\nabla f(x))^\top}{\|\nabla f(x)\|}\Big]\mathrm{diag}\big(\tfrac{\nabla f(x)}{\|\nabla f(x)\|}\big),
\end{aligned}
\tag{20}
$$

where (20) uses $\mu$-strong convexity of $f(x)$. Now, it suffices to verify that

$$
\frac{\nabla f(x - P\nabla f(x))}{\|\nabla f(x)\|}\frac{\nabla f(x - P\nabla f(x))^\top}{\|\nabla f(x)\|} \preceq \frac{\mu}{\beta}I
\tag{21}
$$

for all $x \notin \mathcal{X}^\star$. Write $\frac{\nabla f(x - P\nabla f(x))}{\|\nabla f(x)\|} = \frac{\nabla f(x)}{\|\nabla f(x)\|} + \frac{\nabla f(x - P\nabla f(x)) - \nabla f(x)}{\|\nabla f(x)\|}$ and let $z := \nabla f(x - P\nabla f(x)) - \nabla f(x)$, we have, by $L$-smoothness, that $\|z\| \leq L\|P\nabla f(x)\| \leq LD\|\nabla f(x)\|$ and

$$
\begin{aligned}
\left\|\frac{\nabla f(x - P\nabla f(x))}{\|\nabla f(x)\|}\frac{\nabla f(x - P\nabla f(x))^\top}{\|\nabla f(x)\|}\right\| &= \left\|\Big(\frac{\nabla f(x)}{\|\nabla f(x)\|} + \frac{z}{\|\nabla f(x)\|}\Big)\Big(\frac{\nabla f(x)}{\|\nabla f(x)\|} + \frac{z}{\|\nabla f(x)\|}\Big)^\top\right\| \\
&= \left\|\frac{\nabla f(x)\nabla f(x)^\top}{\|\nabla f(x)\|^2} + \frac{z\nabla f(x)^\top}{\|\nabla f(x)\|^2} + \frac{\nabla f(x)z^\top}{\|\nabla f(x)\|^2} + \frac{zz^\top}{\|\nabla f(x)\|^2}\right\| \\
&\leq 1 + \frac{2\|z\|}{\|\nabla f(x)\|} + \frac{\|z\|^2}{\|\nabla f(x)\|^2} = \big(1 + \frac{\|z\|}{\|\nabla f(x)\|}\big)^2 \leq (1 + LD)^2.
\end{aligned}
$$

Hence, for $\beta \leq \frac{\mu}{(1+LD)^2}$ the relation (21) holds. We conclude that $h_x(P) = h_x(d)$ is $\frac{\mu}{(1+LD)^2}$-exponential concave.

### B.3. Proof of Lemma 2.2

We use Lipschitzness from Lemma 2.1 and Lemma B.1 by taking $\gamma = 1 + LD$ and $\eta_k \equiv \frac{D}{(LD+1)\sqrt{K}}$ or $\eta = \frac{D}{(LD+1)\sqrt{k}}$.
Here, we use a slightly loose bound to unify the constants in static and dynamic regret bounds.

### B.4. Proof of Lemma 2.3

We use Lipschitzness and strong convexity from Lemma 2.1 and invoke Lemma B.3 by taking $\gamma = 1 + LD$.

### B.5. Proof of Lemma 2.4

The proof resembles (Gao et al., 2024) and uses a tighter analysis. Consider the optimality measure $f(x^{K+1}) - f(x^\star)$, and we deduce that

$$
\begin{aligned}
f(x^{K+1}) - f(x^\star) &= \frac{1}{\frac{1}{f(x^{K+1})-f(x^\star)}} \\
&= \frac{1}{\sum_{k=1}^{K} \frac{1}{f(x^{k+1})-f(x^\star)} - \frac{1}{f(x^k)-f(x^\star)} + \frac{1}{f(x^1)-f(x^\star)}} \\
&= \frac{1}{\sum_{k=1}^{K} \frac{f(x^k)-f(x^{k+1})}{[f(x^{k+1})-f(x^\star)][f(x^k)-f(x^\star)]} + \frac{1}{f(x^1)-f(x^\star)}} \\
&= \frac{1}{\sum_{k=1}^{K} \frac{\max\{-h_{x^k}(P_k),0\}\|\nabla f(x^k)\|^2}{[f(x^{k+1})-f(x^\star)][f(x^k)-f(x^\star)]} + \frac{1}{f(x^1)-f(x^\star)}}
\end{aligned}
$$

Next, using $f(x) - f(x^\star) \le \|\nabla f(x)\| \cdot \|x - x^\star\|$,

$$
\frac{\max\{-h_{x^k}(P_k),0\}\|\nabla f(x^k)\|^2}{[f(x^{k+1})-f(x^\star)][f(x^k)-f(x^\star)]} \ge \frac{\max\{-h_{x^k}(P_k),0\}\|\nabla f(x^k)\|^2}{[f(x^k)-f(x^\star)]^2} \ge \frac{\max\{-h_{x^k}(P_k),0\}}{\text{dist}(x^k,\mathcal{X}^\star)^2} \ge \frac{\max\{-h_{x^k}(P_k),0\}}{\Delta^2}.
$$

Finally, we deduce that

$$
\begin{aligned}
f(x^{K+1}) - f(x^\star) &\le \frac{\Delta^2}{\sum_{k=1}^{K} \max\{-h_{x^k}(P_k),0\} + \frac{\Delta^2}{f(x^1)-f(x^\star)}} \\
&\le \frac{\Delta^2}{\max\{\sum_{k=1}^{K} -h_{x^k}(P_k),0\} + \frac{\Delta^2}{f(x^1)-f(x^\star)}} \\
&\le \min\left\{ \frac{\Delta^2}{K \max\{\frac{1}{K}\sum_{k=1}^{K} -h_{x^k}(P_k),0\}}, f(x^1) - f(x^\star) \right\}
\end{aligned}
$$

and this completes the proof.

### B.6. Intermediate Iterate Convergence with Adaptive Online Algorithms

Two disadvantages of constant stepsize in online gradient descent is 1) the dependence on the total number of iterations $K$; and 2) no regret guarantee for the intermediate iterates. One simple fix is let $\eta_k = \mathcal{O}(1/\sqrt{k})$ and online gradient descent achieves the same sublinear regret guarantee (up to a constant multiplicative factor) for any $k$ (Orabona, 2019). Similar arguments hold for adaptive gradient methods (Duchi et al., 2011; McMahan and Streeter, 2010).

## C. Proof of Results in Section 3

### C.1. Proof of Theorem 3.1

Plugging (5) from Lemma 2.2 into Lemma 2.4 completes the proof.

### C.2. Proof of Theorem 3.2

Invoking Lipschitzness from Lemma 2.1 and (16) from Lemma B.2 with $\gamma = 1 + LD, \eta = \frac{D}{(LD+1)\sqrt{K}}$ gives

$$
\sum_{k=1}^{K} h_{x^k}(P_k) - h_{x^k}(\hat{P}_k) \le \rho_K + \frac{LD+1}{2}\sqrt{K}\sum_{k=1}^{K-1} \|\hat{P}_k - \hat{P}_{k+1}\|_F.
$$

Plugging the relation into Lemma 2.4 completes the proof.

## C.3. Proof of Theorem 3.3

For Theorem 3.3 and Lemma 3.1 only, we will define the following modified feedback function by replacing $f(x^k)$ in the numerator by $f(x^\star)$:

$$\hat{h}_x(P) := \frac{f(x - P\nabla f(x)) - f(x^\star)}{\|\nabla f(x)\|^2} \geq 0.$$

For a fixed $x$, $\hat{h}_x(P)$ only differs from the original hypergradient feedback by a constant; it has the same properties as the original feedback function, and the algorithm is exactly the same since only the gradient of $\hat{h}_x$ is considered in the algorithm update. Using the definition of $\hat{h}_x(P)$, we deduce that

$$\begin{aligned}
\frac{f(x^{K+1}) - f(x^\star)}{f(x^1) - f(x^\star)} &= \prod_{k=1}^K \frac{f(x^{k+1}) - f(x^\star)}{f(x^k) - f(x^\star)} \\
&\leq \left(\frac{1}{K}\sum_{k=1}^K \frac{f(x^{k+1}) - f(x^\star)}{f(x^k) - f(x^\star)}\right)^K \\
&= \left(\frac{1}{K}\sum_{k=1}^K \min\left\{\frac{\hat{h}_{x^k}(P_k)\|\nabla f(x^k)\|^2}{f(x^k) - f(x^\star)}, 1\right\}\right)^K &(22) \\
&\leq \left(\frac{1}{K}\sum_{k=1}^K \min\{2L\hat{h}_{x^k}(P_k), 1\}\right)^K &(23) \\
&\leq \left(\min\left\{\frac{2L}{K}\sum_{k=1}^K \hat{h}_{x^k}(P_k), 1\right\}\right)^K,
\end{aligned}$$

where (22) plugs in the definition of $\hat{h}_x$; (23) uses $L$-smoothness and that $\hat{h}_x$ is nonnegative. Using Lemma 2.2, we get $\sum_{k=1}^K \hat{h}_{x^k}(P_k) \leq \sum_{k=1}^K \hat{h}_{x^k}(P) + \rho_K$ for any $P \in \mathcal{P}$. Next, we consider the quantity $\hat{h}_x([\nabla^2 f(x^\star)]^{-1})$ and deduce that

$$\begin{aligned}
\hat{h}_x([\nabla^2 f(x^\star)]^{-1}) &= \frac{f(x - [\nabla^2 f(x^\star)]^{-1}\nabla f(x)) - f(x^\star)}{\|\nabla f(x)\|^2} \\
&\leq \frac{\frac{L}{2}\|x - [\nabla^2 f(x^\star)]^{-1}\nabla f(x) - x^\star\|^2}{\|x - x^\star\|^2} \frac{\|x - x^\star\|^2}{\|\nabla f(x)\|^2} &(24) \\
&\leq \frac{L}{2\mu^2} \frac{\|x - [\nabla^2 f(x^\star)]^{-1}\nabla f(x) - x^\star\|^2}{\|x - x^\star\|^2}, &(25)
\end{aligned}$$

where (24) uses $L$-smoothness $f(x) - f(x^\star) \leq \frac{L}{2}\|x - x^\star\|^2$ and (25) uses $\|\nabla f(x)\|^2 \geq \mu^2\|x - x^\star\|^2$. Then,

$$\begin{aligned}
x - [\nabla^2 f(x^\star)]^{-1}\nabla f(x) - x^\star &= x - x^\star - [\nabla^2 f(x^\star)]^{-1}\nabla f(x) \\
&= [\nabla^2 f(x^\star)]^{-1}[\nabla^2 f(x^\star)(x - x^\star) - (\nabla f(x) - \nabla f(x^\star))]
\end{aligned}$$

since $\nabla f(x^\star) = 0$. Plugging in $\nabla f(x) - \nabla f(x^\star) = \int_0^1 \nabla^2 f(x^\star + t(x - x^\star))(x - x^\star)\mathrm{d}t$, we deduce that

$$\begin{aligned}
\|\nabla^2 f(x^\star)(x - x^\star) - (\nabla f(x) - \nabla f(x^\star))\| &= \|\nabla^2 f(x^\star)(x - x^\star) - \int_0^1 \nabla^2 f(x^\star + t(x - x^\star))(x - x^\star)\mathrm{d}t\| \\
&= \|\int_0^1 [\nabla^2 f(x^\star) - \nabla^2 f(x^\star + t(x - x^\star))](x - x^\star)\mathrm{d}t\| \\
&\leq \int_0^1 tH\|x - x^\star\|^2\mathrm{d}t = \frac{H}{2}\|x - x^\star\|^2, &(26)
\end{aligned}$$

where (26) uses $H$-Lipschitz continuity of $\nabla^2 f(x)$ and, consequently,

$$\begin{aligned}
&\|x - [\nabla^2 f(x^\star)]^{-1}\nabla f(x) - x^\star\| \\
&= \|[\nabla^2 f(x^\star)]^{-1}[\nabla^2 f(x^\star)(x - x^\star) - (\nabla f(x) - \nabla f(x^\star))]\| \leq \frac{H}{2\mu}\|x - x^\star\|^2 &(27)
\end{aligned}$$

since $\nabla^2 f(x^\star) \succeq \mu I$ due to strong convexity. Plugging the relation back, we get

$$\hat{h}_x([\nabla^2 f(x^\star)]^{-1}) \leq \frac{L}{2\mu^2} \frac{\frac{H^2}{4\mu^2}\|x - x^\star\|^4}{\|x - x^\star\|^2} = \frac{H^2\kappa}{8\mu^3}\|x - x^\star\|^2. \quad (28)$$

Since $[\nabla^2 f(x^\star)]^{-1} \in \mathcal{P}$ by assumption,

$$\sum_{k=1}^K \hat{h}_{x^k}(P_k) \leq \sum_{k=1}^K \hat{h}_{x^k}([\nabla^2 f(x^\star)]^{-1}) + \rho_K \leq \frac{H^2\kappa}{8\mu^3}\sum_{k=1}^K \|x^k - x^\star\|^2 + \rho_K,$$

and we get

$$f(x^{K+1}) - f(x^\star) \leq [f(x^1) - f(x^\star)]\left(\min\left\{\frac{H^2\kappa^2}{4\mu^2 K}\sum_{k=1}^K \|x^k - x^\star\|^2 + \frac{2L\rho_K}{K}, 1\right\}\right)^K,$$

which completes the proof.

## C.4. Proof of Lemma 3.1

For brevity let $P^\star = [\nabla^2 f(x^\star)]^{-1}$. We have, according to the update of online gradient descent, that,

$$
\begin{aligned}
\|P_{k+1} - P^\star\|_F^2 &= \|\Pi_{\mathcal{P}}[P_k - \eta \nabla \hat{h}_{x^k}(P_k) - P^\star]\|_F^2 \\
&\leq \|P_k - \eta \nabla \hat{h}_{x^k}(P_k) - P^\star\|_F^2 \\
&= \|P_k - P^\star\|_F^2 - 2\eta \langle \nabla \hat{h}_{x^k}(P_k), P_k - P^\star \rangle + \eta^2 \|\nabla \hat{h}_{x^k}(P_k)\|_F^2 \\
&\leq \|P_k - P^\star\|_F^2 - 2\eta [\hat{h}_{x^k}(P_k) - \hat{h}_{x^k}(P^\star)] + 2L\eta^2 [\hat{h}_{x^k}(P_k) - \inf_{P \in \mathbb{R}^{n \times n}} \hat{h}_{x^k}(P)] \quad (29) \\
&= \|P_k - P^\star\|_F^2 - 2\eta [\hat{h}_{x^k}(P_k) - \hat{h}_{x^k}(P^\star)] + 2L\eta^2 [\hat{h}_{x^k}(P_k) - \hat{h}_{x^k}(P^\star)] + 2L\eta^2 \hat{h}_{x^k}(P^\star) \quad (30) \\
&= \|P_k - P^\star\|_F^2 - 2\eta(1 - \eta L)[\hat{h}_{x^k}(P_k) - \hat{h}_{x^k}(P^\star)] + 2L\eta^2 \hat{h}_{x^k}(P^\star), \quad (31)
\end{aligned}
$$

where (29) uses $L$-smoothness and $\inf_{P \in \mathbb{R}^{n \times n}} \hat{h}_x(P) = 0$ for all $x \notin \mathcal{X}^\star$; (30) is a simple re-arrangement.

Next we lower bound $\hat{h}_{x^k}(P_k) - \hat{h}_{x^k}(P^\star)$. Using strong convexity,

$$
\begin{aligned}
&f(x^k - P_k \nabla f(x^k)) - f(x^k - P^\star \nabla f(x^k)) \\
&= f(x^k - P_k \nabla f(x^k)) - f(x^\star) + f(x^\star) - f(x^k - P^\star \nabla f(x^k)) \\
&\geq \tfrac{\mu}{2} \|x^k - x^\star - P_k \nabla f(x^k)\|^2 + f(x^\star) - f(x^k - P^\star \nabla f(x^k)), \quad (32)
\end{aligned}
$$

where (32) uses $f(x) - f(x^\star) \geq \tfrac{\mu}{2}\|x - x^\star\|^2$. The first term can be bounded as follows:

$$
\begin{aligned}
&\|x^k - x^\star - P_k \nabla f(x^k)\|^2 \\
&= \|x^k - P^\star \nabla f(x^k) - x^\star + (P^\star - P_k) \nabla f(x^k)\|^2 \\
&= \|x^k - P^\star \nabla f(x^k) - x^\star\|^2 + 2\langle x^k - P^\star \nabla f(x^k) - x^\star, (P^\star - P_k)\nabla f(x^k)\rangle + \|(P^\star - P_k)\nabla f(x^k)\|^2 \\
&\geq \tfrac{1}{2}\|(P^\star - P_k)\nabla f(x^k)\|^2 - \|x^k - P^\star \nabla f(x^k) - x^\star\|^2,
\end{aligned}
$$

where we use the inequality $2\langle a, b \rangle \geq -\theta \|a\|^2 - \theta^{-1}\|b\|^2$ with $\theta = 2$. Plugging the relation back into (32) and dividing both sides by $\|\nabla f(x^k)\|^2$,

$$
\begin{aligned}
\hat{h}_{x^k}(P_k) - \hat{h}_{x^k}(P^\star) &= \frac{f(x^k - P_k \nabla f(x^k)) - f(x^\star) + f(x^\star) - f(x^k - P^\star \nabla f(x^k))}{\|\nabla f(x^k)\|^2} \\
&\geq \tfrac{\mu}{4}\|(P^\star - P_k)\tfrac{\nabla f(x^k)}{\|\nabla f(x^k)\|}\|^2 - \tfrac{\mu}{2}\tfrac{\|x^k - P^\star \nabla f(x^k) - x^\star\|^2}{\|\nabla f(x^k)\|^2} + \frac{f(x^\star) - f(x^k - P^\star \nabla f(x^k))}{\|\nabla f(x^k)\|^2} \\
&= \tfrac{\mu}{4}\|(P^\star - P_k)\tfrac{\nabla f(x^k)}{\|\nabla f(x^k)\|}\|^2 - \tfrac{\mu}{2}\tfrac{\|x^k - P^\star \nabla f(x^k) - x^\star\|^2}{\|\nabla f(x^k)\|^2} - \hat{h}_{x^k}(P^\star) \quad (33) \\
&\geq \tfrac{\mu}{4}\|(P^\star - P_k)\tfrac{\nabla f(x^k)}{\|\nabla f(x^k)\|}\|^2 - \tfrac{H^2}{8\mu}\tfrac{\|x^k - x^\star\|^4}{\|\nabla f(x^k)\|^2} - \hat{h}_{x^k}(P^\star) \quad (34) \\
&\geq \tfrac{\mu}{4}\|(P^\star - P_k)\tfrac{\nabla f(x^k)}{\|\nabla f(x^k)\|}\|^2 - \tfrac{H^2 \kappa}{8\mu^3}\|x^k - x^\star\|^2 - \hat{h}_{x^k}(P^\star), \quad (35)
\end{aligned}
$$

where (33) uses the definition of $\hat{h}_{x^k}$; (34) applies the relation $\|x - P^\star \nabla f(x) - x^\star\| \leq \tfrac{H}{2\mu}\|x - x^\star\|^2$ from (27); (35) again uses the fact $\|\nabla f(x)\|^2 \geq \mu^2 \|x - x^\star\|^2$. Putting the relations back into (31) and assuming $\eta \leq \tfrac{1}{2L}$,

$$
\begin{aligned}
&\|P_{k+1} - P^\star\|_F^2 \\
&\leq \|P_k - P^\star\|_F^2 - \tfrac{\mu(\eta - L\eta^2)}{2}\|(P_k - P^\star)\tfrac{\nabla f(x^k)}{\|\nabla f(x^k)\|}\|^2 \\
&\quad + \tfrac{H^2 \kappa(\eta - L\eta^2)}{4\mu^3}\|x^k - x^\star\|^2 + 2(\eta - L\eta^2)\hat{h}_{x^k}(P^\star) + 2L\eta^2 \hat{h}_{x^k}(P^\star) \\
&= \|P_k - P^\star\|_F^2 - \tfrac{\mu(\eta - L\eta^2)}{2}\|(P_k - P^\star)\tfrac{\nabla f(x^k)}{\|\nabla f(x^k)\|}\|^2 + \tfrac{H^2 \kappa(\eta - L\eta^2)}{4\mu^3}\|x^k - x^\star\|^2 + 2\eta \hat{h}_{x^k}(P^\star) \\
&\leq \|P_k - P^\star\|_F^2 - \tfrac{\mu(\eta - L\eta^2)}{2}\|(P_k - P^\star)\tfrac{\nabla f(x^k)}{\|\nabla f(x^k)\|}\|^2 + \tfrac{H^2 \kappa(\eta - L\eta^2)}{4\mu^3}\|x^k - x^\star\|^2 + 2\eta\tfrac{H^2 \kappa}{8\mu^3}\|x^k - x^\star\|^2 \quad (36) \\
&= \|P_k - P^\star\|_F^2 - \tfrac{\mu(\eta - L\eta^2)}{2}\|(P_k - P^\star)\tfrac{\nabla f(x^k)}{\|\nabla f(x^k)\|}\|^2 + (2\eta - L\eta^2)\tfrac{H^2 \kappa}{4\mu^3}\|x^k - x^\star\|^2,
\end{aligned}
$$

where (36) uses the relation (28) and this completes the proof.

### C.5. Proof of Theorem 3.4

The proof of Theorem 3.4 relies on the following auxiliary results.

**Lemma C.1.** *Under* **A1** *to* **A4**, $h_x(P) - \inf_{Q \in \mathbb{R}^{n \times n}} h_x(Q) \leq \frac{1}{2\mu}(LD+1)^2$.

*Proof.* Note that $h_x(P) = \frac{f(x - P\nabla f(x)) - f(x)}{\|\nabla f(x)\|^2} \geq \frac{f(x^\star) - f(x)}{\|\nabla f(x)\|^2}$ for all $P \in \mathcal{P}$, we deduce that

$$h_x(P) - \inf_{Q \in \mathbb{R}^{n \times n}} h_x(Q) \leq \frac{f(x - P\nabla f(x)) - f(x)}{\|\nabla f(x)\|^2} - \frac{f(x^\star) - f(x)}{\|\nabla f(x)\|^2} \tag{37}$$

$$= \frac{f(x - P\nabla f(x)) - f(x^\star)}{\|\nabla f(x)\|^2}$$

$$\leq \frac{1}{2\mu} \frac{\|\nabla f(x - P\nabla f(x))\|^2}{\|\nabla f(x)\|^2} \tag{38}$$

$$\leq \frac{1}{2\mu} \frac{[\|\nabla f(x)\| + \|P\| \cdot \|\nabla f(x)\|]^2}{\|\nabla f(x)\|^2} \tag{39}$$

$$\leq \frac{1}{2\mu}(LD+1)^2, \tag{40}$$

where (37) applies $h_x(P) \geq \frac{f(x^\star) - f(x)}{\|\nabla f(x)\|^2}$; (38) uses $f(x) - f(x^\star) \leq \frac{1}{2\mu}\|\nabla f(x)\|^2$; (39) uses $L$-smoothness and (40) uses $\|P\| \leq D$. $\qquad\square$

Then we show that HDM converges even when $\eta$ is a constant that does not depend on $K$.

**Lemma C.2.** *Under* **A1** *to* **A4**, *Algorithm 1 with* $\eta_k \equiv \eta \in (0, \frac{1}{2L(LD+1)^2\kappa}]$ *satisfies*

- $\lim_{k \to \infty} \|x^k - x^\star\| = 0$.

- $\lim_{K \to \infty} \sum_{k=1}^{K} \|x^k - x^\star\|^2 < \infty$.

*Proof.* Using the online gradient descent update, we have

$$\|P_{k+1} - P\|_F^2 \leq \|P_k - \eta \nabla h_{x^k}(P_k) - P\|_F^2$$

$$= \|P_k - P\|_F^2 - 2\eta \langle \nabla h_{x^k}(P_k), P_k - P \rangle + \eta^2 \|\nabla h_{x^k}(P_k)\|_F^2$$

$$\leq \|P_k - P\|_F^2 - 2\eta[h_{x^k}(P_k) - h_{x^k}(P)] + 2L\eta^2[h_{x^k}(P_k) - \inf_{P \in \mathbb{R}^{n \times n}} h_{x^k}(P)] \tag{41}$$

$$= \|P_k - P\|_F^2 - 2\eta h_{x^k}(P_k) + 2\eta h_{x^k}(P) + 2L\eta^2[h_{x^k}(P_k) - \inf_{P \in \mathbb{R}^{n \times n}} h_{x^k}(P)],$$

where (41) follows from convexity $h_{x^k}(P) \geq h_{x^k}(P_k) + \langle \nabla h_{x^k}(P_k), P - P_k \rangle$ and $L$-smoothness of $h_x(P)$. Next, we invoke the upperbound on $h_{x^k}(P_k) - \inf_{Q \in \mathbb{R}^{n \times n}} h_{x^k}(Q)$ from Lemma C.1:

$$2L\eta^2[h_{x^k}(P_k) - \inf_{P \in \mathbb{R}^{n \times n}} h_{x^k}(P)] \leq \frac{2L}{2\mu}(LD+1)^2\eta^2 = \kappa(LD+1)^2\eta^2.$$

and deduce that

$$2\eta h_{x^k}(P_k) \leq 2\eta h_{x^k}(P) + \|P_k - P\|_F^2 - \|P_{k+1} - P\|_F^2 + 2L\eta^2[h_{x^k}(P_k) - \inf_{P \in \mathbb{R}^{n \times n}} h_{x^k}(P)]$$

$$\leq 2\eta h_{x^k}(P) + \|P_k - P\|_F^2 - \|P_{k+1} - P\|_F^2 + \eta^2 \kappa(LD+1)^2.$$

Next, we divide both sides of the inequality by $2\eta$ and

$$h_{x^k}(P_k) \leq h_{x^k}(P) + \frac{\|P_k - P\|_F^2 - \|P_{k+1} - P\|_F^2}{2\eta} + \frac{\eta \kappa(LD+1)^2}{2}.$$

Telescoping the relation and using $\mathrm{diam}(\mathcal{P}) \leq D$, we get

$$\sum_{k=1}^{K} h_{x^k}(P_k) \leq \sum_{k=1}^{K} h_{x^k}(P) + \frac{D^2}{2\eta} + \frac{\eta \kappa(LD+1)^2}{2}K$$

Taking $P = (1/L)I$ and taking average, $\sum_{k=1}^{K} h_{x^k}(P) \le -\frac{1}{2L}K$ and

$$\frac{1}{K}\sum_{k=1}^{K} h_{x^k}(P_k) \le -\frac{1}{2L} + \frac{D^2}{2\eta K} + \frac{\eta\kappa(LD+1)^2}{2} = -\frac{1}{4L} + \frac{D^2}{2\eta K} + \frac{\eta\kappa(LD+1)^2}{2} - \frac{1}{4L}$$

With $\eta \le \frac{1}{2L(LD+1)^2\kappa}$, we have $\frac{\eta\kappa(LD+1)^2}{2} - \frac{1}{4L} \le 0$ and

$$\frac{1}{K}\sum_{k=1}^{K} h_{x^k}(P_k) \le -\frac{1}{4L} + \frac{D^2 L(LD+1)^2\kappa}{K}.$$

Using the reduction Lemma 2.4, we get, for any $k \ge 1$ (since $\eta$ does not depend on the iteration number),

$$f(x^{k+1}) - f(x^\star) \le [f(x^1) - f(x^\star)](1 - 2\mu\max\{\tfrac{1}{4L} - \tfrac{D^2 L(LD+1)^2\kappa}{k}, 0\})^k$$

and there exists some $K_0$ such that for all $k \ge K_0$, that $[f(x^k) - f(x^\star)](1 - \frac{1}{4\kappa})^k \le [f(x^1) - f(x^\star)]$ since

$$\lim_{k\to\infty} 1 - 2\mu\max\{\tfrac{1}{4L} - \tfrac{2D^2 L(LD+1)^2\kappa}{k}, 0\} = 1 - \tfrac{1}{2\kappa} < 1 - \tfrac{1}{4\kappa}.$$

This proves the first relation $\lim_{k\to\infty}\|x^k - x^\star\| = 0$ since $\|x - x^\star\|^2 \le \frac{2}{\mu}[f(x) - f(x^\star)]$ and the second relation follows directly from

$$\sum_{k=1}^{\infty}\|x^k - x^\star\|^2 = \sum_{k=1}^{K_0}\|x^k - x^\star\|^2 + \sum_{k=K_0+1}^{\infty}\|x^k - x^\star\|^2 \tag{42}$$

$$\le \sum_{k=1}^{K_0}\|x^k - x^\star\|^2 + \sum_{k=K_0+1}^{\infty}\frac{2}{\mu}[f(x^1) - f(x^\star)](1 - \tfrac{1}{4\kappa})^{-k} < \infty. \tag{43}$$

$\square$

Now we are ready to prove Theorem 3.4, and we start by stating the precise definition of a uniformly independent sequence.

**Definition C.1** (Uniformly linearly indepdendent sequence (Conn et al., 1991)). *A sequence of unit-norm vectors* $\{g^k\}, g^k \in \mathbb{R}^n, \|g^k\| = 1$ *is uniformly linearly independent if there exists a constant* $c > 0, K_0 \ge 0$ *and* $m \ge n$ *such that for each* $k \ge K_0$, *one can choose* $n$ *distinct indices*

$$k \le k_1 < \cdots < k_n \le k + m$$

*with* $\sigma_{\min}([g^{k_1}, \ldots, g^{k_n}]) \ge c$.

We prove by contradiction. For brevity we denote $g^k := \frac{\nabla f(x^k)}{\|\nabla f(x^k)\|}$ and $e_k := \|P_k - P^\star\|_F^2$. Recall that $P^\star = [\nabla^2 f(x^\star)]^{-1}$. First, using Lemma C.2, for any $\varepsilon > 0$, there exists some index $K_1$ such that for all $k \ge K_1$ we have $\|x^k - x^\star\|^2 \le \varepsilon$ and that $\sum_{k=1}^{\infty}\|x^k - x^\star\|^2$ is bounded. Then we show that $\lim_{k\to\infty}\|\nabla h_{x^k}(P_k)\|_F = 0$ using (31): after re-arrangement, for any $K \ge 1$,

$$\sum_{k=1}^{K}\hat{h}_{x^k}(P_k) \le \frac{2\eta}{2\eta(1-\eta L)}\sum_{k=1}^{K}\hat{h}_{x^k}(P^\star) + \frac{1}{2\eta(1-\eta L)}\|P_1 - P^\star\|_F^2$$

$$\le \frac{2\eta}{2\eta(1-\eta L)}\frac{H^2\kappa}{8\mu^3}\sum_{k=1}^{K}\|x^k - x^\star\|^2 + \frac{1}{2\eta(1-\eta L)}\|P_1 - P^\star\|_F^2. \tag{44}$$

$$\le \frac{2\eta}{2\eta(1-\eta L)}\frac{H^2\kappa}{8\mu^3}\sum_{k=1}^{\infty}\|x^k - x^\star\|^2 + \frac{1}{2\eta(1-\eta L)}\|P_1 - P^\star\|_F^2,$$

where (44) applies (28). Since $\sum_{k=1}^{\infty}\|x^k - x^\star\|^2$ is bounded and $\hat{h}_x(P)$ is nonnegative, we must have $\lim_{k\to\infty}\hat{h}_{x^k}(P_k) = 0$. Further notice that $\|\nabla\hat{h}_{x^k}(P_k)\|_F^2 \le 2L\hat{h}_{x^k}(P_k)$, it implies $\lim_{k\to\infty}\sum_{k=1}^{K}\|\nabla h_{x^k}(P_k)\|_F^2 < \infty$, giving $\lim_{k\to\infty}\|\nabla h_{x^k}(P_k)\|_F = 0$ and $\lim_{k\to\infty} P_k = \bar{P}$ also exists. Now suppose by contradiction that $\|\bar{P} - P^\star\|_F = \theta > 0$. Then there exists some $K_2 > 0$ such that for all $k \ge K_2$, $\|P_k - \bar{P}\|_F \le \varepsilon$. For $k \ge \max\{K_0, K_1, K_2\} + 1$, we invoke Lemma 3.1 with $\eta \in (0, \frac{1}{2L}]$ to get

$$\|P_{k+1} - P^\star\|_F^2 \le \|P_k - P^\star\|_F^2 - \alpha_1\|(P_k - P^\star)g^k\|^2 + \alpha_2\varepsilon$$

$$= \|P_k - P^\star\|_F^2 - \alpha_1\|(P_k - \bar{P} + \bar{P} - P^\star)g^k\|^2 + \alpha_2\varepsilon$$

$$\le \|P_k - P^\star\|_F^2 - \frac{\alpha_1}{2}\|(\bar{P} - P^\star)g^k\|^2 + 3\alpha_1\|(P_k - \bar{P})g^k\|^2 + \alpha_2\varepsilon$$

$$\le \|P_k - P^\star\|_F^2 - \frac{\alpha_1}{2}\|(\bar{P} - P^\star)g^k\|^2 + 3\alpha_1\varepsilon^2 + \alpha_2\varepsilon \tag{45}$$

$$= \|P_k - P^\star\|_F^2 - \frac{\alpha_1}{2}\operatorname{tr}(g^k(g^k)^\top, (\bar{P} - P^\star)^\top(\bar{P} - P^\star)) + 3\alpha_1\varepsilon^2 + \alpha_2\varepsilon, \tag{46}$$

where $\alpha_1 = \frac{\mu(\eta - L\eta^2)}{2} > 0$, $\alpha_2 = \frac{1}{4}(2\eta - L\eta^2)H^2\kappa\mu^{-3}$, and (45) uses the fact that $\|P_k - P^\star\|_F \leq \varepsilon$. Telescoping (46) for the next $m + 1$ iterations, we deduce that

$$
\begin{aligned}
e_{k+m+1} &= \|P_{k+m+1} - P^\star\|_F^2 \\
&\leq \|P_k - P^\star\|_F^2 - \tfrac{\alpha_1}{2}\sum_{j=0}^{m} \mathrm{tr}(g^{k+j}(g^{k+j})^\top, (\bar{P} - P^\star)^\top(\bar{P} - P^\star)) + (3\alpha_1\varepsilon^2 + \alpha_2\varepsilon)(m+1) \\
&= e_k - \tfrac{\alpha_1}{2}\mathrm{tr}(\textstyle\sum_{j=0}^{m} g^{k+j}(g^{k+j})^\top, (\bar{P} - P^\star)^\top(\bar{P} - P^\star)) + (3\alpha_1\varepsilon^2 + \alpha_2\varepsilon)(m+1)
\end{aligned}
$$

and using the independent sequence assumption, we can pick $k_1, \ldots, k_n$ such that

$$
\sigma_{\min}([g^{k_1}, \ldots, g^{k_n}]) \geq c
$$

and $\sum_{j=0}^{m} g^{k+j}(g^{k+j})^\top \succeq \sum_{i=1}^{n} g^{k_i}(g^{k_i})^\top \succeq c^2 I$. Hence

$$
\mathrm{tr}(\textstyle\sum_{j=0}^{m} g^{k+j}(g^{k+j})^\top, (\bar{P} - P^\star)^\top(\bar{P} - P^\star)) \geq c^2\,\mathrm{tr}((\bar{P} - P^\star)^\top(\bar{P} - P^\star)) = c^2\|\bar{P} - P^\star\|_F^2 = c^2\theta^2
$$

and $e_{k+m+1} \leq e_k - \frac{\alpha_1 c^2\theta^2}{2} + (3\alpha_1\varepsilon^2 + \alpha_2\varepsilon)(m+1)$. Since $\varepsilon$ is arbitrary, we can repeat the same argument till $e_{k+m+1} < 0$, which leads to contradiction unless $\theta = 0$. This completes the proof.

# D. Proof of Results in Section 4

## D.1. `HDM` + Heavy-ball Momentum (`HDM-HB`)

Algorithm 2 uses the following *heavy-ball feedback function* to guide the online learning for $(P_k, B_k)$:

$$
h_{x,x^-}(P, B) := \frac{\psi(x^+, x) - \psi(x, x^-)}{\|\nabla f(x)\|^2 + \frac{\tau}{2}\|x - x^-\|^2} = \frac{[f(x^+) + \frac{\omega}{2}\|x^+ - x\|^2] - [f(x) + \frac{\omega}{2}\|x - x^-\|^2]}{\|\nabla f(x)\|^2 + \frac{\tau}{2}\|x - x^-\|^2},
$$

where $\omega > 0$, $\tau > 0$, $x^+ = x - P\nabla f(x) + B(x - x^-)$, and $\psi(x, x^-) = f(x) + \frac{\omega}{2}\|x - x^-\|^2$. To show that online learning can be applied to $h_{x,x^-}(P, B)$ with regret guarantees, we need to verify the convexity and Lipschitz continuity of $h_{x,x^-}(P, B)$ with respect to the norm defined by

$$
\|(P, B)\| := \sqrt{\|P\|_F^2 + \|B\|_F^2}. \tag{47}
$$

**Lemma D.1.** *Under* **A1**, **A3**, *and* **A5**, *the heavy-ball feedback function $h_{x,x^-}(P, B)$ is jointly convex in $(P, B)$ and $c$-Lipschitz with respect to the norm defined in (47), where* $c := \sqrt{2}(1 + \frac{2}{\tau})[1 + 2(1 + \frac{2}{\tau})D(L + \omega)]$.

*Proof.* Denote $x^+(P, B) := x - P\nabla f(x) + B(x - x^-)$. Recall that the feedback function is

$$
h_{x,x^-}(P, B) = \frac{[f(x^+(P, \beta)) + \frac{\omega}{2}\|x^+(P, \beta) - x\|^2] - [f(x) + \frac{\omega}{2}\|x - x^-\|^2]}{\|\nabla f(x)\|^2 + \frac{\tau}{2}\|x - x^-\|^2}.
$$

Since $x^+(P, B)$ is affine in $(P, B)$ and $f$ is convex, the term $f(x^+(P, \beta)) + \frac{\omega}{2}\|x^+(P, \beta) - x\|^2$ is jointly convex as a function of $(P, B)$. The other terms in the feedback function $h_{x,x^-}(P, B)$ are constants, so $h_{x,x^-}(P, B)$ is also jointly convex in $(P, B)$.

To prove the Lipschitz continuity of $h_{x,x^-}(P, B)$, it suffices to show that the gradients of $h_{x,x^-}(P, B)$ are bounded. The gradients of $h_{x,x^-}(P, B)$ with respect to $P$ and $B$ are

$$
\nabla_P h_{x,x^-}(P, B) = \frac{[-\nabla f(x^+(P,B)) + \omega P\nabla f(x) - \omega B(x - x^-)]\nabla f(x)^\top}{\|\nabla f(x)\|^2 + \frac{\tau}{2}\|x - x^-\|^2},
$$

$$
\nabla_B h_{x,x^-}(P, B) = \frac{[\nabla f(x^+(P,B)) - \omega P\nabla f(x) + \omega B(x - x^-)](x - x^-)^\top}{\|\nabla f(x)\|^2 + \frac{\tau}{2}\|x - x^-\|^2}.
$$

Using the fact $\|ab^\top\|_F = \|a\| \cdot \|b\|$, the gradients have norms

$$
\|\nabla_P h_{x,x^-}(P, B)\|_F = \frac{\|\nabla f(x^+(P,B)) - \omega P\nabla f(x) + \omega B(x - x^-)\|\|\nabla f(x)\|}{\|\nabla f(x)\|^2 + \frac{\tau}{2}\|x - x^-\|^2}, \tag{48}
$$

$$
\|\nabla_B h_{x,x^-}(P, B)\|_F = \frac{\|\nabla f(x^+(P,B)) - \omega P\nabla f(x) + \omega B(x - x^-)\|\|x - x^-\|}{\|\nabla f(x)\|^2 + \frac{\tau}{2}\|x - x^-\|^2}. \tag{49}
$$

Using **A1**, we have the Lipschitz continuity of $\nabla f(x)$ and thus

$$\|\nabla f(x^+(P,B)) - \omega P \nabla f(x) + \omega B(x - x^-)\|$$
$$\leq \|\nabla f(x^+(P,B)) - \nabla f(x)\| + \|(I - \omega P)\nabla f(x)\| + \omega\|B\|\|x - x^-\|$$
$$\leq L\|P\nabla f(x) - B(x - x^-)\| + (1 + \omega\|P\|)\|\nabla f(x)\| + \omega\|B\|\|x - x^-\|$$
$$\leq LD(\|\nabla f(x)\| + \|x - x^-\|) + (1 + \omega D)\|\nabla f(x)\| + \omega D\|x - x^-\|$$
$$= (1 + LD + \omega D)\|\nabla f(x)\| + (\omega + L)D\|x - x^-\|. \tag{50}$$

Now, we bound the norms in (48)–(49) by the case analysis.

**Case 1.** If $\frac{\tau}{2}\|x - x^-\|^2 \leq \|\nabla f(x)\|^2$, then together with (50), we have

$$\max\{\|\nabla_P h_{x,x^-}(P,B)\|_F, \|\nabla_B h_{x,x^-}(P,B)\|_F\} \leq \frac{[(1+LD+\omega D)\|\nabla f(x)\|+(\omega+L)D\|x-x^-\|]\max\{\sqrt{2\tau^{-1}},1\}\|\nabla f(x)\|}{\|\nabla f(x)\|^2}$$
$$\leq \max\{\sqrt{2\tau^{-1}},1\}[(1 + LD + \omega D) + \frac{\sqrt{2}D(\omega+L)}{\sqrt{\tau}}]$$
$$= \max\{\sqrt{2\tau^{-1}},1\}(1 + D(L + \omega)(1 + \sqrt{2\tau^{-1}})).$$

**Case 2.** If $\frac{\tau}{2}\|x - x^-\|^2 \geq \|\nabla f(x)\|^2$, then

$$\max\{\|\nabla_P h_{x,x^-}(P,B)\|_F, \|\nabla_B h_{x,x^-}(P,B)\|_F\} \leq \frac{[(1+LD+\omega D)\|\nabla f(x)\|+(\omega+L)D\|x-x^-\|]\max\{\sqrt{\frac{\tau}{2}},1\}\|x-x^-\|}{\frac{\tau}{2}\|x-x^-\|^2}$$
$$\leq \max\{\sqrt{\tau/2},1\}[\frac{\sqrt{2}(1+LD+\omega D)}{\sqrt{\tau}} + \frac{2D(\omega+L)}{\tau}]$$
$$= \sqrt{2\tau^{-1}}\max\{\sqrt{2\tau^{-1}},1\}(1 + D(L + \omega)(1 + \sqrt{2\tau^{-1}})).$$

Combining the two cases, we have

$$\max\{\|\nabla_P h_{x,x^-}(P,B)\|_F, \|\nabla_B h_{x,x^-}(P,B)\|_F\} \leq \max\{\frac{2}{\tau},1\}(1 + D(L + \omega)(1 + \sqrt{2\tau^{-1}}))$$
$$\leq (1 + \frac{2}{\tau})[1 + 2(1 + \frac{2}{\tau})D(L + \omega)].$$

Then the gradient of $h_{x,x^-}(P,B)$ under the norm defined in (47) is bounded by the constant $c := \sqrt{2}(1 + \frac{2}{\tau})[1 + 2(1 + \frac{2}{\tau})D(L + \omega)]$. $\qquad\square$

The next lemma bounds the potential at the last iterate $x^{K+1}$ from Algorithm 2 in terms of the sum of feedback functions $h_{x^k,x^{k-1}}(P_k,B_k)$.

**Lemma D.2.** *The sequence $\{x^k\}$ generated from Algorithm 2 satisfies*

$$f(x^{K+1}) - f(x^\star) + \frac{\omega}{2}\|x^{K+1} - x^K\|^2 \leq \frac{f(x^1)-f(x^\star)}{1+\sum_{k=1}^K \max\{-h_{x^k,x^{k-1}}(P_k,B_k),0\}V}, \tag{51}$$

*where $V := \min\left\{\frac{f(x^1)-f(x^\star)}{4\Delta^2}, \frac{\tau}{4\omega}\right\}$ and $\Delta := \max_{x \in \mathcal{L}_{f(x^1)}} \min_{x^\star \in \mathcal{X}^\star} \|x - x^\star\|$.*

*Proof.* The null step guarantees

$$\frac{\psi(x^{k+1},x^k)-\psi(x^k,x^{k-1})}{\|\nabla f(x^k)\|^2+\frac{\tau}{2}\|x^k-x^{k-1}\|^2} = \min\{h_{x^k,x^{k-1}}(P_k,B_k),0\}.$$

Using the initial condition $x^1 = x^0$, we have

$$\psi(x^{K+1}, x^K) - f(x^\star) = \frac{1}{\frac{1}{\psi(x^{K+1}, x^K) - f(x^\star)}}$$

$$= \frac{1}{\sum_{k=1}^K \frac{1}{\psi(x^{k+1}, x^k) - f(x^\star)} - \frac{1}{\psi(x^k, x^{k-1}) - f(x^\star)} + \frac{1}{\psi(x^1, x^0) - f(x^\star)}}$$

$$= \frac{1}{\sum_{k=1}^K \frac{\psi(x^k, x^{k-1}) - \psi(x^{k+1}, x^k)}{[\psi(x^{k+1}, x^k) - f(x^\star)][\psi(x^k, x^{k-1}) - f(x^\star)]} + \frac{1}{\psi(x^1, x^0) - f(x^\star)}}$$

$$= \frac{1}{\sum_{k=1}^K \frac{\max\{-h_{x^k, x^{k-1}}(P_k, B_k), 0\}[\|\nabla f(x^k)\|^2 + \frac{\tau}{2}\|x^k - x^{k-1}\|^2]}{[\psi(x^{k+1}, x^k) - f(x^\star)][\psi(x^k, x^{k-1}) - f(x^\star)]} + \frac{1}{f(x^1) - f(x^\star)}}. \quad (52)$$

Then, by monotonicity, $\frac{\|\nabla f(x^k)\|^2 + \frac{\tau}{2}\|x^k - x^{k-1}\|^2}{[\psi(x^{k+1}, x^k) - f(x^\star)][\psi(x^k, x^{k-1}) - f(x^\star)]} \geq \frac{\|\nabla f(x^k)\|^2 + \frac{\tau}{2}\|x^k - x^{k-1}\|^2}{[\psi(x^k, x^{k-1}) - f(x^\star)]^2}$.

Now we do case analysis to bound

$$\frac{\|\nabla f(x^k)\|^2 + \frac{\tau}{2}\|x^k - x^{k-1}\|^2}{[\psi(x^k, x^{k-1}) - f(x^\star)]^2} = \frac{\|\nabla f(x^k)\|^2 + \frac{\tau}{2}\|x^k - x^{k-1}\|^2}{[f(x^k) + \frac{\omega}{2}\|x^k - x^{k-1}\|^2 - f(x^\star)]^2}$$

**Case 1.** If $\frac{\omega}{2}\|x^k - x^{k-1}\|^2 \leq f(x^k) - f(x^\star)$, then

$$\frac{\|\nabla f(x^k)\|^2 + \frac{\tau}{2}\|x^k - x^{k-1}\|^2}{[f(x^k) + \frac{\omega}{2}\|x^k - x^{k-1}\|^2 - f(x^\star)]^2} \geq \frac{\|\nabla f(x^k)\|^2}{4[f(x^k) - f(x^\star)]^2} \geq \frac{1}{4\Delta^2},$$

where $\Delta := \max_{x \in \mathcal{L}_{f(x^1)}} \min_{x^\star \in \mathcal{X}^\star} \|x - x^\star\|$.

**Case 2.** If $\frac{\omega}{2}\|x^k - x^{k-1}\|^2 \geq f(x^k) - f(x^\star)$, then $\frac{\tau}{2}\|x^k - x^{k-1}\|^2 \geq \frac{\tau}{\omega}[f(x^k) - f(x^\star)]$ and

$$\frac{\|\nabla f(x^k)\|^2 + \frac{\tau}{2}\|x^k - x^{k-1}\|^2}{[f(x^k) + \frac{\omega}{2}\|x^k - x^{k-1}\|^2 - f(x^\star)]^2} \geq \frac{\frac{\tau}{2}\|x^k - x^{k-1}\|^2}{\omega^2\|x^k - x^{k-1}\|^4} = \frac{\tau}{2\omega^2}\frac{1}{\|x^k - x^{k-1}\|^2} \geq \frac{\tau}{4\omega}\frac{1}{f(x^1) - f(x^\star)}.$$

since $\frac{\omega}{2}\|x^k - x^{k-1}\|^2 \leq \psi(x^k, x^{k-1}) - f(x^\star) \leq \psi(x^1, x^0) - f(x^\star) = f(x^1) - f(x^\star)$.

In both cases, we have $\frac{\|\nabla f(x^k)\|^2 + \frac{\tau}{2}\|x^k - x^{k-1}\|^2}{[\psi(x^k, x^{k-1}) - f(x^\star)]^2} \geq \min\{\frac{1}{4\Delta^2}, \frac{\tau}{4\omega}\frac{1}{f(x^1) - f(x^\star)}\} = \frac{V}{f(x^1) - f(x^\star)}$, where the constant $V$ is defined in the lemma. Finally, plugging in the definition of $\psi$, (52) gives

$$f(x^{K+1}) - f(x^\star) + \frac{\omega}{2}\|x^{K+1} - x^K\|^2 \leq \frac{f(x^1) - f(x^\star)}{1 + \sum_{k=1}^K \max\{-h_{x^k, x^{k-1}}(P_k, B_k), 0\}V}.$$

$\square$

The next lemma shows that there exist hindsight $\bar{P}, \bar{B}$ such that $h_{x, x^-}(\bar{P}, \bar{B}) \leq -\theta < 0$ for some $\theta$.

**Lemma D.3.** *Let $\omega = 3L$ and $\tau = 16L^2$. Then for any $x, x^- \notin \mathcal{X}^\star$, we have $h_{x, x^-}(\frac{1}{4L}I, \frac{1}{2}I) \leq -\frac{1}{8L}$. In particular, if $\frac{1}{4L}I \in \mathcal{P}$, $\frac{1}{2}I \in \mathcal{B}$, and $\{x^k\}_{k=1}^K \cap \mathcal{X}^\star = \varnothing$, then*

$$\gamma_K^\star := -\min_{(P, B) \in \mathcal{P} \times \mathcal{B}} \frac{1}{K}\sum_{k=1}^K h_{x^k, x^{k-1}}(P, B) \geq \frac{1}{8L}.$$

*Proof.* When $P = \alpha I$ and $B = \beta I$ for some $\alpha, \beta > 0$, the classical analysis for the heavy-ball momentum (Danilova et al., 2020) gives

$$f(x^+) + \frac{1 - \alpha L}{2\alpha}\|x^+ - x\|^2 \leq f(x) + \frac{\beta^2}{2\alpha}\|x - x^-\|^2 - \frac{\alpha}{2}\|\nabla f(x)\|^2.$$

Let $\alpha = \frac{1}{4L}$ and $\beta = \frac{1}{2}$, we have

$$f(x^+) + \frac{3L}{2}\|x^+ - x\|^2 \leq f(x) + \frac{L}{2}\|x - x^-\|^2 - \frac{1}{8L}\|\nabla f(x)\|^2$$

$$= f(x) + \frac{3L}{2}\|x - x^-\|^2 - \frac{1}{8L}\|\nabla f(x)\|^2 - L\|x - x^-\|^2$$

$$= f(x) + \frac{3L}{2}\|x - x^-\|^2 - \frac{1}{8L}[\|\nabla f(x)\|^2 + 8L^2\|x - x^-\|^2]$$

and re-arranging the terms, we get

$$\frac{f(x^+)+\frac{3L}{2}\|x^+-x\|^2-[f(x)+\frac{3L}{2}\|x-x^-\|^2]}{\|\nabla f(x)\|^2+8L^2\|x-x^-\|^2} \leq -\frac{1}{8L}$$

and this completes the proof. □

### D.1.1. PROOF OF THEOREM 4.1

By Lemma D.1, the heavy-ball feedback is convex and Lipschitz, and thus the same proof of Lemma 2.2 guarantees that online gradient descent

$$(P_{k+1}, B_{k+1}) = \Pi_{\mathcal{P}\times\mathcal{B}}[(P_k, B_k) - \eta\nabla h_{x^k,x^{k-1}}(P_k, B_k)]$$

(with $\eta_p = \eta_b = \eta$) gives the regret bound

$$\tfrac{1}{K}\sum_{k=1}^K -h_{x^k,x^{k-1}}(P_k, B_k) \geq \gamma_K^\star - \tfrac{\rho_K}{K}$$

for some sublinear regret $\rho_K = \mathcal{O}(\sqrt{K})$ and the constant $\gamma_K^\star$ as defined in Lemma D.3. Using the inequality

$$\tfrac{1}{K}\sum_{k=1}^K \max\{-h_{x^k,x^{k-1}}(P_k, B_k), 0\} \geq \max\big\{\tfrac{1}{K}\sum_{k=1}^K -h_{x^k,x^{k-1}}(P_k, B_k), 0\big\} \geq \max\{\gamma_K^\star - \tfrac{\rho_K}{K}, 0\},$$

the desired result follows directly from (51) in Lemma D.2.

### D.2. `HDM` + Nesterov Momentum (`HDM-AGD`)

#### D.2.1. `HDM` WITH NESTEROV MOMENTUM

In smooth convex optimization, accelerated gradient descent (`AGD`) achieves the convergence rate $\mathcal{O}(\frac{L}{K^2})$ (Nesterov, 1983):

$$\begin{aligned}
y^k &= x^k + (1 - \tfrac{A_k}{A_{k+1}})(z^k - x^k) \\
x^{k+1} &= y^k - \tfrac{1}{L}\nabla f(y^k) \\
z^{k+1} &= z^k + \tfrac{A_{k+1}-A_k}{L}\nabla f(y^k),
\end{aligned} \tag{53}$$

where $\{A_k\}$ is a pre-specified sequence. To apply `HDM`, we can replace the gradient descent step (53) by hypergradient descent: $x^{k+1} = y^k - P_k\nabla f(y^k)$ and $P_{k+1} = \Pi_{\mathcal{P}}[P_k - \eta\nabla h_{y^k}(P_k)]$. That is, we only accelerate the gradient descent step in `AGD`. Algorithm 4 provides a realization of the `HDM-AGD` based on a monotone variant of `AGD` (d'Aspremont et al., 2021). The convergence of `HDM-AGD` is established in Theorem D.1, the proof of which is deferred to Appendix D.2.3.

---

**Algorithm 4** HDM with Nesterov momentum

---

**input** starting point $x^1, z^1, \eta > 0, \theta \in [\frac{1}{2}, LD), A_0 = 0$
**for** $k = 1, 2,...$ **do**

$\quad A_{k+1} = (A_{k+1} - A_k)^2$

$\quad y^k = x^k + (1 - \tfrac{A_k}{A_{k+1}})(z^k - x^k)$

$\quad x^{k+1} = \underset{x\in\{y^k-\frac{1}{L}\nabla f(y^k),y^k-P_k\nabla f(y^k),x^k\}}{\arg\min} f(x)$

$\quad P_{k+1} = \Pi_{\mathcal{P}}[P_k - \eta\nabla h_{y^k}(P_k)]$

$\quad v_k = \max\{\tfrac{1}{2\max\{-h_{y^k}(P_k),1/(2L)\}}, \tfrac{L}{2\theta}\}$

$\quad z^{k+1} = z^k + \tfrac{(A_{k+1}-A_k)}{v_k}\nabla f(y^k)$

**end**
**output** $x^{K+1}$

---

**Theorem D.1.** *Assume* **A1** *and* **A3**. *Suppose* `AGD` *starts from* $(x', z')$ *and runs for $K$ iterations to output $\hat{x}$. Then Algorithm 4 starting from* $(x^1, z^1) = (\hat{x}, z')$ *and* $\theta \in [\frac{1}{2}, LD)$ *satisfies*

$$f(x^{K+1}) - f(x^\star) \leq \big[\tfrac{1}{2\theta} + (8 - \tfrac{4}{\theta})(\tfrac{LD-\omega_K^\star}{LD-\theta})\big]\tfrac{2L\|z'-x^\star\|^2}{K^2} + \mathcal{O}(\tfrac{\rho_K}{K^3}),$$

*where* $\omega_K^\star = -\min_{P\in\mathcal{P}}\frac{L}{K}\sum_{k=1}^K h_{y^k}(P)$ *depends on the iteration trajectory* $\{x^k\}_{k\leq K}$.

The parameter $\theta$ serves as a smooth interpolation between HDM and HDM-AGD: when $\theta = 1/2$, Theorem D.1 recovers the convergence rate of vanilla AGD; when $\theta > 1/2$ and $\omega_K^\star \to LD$, we expect HDM-AGD to yield faster convergence. As suggested by Figure 3b, HDM-AGD achieves faster convergence than AGD.

*Remark* 2. To mitigate the effect of regret, Algorithm 4 needs a warm start from vanilla AGD. However, experiments suggest that it is unnecessary in practice, and we leave an improved analysis to future work.

*Remark* 3. For strongly convex problems, we can combine Theorem D.1 with a standard restart argument (d'Aspremont et al., 2021; Roulet and d'Aspremont, 2017) and achieve a similar trajectory-based linear convergence rate.

### D.2.2. AUXILIARY RESULTS

**Lemma D.4.** *Suppose a nonnegative sequence* $\{A_k\}$ *satisfies* $A_{k+1} = (A_{k+1} - A_k)^2$ *and* $A_0 = 0$, *then* $A_{k+1} - A_k \le k+1$ *for all* $k \ge 1$.

*Proof.* We prove by induction. The induction hypothesis is $A_{k+1} - A_k \le k + 1$.

**Base case**. For $k = 1$, $A_2 - A_1 = \frac{\sqrt{5}+1}{2} < 2$ and the relation holds.

**Inductive step**. Suppose $A_{k+1} - A_k \le k$. Using $A_{k+2} = A_{k+1} + \frac{1}{2}\left(1 + \sqrt{4A_{k+1} + 1}\right)$, we deduce that

$$
\begin{aligned}
A_{k+2} - A_{k+1} &= \tfrac{1}{2}(1 + \sqrt{4A_{k+1} + 1}) \\
&= \tfrac{1}{2}(1 + \sqrt{4(A_{k+1} - A_k)^2 + 1}) \\
&\le \tfrac{1}{2}(1 + 2(A_{k+1} - A_k) + 1) \\
&\le 1 + A_{k+1} - A_k \\
&\le k + 2,
\end{aligned}
$$

(54)

(55)

where (54) uses $\sqrt{a + b} \le \sqrt{a} + \sqrt{b}$ and (55) uses the induction hypothesis $A_{k+1} - A_k \le k + 1$. By the principle of mathematical induction, this completes the proof. $\qquad\square$

**Lemma D.5** ((d'Aspremont et al., 2021)). *Under the same conditions as Lemma D.4,* $A_k \ge \frac{k^2}{4}$ *for all* $k \ge 1$.

### D.2.3. PROOF OF THEOREM D.1

Using the definition $h_y(P) = \frac{f(y - P\nabla f(y)) - f(y)}{\|\nabla f(y)\|^2}$ and the $x$-update in Algorithm 4:

$$
x^{k+1} = \operatorname*{arg\,min}_{x \in \{y^k - \frac{1}{L}\nabla f(y^k), y^k - P_k \nabla f(y^k), x^k\}} f(x),
$$

we have the following two inequalities:

$$
f(x^{k+1}) - f(y^k) \le \min\{h_{y^k}(P_k), -\tfrac{1}{2L}\}\|\nabla f(y^k)\|^2 \tag{56}
$$
$$
f(x^{k+1}) \le f(x^k). \tag{57}
$$

In other words, with $v_k = \max\{-\frac{1}{2\min\{h_{y^k}(P_k), -1/(2L)\}}, \frac{L}{2\theta}\}$, we have

$$
f(y^k) - \tfrac{1}{2v_k}\|\nabla f(y^k)\|^2 \ge f(x^{k+1}) \tag{58}
$$

and using $z^{k+1} = z^k + \frac{(A_{k+1} - A_k)}{v_k}\nabla f(y^k)$, by algebraic rearrangement

$$
\tfrac{v_k}{2}\|z^{k+1} - x^\star\|^2 \tag{59}
$$
$$
= \tfrac{v_k}{2}\|z^k - x^\star + \tfrac{(A_{k+1} - A_k)}{v_k}\nabla f(y^k)\|^2
$$
$$
= \tfrac{v_k}{2}\|z^k - x^\star\|^2 - (A_{k+1} - A_k)\langle \nabla f(y^k), z^k - x^\star \rangle + \tfrac{1}{2v_k}(A_{k+1} - A_k)^2\|\nabla f(y^k)\|^2. \tag{60}
$$

Next, we apply convexity and have

$$f(x^\star) \geq f(y^k) + \langle \nabla f(y^k), x^\star - y^k \rangle \tag{61}$$

$$f(x^k) \geq f(y^k) + \langle \nabla f(y^k), x^k - y^k \rangle \tag{62}$$

Taking a weighted summation between (58), (61) and (62), we deduce that

$$
\begin{aligned}
0 \geq\ & (A_{k+1} - A_k)[f(y^k) + \langle \nabla f(y^k), x^\star - y^k \rangle - f(x^\star)] + A_k[f(y^k) + \langle \nabla f(y^k), x^k - y^k \rangle - f(x^k)] \\
& + A_{k+1}[f(x^{k+1}) - f(y^k) + \tfrac{1}{2v_k}\|\nabla f(y^k)\|^2] \\
=\ & (A_{k+1} - A_k)\langle \nabla f(y^k), x^\star - y^k \rangle - (A_{k+1} - A_k)f(x^\star) \\
& + A_k \langle \nabla f(y^k), x^k - y^k \rangle - A_k f(x^k) + A_{k+1} f(x^{k+1}) + \tfrac{A_{k+1}}{2v_k}\|\nabla f(y^k)\|^2 \tag{63} \\
=\ & A_{k+1}[f(x^{k+1}) - f(x^\star)] - A_k[f(x^k) - f(x^\star)] \\
& + (A_{k+1} - A_k)\langle \nabla f(y^k), x^\star - y^k \rangle + A_k \langle \nabla f(y^k), x^k - y^k \rangle + \tfrac{A_{k+1}}{2v_k}\|\nabla f(y^k)\|^2 \tag{64} \\
=\ & A_{k+1}[f(x^{k+1}) - f(x^\star)] - A_k[f(x^k) - f(x^\star)] \\
& A_{k+1}\langle \nabla f(y^k), x^\star - y^k \rangle + A_k \langle \nabla f(y^k), x^k - x^\star \rangle + \tfrac{A_{k+1}}{2v_k}\|\nabla f(y^k)\|^2, \\
=\ & A_{k+1}[f(x^{k+1}) - f(x^\star)] - A_k[f(x^k) - f(x^\star)] \\
& - (A_{k+1} - A_k)\langle \nabla f(y^k), z^k - x^\star \rangle + \tfrac{A_{k+1}}{2v_k}\|\nabla f(y^k)\|^2, \tag{65}
\end{aligned}
$$

where (63) to (64) simply re-arrange the terms and (65) uses the identity $y^k = x^k + (1 - \frac{A_k}{A_{k+1}})(z^k - x^k)$:

$$A_{k+1}\langle \nabla f(y^k), x^\star - y^k \rangle + A_k \langle \nabla f(y^k), x^k - x^\star \rangle = -(A_{k+1} - A_k)\langle \nabla f(y^k), z^k - x^\star \rangle.$$

Putting the relations together, we arrive at

$$A_{k+1}[f(x^{k+1}) - f(x^\star)] \leq A_k[f(x^k) - f(x^\star)] + (A_{k+1} - A_k)\langle \nabla f(y^k), z^k - x^\star \rangle - \tfrac{A_{k+1}}{2v_k}\|\nabla f(y^k)\|^2.$$

and adding (60) gives

$$
\begin{aligned}
& A_{k+1}[f(x^{k+1}) - f(x^\star)] + \tfrac{v_k}{2}\|z^{k+1} - x^\star\|^2 \\
& \leq A_k[f(x^k) - f(x^\star)] + \tfrac{v_k}{2}\|z^k - x^\star\|^2 + \tfrac{A_{k+1} - (A_{k+1} - A_k)^2}{2v_k}\|\nabla f(y^k)\|^2 \\
& = A_k[f(x^k) - f(x^\star)] + \tfrac{v_k}{2}\|z^k - x^\star\|^2, \tag{66}
\end{aligned}
$$

where (66) uses the relation $A_{k+1} - (A_{k+1} - A_k)^2 = 0$.

We are now ready to analyze the acceleration effect of online hypergradient. Recall that we can guarantee

$$\tfrac{1}{K}\sum_{k=1}^K h_{y^k}(P_k) \leq -\gamma_K^\star + \tfrac{\rho_K}{K}, \tag{67}$$

where $\gamma_K^\star := -\min_{P \in \mathcal{P}} \sum_{k=1}^K h_{y^k}(P)$ is expected to be larger than $1/(2L)$ to improve performance. Recall that $\gamma_K^\star := \frac{\omega_K^\star}{L}, \omega_K^\star \geq 0$ and note that $\omega_K^\star$ depends on the iteration trajectory. Moreover, we have, by convexity of $f(x)$,

$$\tfrac{f(x - P\nabla f(x)) - f(x)}{\|\nabla f(x)\|^2} \geq \tfrac{f(x) - \langle \nabla f(x), P\nabla f(x) \rangle - f(x)}{\|\nabla f(x)\|^2} = -\tfrac{\langle \nabla f(x), P\nabla f(x) \rangle}{\|\nabla f(x)\|^2} \geq -D$$

and $\gamma_K^\star \leq D$ implies $\omega_K^\star \leq LD$. Define $\mathcal{I} := \{k : h_{y^k}(P_k) \leq -\frac{\theta}{L}\}$ for $\theta \in [\frac{1}{2}, LD)$. Then, according to (67),

$$-\tfrac{\omega_K^\star}{L} + \tfrac{\rho_K}{K} \geq \tfrac{1}{K}\sum_{k=1}^K h_{y^k}(P_k) = \tfrac{1}{K}[\sum_{k \in \mathcal{I}} h_{y^k}(P_k) + \sum_{k \in \bar{\mathcal{I}}} h_{y^k}(P_k)] \geq \tfrac{1}{K}\sum_{k \in \mathcal{I}} h_{y^k}(P_k) - \tfrac{\theta}{L}\tfrac{K - |\mathcal{I}|}{K}.$$

Using $h_{y^k}(P_k) \geq -D$, we get

$$-\tfrac{D}{K}|\mathcal{I}| \leq \tfrac{1}{K}\sum_{k \in \mathcal{I}} h_{y^k}(P_k) \leq -\tfrac{\omega_K^\star}{L} + \tfrac{\theta}{L}\tfrac{K - |\mathcal{I}|}{K} + \tfrac{\rho_K}{K}.$$

Re-arranging the terms,

$$\left(-\tfrac{D}{K} + \tfrac{\theta}{KL}\right)|\mathcal{I}| \leq \tfrac{\theta - \omega_K^\star}{L} + \tfrac{\rho_K}{K}.$$

Using $D > \frac{\theta}{L}$, we get

$$|\mathcal{I}| \geq \frac{\frac{\theta - \omega_K^\star}{L} + \frac{\rho_K}{K}}{-\frac{D}{K} + \frac{\theta}{KL}} = \frac{(\theta - \omega_K^\star)K + L\rho_K}{\theta - LD} = \frac{(\omega_K^\star - \theta)K}{LD - \theta} - \frac{L}{LD - \theta}\rho_K.$$

We have, if $k \in \mathcal{I}$, that using the fact that (66) holds for $v_k = \max\{-\frac{1}{2\min\{h_{y^k}(P_k), -1/(2L)\}}, \frac{L}{2\theta}\} = \frac{L}{2\theta}$,

$$A_{k+1}[f(x^{k+1}) - f(x^\star)] + \tfrac{L}{4\theta}\|z^{k+1} - x^\star\|^2 \leq A_k[f(x^k) - f(x^\star)] + \tfrac{L}{4\theta}\|z^k - x^\star\|^2. \tag{68}$$

On the other hand, if $k \notin \mathcal{I}$, $v_k \leq L$ and

$$A_{k+1}[f(x^{k+1}) - f(x^\star)] + \tfrac{v_k}{2}\|z^{k+1} - x^\star\|^2 \leq A_k[f(x^k) - f(x^\star)] + \tfrac{v_k}{2}\|z^k - x^\star\|^2 \tag{69}$$

and $f(x^{k+1}) \leq f(x^k)$ implies

$$\begin{aligned}
A_{k+1}[f(x^{k+1}) - f(x^\star)] &\leq A_{k+1}[f(x^k) - f(x^\star)] \\
&\leq A_k[f(x^k) - f(x^\star)] + k[f(x^k) - f(x^\star)],
\end{aligned} \tag{70}$$

where (70) uses the condition that $A_{k+1} - A_k \leq k$ from Lemma D.4.

Taking a weighted summation of (69) and (70), combining (68),

$$\begin{aligned}
&A_{k+1}[f(x^{k+1}) - f(x^\star)] + \tfrac{L}{4\theta}\|z^{k+1} - x^\star\|^2 \\
&\leq A_k[f(x^k) - f(x^\star)] + \tfrac{L}{4\theta}\|z^k - x^\star\|^2 + (1 - \tfrac{L}{2\theta v_k})k[f(x^k) - f(x^\star)] \cdot \mathbb{I}\{k \in \bar{\mathcal{I}}\} \\
&\leq A_k[f(x^k) - f(x^\star)] + \tfrac{L}{4\theta}\|z^k - x^\star\|^2 + (1 - \tfrac{1}{2\theta})k[f(x^k) - f(x^\star)] \cdot \mathbb{I}\{k \in \bar{\mathcal{I}}\}.
\end{aligned}$$

Telescoping the relation from 1 to $K$,

$$A_{K+1}[f(x^{K+1}) - f(x^\star)] \leq \tfrac{L}{4\theta}\|z^1 - x^\star\|^2 + \sum_{k \in \bar{\mathcal{I}}}(1 - \tfrac{1}{2\theta})k[f(x^k) - f(x^\star)].$$

Using $A_{K+1} \geq \frac{K^2}{4}$ from Lemma D.5 and that $|\bar{\mathcal{I}}| = K - |\mathcal{I}| \leq K - \frac{(\omega_K^\star - \theta)K}{LD - \theta} + \frac{L\rho_K}{LD - \theta}$,

$$\begin{aligned}
f(x^{K+1}) - f(x^\star) &\leq \tfrac{L}{4\theta A_{K+1}}\|z^1 - x^\star\|^2 + \tfrac{1}{A_{K+1}}(1 - \tfrac{1}{2\theta})\sum_{k \in \bar{\mathcal{I}}}k[f(x^k) - f(x^\star)] \\
&\leq \tfrac{L}{\theta K^2}\|z^1 - x^\star\|^2 + \tfrac{4}{K^2}(1 - \tfrac{1}{2\theta})\sum_{k \in \bar{\mathcal{I}}}k[f(x^k) - f(x^\star)] \\
&\leq \tfrac{L}{\theta K^2}\|z^1 - x^\star\|^2 + \tfrac{4}{K}(1 - \tfrac{1}{2\theta})[f(x^1) - f(x^\star)] \cdot |\bar{\mathcal{I}}| \\
&\leq \tfrac{L}{\theta K^2}\|z^1 - x^\star\|^2 + 4(1 - \tfrac{1}{2\theta})[f(x^1) - f(x^\star)](1 - \tfrac{\omega_K^\star - \theta}{LD - \theta} + \tfrac{L}{LD - \theta}\tfrac{\rho_K}{K}).
\end{aligned}$$

Suppose we run accelerated gradient descent from $z'$ for $K$ iterations and obtain $x^1$.

Plugging in $f(x^1) - f(x^\star) \leq \frac{2L}{K^2}\|z' - x^\star\|^2$ we get, using $z^1 = z'$, that

$$\begin{aligned}
f(x^{K+1}) - f(x^\star) &\leq \tfrac{L}{\theta K^2}\|z' - x^\star\|^2 + \tfrac{L}{\theta K^2}\|z' - x^\star\|^2 8(2\theta - 1)(1 - \tfrac{\omega_K^\star - \theta}{LD - \theta}) + \mathcal{O}(\tfrac{\rho_K}{K^3}) \\
&= \tfrac{L}{\theta K^2}\|z' - x^\star\|^2 + \tfrac{L}{K^2}\|z' - x^\star\|^2(16 - \tfrac{8}{\theta})\tfrac{LD - \omega_K^\star}{LD - \theta} + \mathcal{O}(\tfrac{\rho_K}{K^3}) \\
&\leq \tfrac{L}{\theta K^2}\|z' - x^\star\|^2 + \tfrac{L}{K^2}\|z' - x^\star\|^2(16 - \tfrac{8}{\theta})\tfrac{LD - \omega_K^\star}{LD - \theta} + \mathcal{O}(\tfrac{\rho_K}{K^3}) \\
&\leq [\tfrac{1}{2\theta} + (8 - \tfrac{4}{\theta})(\tfrac{LD - \omega_K^\star}{LD - \theta})]\tfrac{2L\|z' - x^\star\|^2}{K^2} + \mathcal{O}(\tfrac{\rho_K}{K^3}).
\end{aligned}$$

This completes the proof.

Table 3: Algorithm benchmark

| Algorithm | Explanation |
|---|---|
| GD | Vanilla gradient descent |
| GD-HB (Polyak, 1964) | Gradient descent with heavy-ball momentum |
| AGD-CVX (d'Aspremont et al., 2021) | Smooth convex version of accelerated gradient |
| AGD-SCVX (d'Aspremont et al., 2021) | Strongly convex version of accelerated gradient |
| Adam (Kingma, 2014) | Adaptive momentum estimation |
| AdaGrad (Duchi et al., 2011) | Adaptive (sub)gradient method |
| BFGS (Nocedal and Wright, 1999) | BFGS from scipy |
| L-BFGS-Mk (Nocedal and Wright, 1999) | L-BFGS with memory k from scipy (Virtanen et al., 2020) |
| HDM-Best | Practical hypergradient descent method |

## E. Additional Experiments

### E.1. Details of the Algorithms

Table 3 details the algorithms used in our experiments.

### E.2. Ablation Study of **HDM-Best**

This section evaluates the effect of different components in HDM-Best, including null-step and AdaGrad. In particular, we consider the following versions of HDM-Best

- HDM Raw.
  HDM-Best without null step and online gradient descent with constant stepsize is used.

- HDM+Null step.
  HDM-Best with null step and online gradient descent with constant stepsize is used.

- HDM+Null step+AdaGrad.
  HDM-Best with all the components.

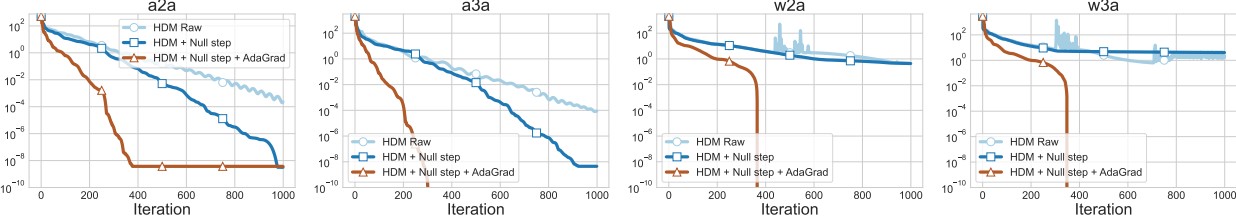

Figure 6: Ablation on different components of HDM-Best

As Figure 6 shows, both the null step and AdaGrad bring significant speedup and justify our theoretical results.

### E.3. Additional Experiments on Support Vector Machine Problems

See Figure 7 and Figure 8.

### E.4. Additional Experiments on Logistic Regression Problems

See Figure 9 and Figure 10.

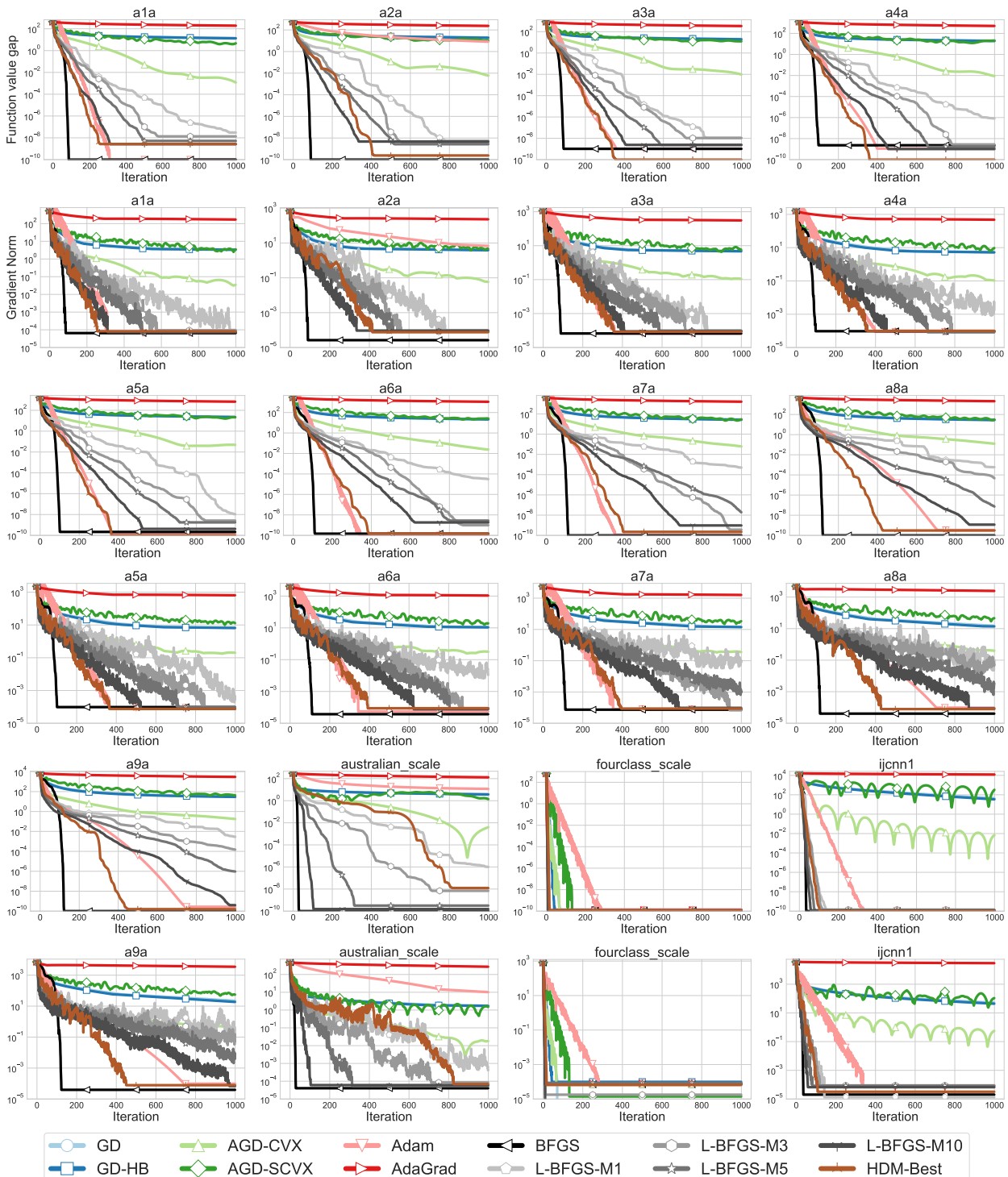

Figure 7: More experiments on support vector-machine problem

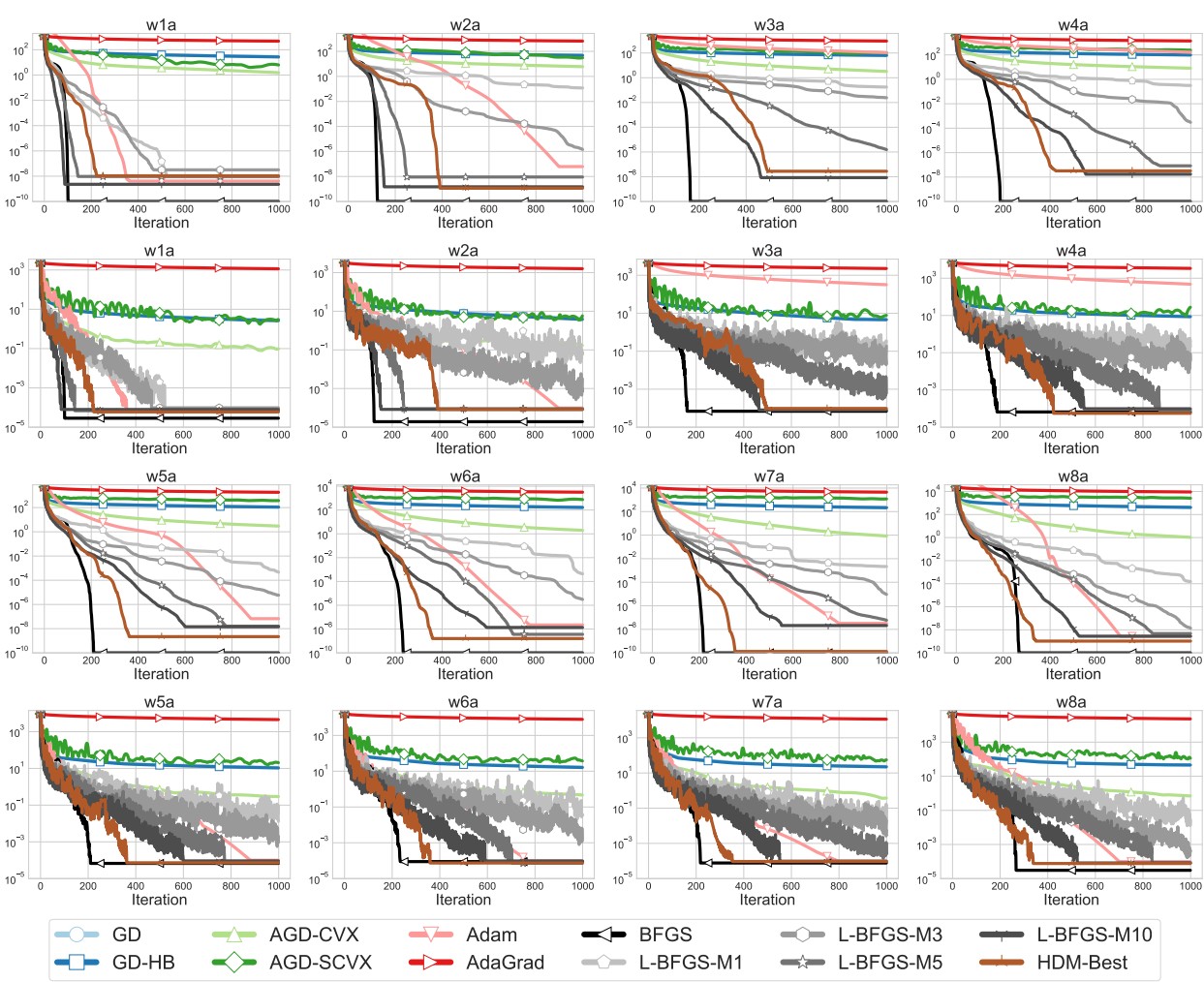

Figure 8: More experiments on support vector-machine problem

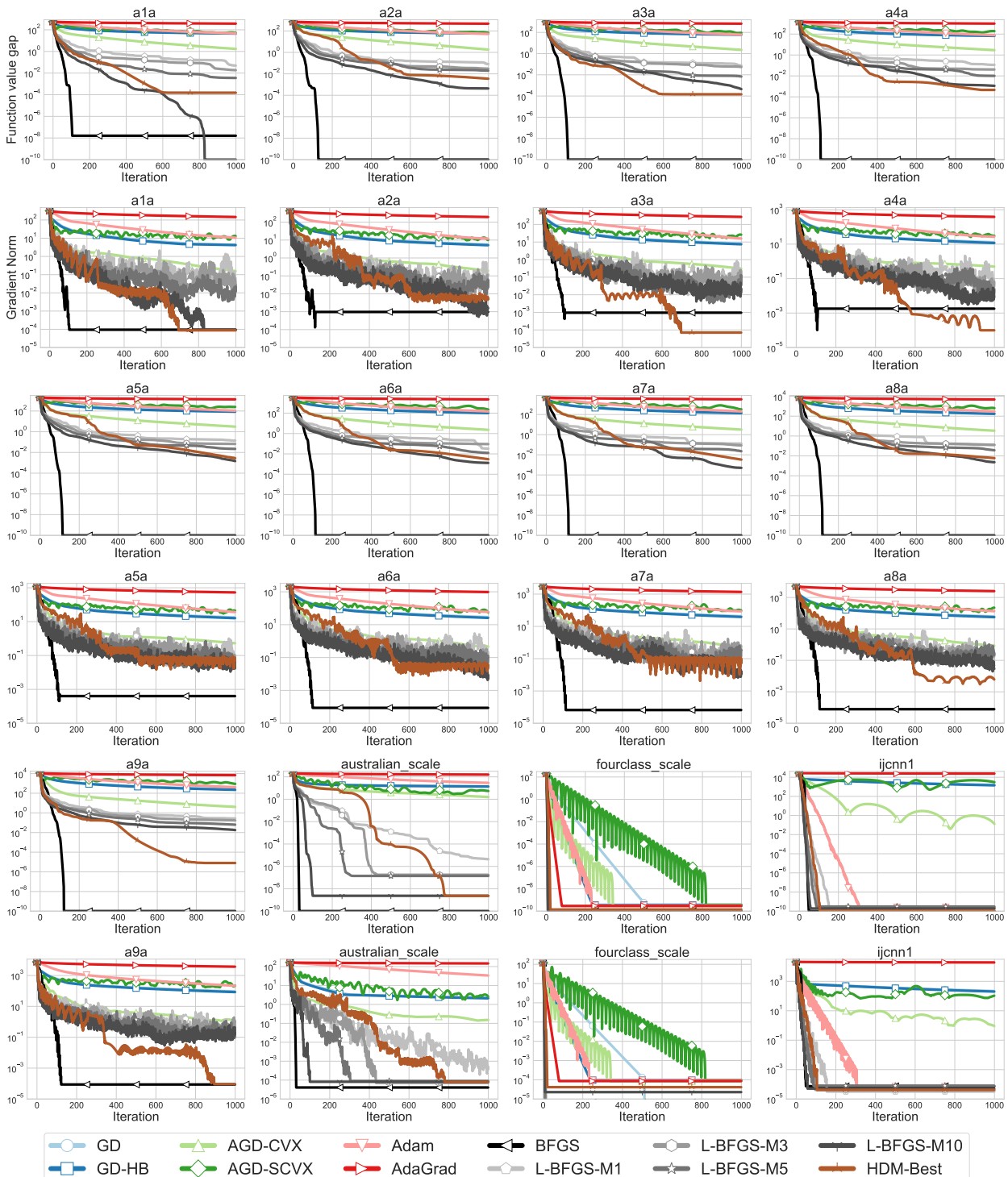

Figure 9: More experiments on logistic regression problem

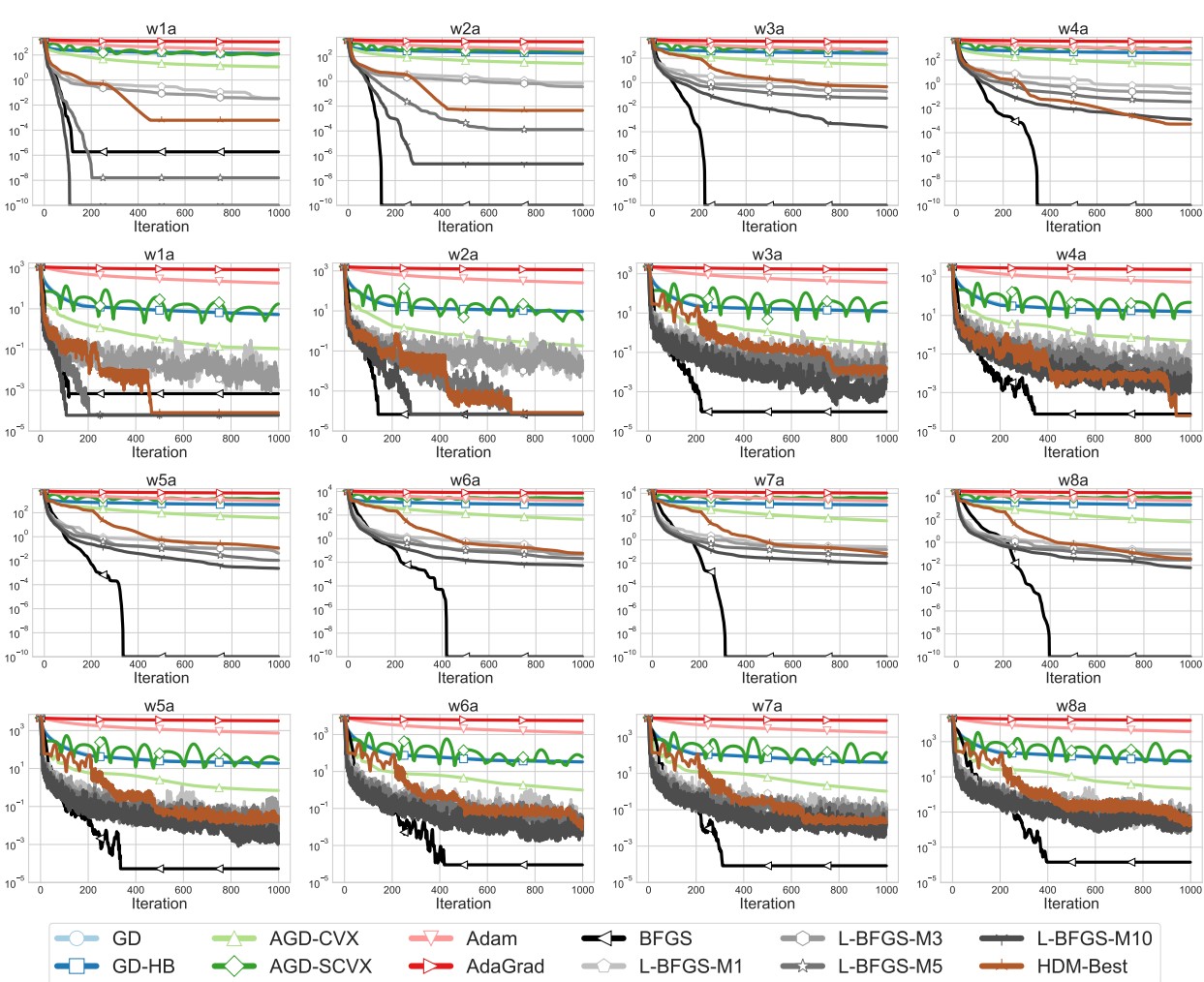

Figure 10: More experiments on logistic regression problem

### E.5. Experiments against Additional Benchmark Algorithms

This section compares `HDM-Best` with the following additional benchmark algorithms:

- `GD-LS`. Gradient descent with Armijo line-search.
- `AGD-CVX-LS`. Accelerated gradient descent with Armijo line-search applied to the descent step.
- `ADPGD`. Adaptive gradient descent without descent in Malitsky and Mishchenko (2020).
- `ADPGDACC`. `ADPGD` with acceleration in Malitsky and Mishchenko (2020).

**Algorithm Configuration.**

- Line-search condition is taken to be standard Armijo rule:

$$f(x - \alpha \nabla f(x)) \leq f(x) - \alpha c \|\nabla f(x)\|^2,$$

  where $c = 10^{-4}$ and $\alpha$ is obtained using backtracking. Whenever the test passes without backtracking, the initial backtracking stepsize $\alpha_0$ is updated by $\alpha_0 \leftarrow 1.2\alpha_0$.
- The $\lambda_0$ parameter in `ADPGD` is chosen within the range $\{0.1/L, 1/L, 10/L, 100/L\}$.
- `ADPGDACC` uses the default heuristic parameters provided by Malitsky and Mishchenko (2020).

**Results.** Table 4 shows the number of problems solved by each additional benchmark algorithms. The numbers in parentheses indicate the total number of test problems. We observe that although line-search improves the convergence behavior of the baseline algorithms, the algorithms with a single stepsize seem less competitive with algorithms with a diagonal preconditioner when a moderate-to-high-accuracy solution is needed. Our requirement $\|\nabla f(x)\|_\infty \leq 10^{-4}$ (which corresponds to function value gap of $10^{-7} \sim 10^{-9}$) is often too high for typical first-order methods.

Table 4: Number of solved problems for additional benchmark algorithms.

| Algorithm/Problem | SVM (33) | Logistic Regression (33) |
|---|---|---|
| `GD-LS` | 9 | 9 |
| `AGD-CVX-LS` | 10 | 9 |
| `ADPGD` | 9 | 9 |
| `ADPGDACC` | 11 | 11 |
| `HDM-Best` | 32 | 21 |

