# OpenReview forum: "Provable and Practical Online Learning Rate Adaptation with Hypergradient Descent"
_ICML.cc/2025/Conference — ICML 2025 poster_

### Official Review · Reviewer_mYzg · 2025-03-03

**Overall Recommendation:** 4

**Summary:**

The authors addressed the challenge of establishing convergence of the hypergradient descent heuristic in this submission. They provided the first rigorous theoretical foundation for hypergradient descent and introduce a novel online learning perspective that extends to other first-order methods with adaptive hyperparameter updates. They also proposed HDM-Best, an efficient variant of HDM that performs competitively with the widely used L-BFGS method on convex problems. They also conduct numerical experiments and the empirical results are consistent with the theoretical analysis.

**Claims And Evidence:**

The claims in this submission are supported by both theoretical proof and numerical experiments.

**Essential References Not Discussed:**

The following are the newest results about the superliner convergence rates of the quasi-Newton methods and I suggest that the authors should refer these works and compare the superliner convergence rates of the Hypergradient Descent in Theorem 3.3 with all these rates:

https://arxiv.org/abs/2002.00657

https://arxiv.org/abs/2003.09174

https://arxiv.org/abs/2004.14866

https://arxiv.org/abs/2003.13607

https://arxiv.org/abs/2404.16731

https://arxiv.org/abs/2404.01267

**Experimental Designs Or Analyses:**

The empirical results shown in the numerical experiments are valid after checking.

**Methods And Evaluation Criteria:**

The methods from theoretical analysis and benchmark datasets from empirical results make sense for the convergence analysis of the Hypergradient Descent method.

**Other Comments Or Suggestions:**

Please compare the superliner convergence rates of the Hypergradient Descent in Theorem 3.3 with the latest non-asymptotic superlinear convergence rates of the quasi-Newton methods listed in the essential references not discussed section.

**Other Strengths And Weaknesses:**

This paper is a high-quality optimization theory submission where the authors proved the first rigorous convergence analysis for
HDM, including both global and local convergence guarantees. They developed and analyze two improved variants of HDM:
HDM + heavy-ball momentum and HDM + Nesterov momentum. They provided all the mathematical proof and the emprical results from the numerical experiments are consistent with the theoretical results. This submission deserves a publication for ICML.

**Questions For Authors:**

No other questions

**Relation To Broader Scientific Literature:**

All the related works discussed in the section 1.1 are closely related to the contributions of this submission.

**Theoretical Claims:**

The proof of the theatrical claims shown in the appendix is correct after checking

---

> ### Author Rebuttal · Authors · 2025-03-30
>
> Thank you for your valuable feedback on our paper and for acknowledging the contribution of our results!
>
> **Essential References Not Discussed**
>
> Thank you for pointing out these references. We will thoroughly discuss the superlinear convergence results in the revised version. Here, we give a brief comparison with the existing results in terms of the superlinear convergence rate, where for HDM we plug in $\rho_K = \mathcal{O}(\sqrt{K})$ and get $\mathcal{O}((\frac{1}{\sqrt{K}})^K)$ convergence.
>
> | Paper |       Superlinear convergence rate       |
> | :---: | :--------------------------------------: |
> |  [1]  |    $\mathcal{O}(e^{-\frac{1}{2}K^2})$    |
> |  [2]  | $\mathcal{O}(e^{-\frac{1}{2} K \log K})$ |
> |  [3]  | $\mathcal{O}(e^{-\frac{1}{2}K \log K})$  |
> |  [4]  | $\mathcal{O}(e^{-\frac{1}{2}K \log K})$  |
> |  [5]  |       $\mathcal{O}(e^{-K \log K})$       |
> |  [6]  |       $\mathcal{O}(e^{-K \log K})$       |
> |  HDM  | $\mathcal{O}(e^{-\frac{1}{2} K \log K})$ |
>
> Convergence rate in terms of $K$ is discussed since these references uses different notation for the problem-dependent constants and convergence metric, but in the revised version of the paper we will choose a unified notation and compare them, with problem-dependent constants.
>
> We again appreciate the constructive feedback of the reviewer and thank you for the time spent on our paper!
>
> **References**
>
> [1] Rodomanov, A., & Nesterov, Y. (2021). Greedy quasi-Newton methods with explicit superlinear convergence. *SIAM Journal on Optimization, 31*(1), 785–811.
>
> [2] Rodomanov, A., & Nesterov, Y. (2021). Rates of superlinear convergence for classical quasi-Newton methods. *Mathematical Programming*.
>
> [3] Rodomanov, A., & Nesterov, Y. (2021). New results on superlinear convergence of classical quasi-Newton methods. *Journal of Optimization Theory and Applications, 188*, 744–769.
>
> [4] Jin, Q., & Mokhtari, A. (2021). Non-asymptotic superlinear convergence of standard quasi-Newton methods. arXiv preprint arXiv:2003.13607.
>
> [5] Jin, Q., Jiang, R., & Mokhtari, A. (2025). Non-asymptotic global convergence analysis of BFGS with the Armijo-Wolfe line search. arXiv preprint arXiv:2404.16731.
>
> [6] Jin, Q., Jiang, R., & Mokhtari, A. (2024). Non-asymptotic global convergence rates of BFGS with exact line search. arXiv preprint arXiv:2404.01267.

---

### Official Review · Reviewer_htFv · 2025-03-09

**Overall Recommendation:** 2

**Summary:**

In this paper, the authors study the hypergradient descent method for smooth convex optimization, where the preconditioner is updated using online gradient descent. Building on the online learning framework of (Gao et al., 2024), they prove sublinear and linear convergence rates in the convex and strongly-convex settings, respectively. Under additional assumptions, they further prove local superlinear convergence and show that the preconditioner converges to the Hessian at the optimum. The authors also analyze two momentum variants, where both the preconditioner and the momentum parameter are updated via online gradient descent. Finally, they develop a practical variant and demonstrate that it performs comparably to L-BFGS while requiring less memory.

## Update after rebuttal
During the rebuttal, the authors conducted additional experiments to include suggested baselines, such as GD/AGD with line search and the adaptive gradient method of [MM 2020]. The proposed method continues to outperform these baselines, which further validates its empirical efficiency.

However, I remain on the fence regarding acceptance. On the theoretical side, the superlinear convergence guarantees apply only to the full-matrix version of the algorithm, which differs from the diagonal version used in the experiments. This discrepancy makes me somewhat cautious about the strength of the theoretical claims and their alignment with the empirical results. On the experimental side, the case would be more compelling if the authors included additional large-scale experiments in stochastic, non-convex settings.

**Claims And Evidence:**

Overall, most of the claims in the paper are well supported by theorems and their proofs. That said, I have some reservations regarding certain statements:
- In Section 3.1, the authors claim that HDM can adapt to the local optimization landscape since the choice of $P\_K^\*$ in Theorem 3.1 or $\\{\hat{P}\_k\\}$ in Theorem 3.2 only depends on the past trajectory $\\{x^k\\}_{k \leq K}$. However, this argument appears somewhat imprecise. From my understanding, there is no guarantee that the iterates of HDM will remain confined to a local region. Given that the initial point is typically far from the optimal solution, the iterates may traverse a large region of space before converging. As such, the convergence bound likely still depends on the global properties of the objective function rather than solely on its local behavior.
- In Section 3.3, the authors establish that HDM achieves local superlinear convergence and learns the Hessian at the optimum. However, these results rely on the crucial assumption that $[\nabla^2 f(x^*)]^{-1} \in \mathcal{P}$, which is unlikely to hold when $\mathcal{P}$ is the set of diagonal matrices, as considered in the numerical experiments. This mismatch between the theoretical assumptions and practical implementation should be better clarified.

**Essential References Not Discussed:**

I suggest that the authors to include a more detailed discussion and comparison with the adaptive gradient method by Malitsky and Mishchenko (2020; 2024), which introduces a step-size selection strategy based on observed gradient differences. Moreover, it is worth mentioning the works of Zhuang et al. (2019) and Jiang et al. (2023), both of which employ an online learning framework to adaptively update hyperparameters in optimization algorithms, such as the step size and Hessian approximation matrix.

That said, I would like to note that these references do not diminish the contributions of this work.

**Experimental Designs Or Analyses:**

I have some concerns regarding the choice of baselines in the numerical experiments.
- It appears that the authors use a grid search to find the optimal $\eta_p$ for HDM-Best but employ a fixed step size for GD, GD-HB, AGD-CVX, and SGD-SCVX. This seems unfair, as all the algorithms should receive the same level of tuning. To ensure a fair comparison, the authors should also conduct a grid search for the step size of the other methods.
- Additionally, I think it is important to consider GD, GD-HB, AGD-CVX, and SGD-SCVX with the classical backtracking line search. Notably, the use of a null step in HDM-Best resembles a line search procedure, where the step size is reduced until the function value decreases sufficiently. Since both rely on function evaluations to adjust the step size, these line-search-based methods should also be included as baselines.
- Another relevant baseline is the adaptive gradient method proposed by Malitsky and Mishchenko (2020; 2024). As further detailed in "Essential References Not Discussed", their algorithm is also designed to adaptively select the step size for gradient descent in deterministic convex optimization problems. Including this method in the comparison would strengthen the evaluation.
---

Malitsky, Y., & Mishchenko, K (2020). Adaptive Gradient Descent without Descent. ICML 2020.

Malitsky, Y., & Mishchenko, K (2024). Adaptive proximal gradient method for convex optimization. NeurIPS 2024.

**Methods And Evaluation Criteria:**

Yes, the hypergradient descent method is a common heuristic for hyperparameter selection, and LIBSVM is also a common benchmark dataset for deterministic convex problems.

**Other Comments Or Suggestions:**

- Most of the convergence results in this paper depend on the diameter of the set $\mathcal{P}$ defined in **A3**. However, note that this quantity can scale with the problem dimension $d$. Thus, it would be better to make this dimensional dependence more explicit.
- In Theorem 3.1, the authors follow the standard analysis of online gradient descent, which achieves a regret bound of $\rho_K= D(LD+1)\sqrt{K}$ in the given setting. However, since $h_x(P)$ is known to be $L$-smooth from Lemma 2.1, it may be possible to leverage a small-loss bound to establish a constant regret independent of the time horizon. Would this alternative approach lead to a better convergence bound?

Minor suggestions:
- Some of the notations are not clearly defined, leading to potential confusion. For instance, $h_x(\alpha)$ and $h_x(d)$ in Lemma 2.1 are not properly introduced. Moreover, $\rho_K$ is used throughout the paper, but it is only defined in (5), where it is not particularly prominent (I initially overlooked it on my first read).

**Other Strengths And Weaknesses:**

Another concern is that the theoretical results are not particularly strong.
- Theorem 3.1 appears to be nearly identical to Theorem 6.1 in Gao et al. (2024). The only difference is that the bound in (A1) also takes the minimum with $\phi(x^1)$, which follows directly from the definition of null steps. While Theorem 3.2 is new, its proof largely follows the same strategy as Theorem 3.1, relying on a standard application of the dynamic regret argument.
- As mentioned earlier, while the authors establish local superlinear convergence, this result hinges on the assumption that $[\nabla^2 f(x^*)]^{-1} \in \mathcal{P}$. Thus, it may not be applicable when $\mathcal{P}$ is the set of diagonal matrices, which are the main focus of the paper.
- For HDM with heavy-ball momentum, Theorem 4.1 does not demonstrate an improved convergence rate compared to vanilla HDM in Theorem 3.1. Moreover, for HDM with Nesterov momentum, while Theorem 4.2 seemingly establishes a faster rate of $O(1/K^2)$, this result actually requires the initial suboptimality gap to be already in the order of $O(1/K^2)$, as discussed in Theorem D.1. This limitation diminishes its significance.

**Questions For Authors:**

To summarize, my concerns are two-fold:
- The numerical comparison omits important baseline methods and does not properly tune their step sizes. As a result, it remains unclear whether the proposed method truly outperforms existing approaches.
- The theoretical results are not particularly strong. In particular, they do not demonstrate a clear advantage over standard methods such as gradient descent with line search.

I am willing to raise my score if these two issues are properly addressed.

**Relation To Broader Scientific Literature:**

This paper analyzes the convergence properties of hypergradient descent and its extensions, a strategy for adaptively selecting the step size.

- The most relevant prior work is Gao et al. (2024), which also employs an online learning framework with hypergradient feedback to analyze hypergradient descent methods. The main additional contributions of this paper are the local superlinear convergence analysis and the introduction of two momentum-based variants.

- Other adaptive step-size selection strategies include line-search-based methods, AdaGrad, and the adaptive gradient method proposed by Malitsky and Mishchenko (2020; 2024). These approaches provide alternative ways to adjust step sizes dynamically based on problem-dependent quantities.

- In terms of methodology, this paper selects the step size by formulating an online learning problem and applying online gradient descent. In this sense, their framework is similar to that of Zhuang et al. (2019), where the authors address stochastic non-convex optimization and adapt the step size by running an online learning algorithm on a surrogate loss. Additionally, the propose framework is related to Jiang et al. (2023), which proposes learning a Hessian approximation matrix in an online manner rather than a preconditioner.

----

Malitsky, Y., & Mishchenko, K (2020). Adaptive Gradient Descent without Descent. ICML 2020.

Malitsky, Y., & Mishchenko, K (2024). Adaptive proximal gradient method for convex optimization. NeurIPS 2024.

Zhuang, Z., Cutkosky, A., & Orabona, F. (2019). Surrogate losses for online learning of stepsizes in stochastic non-convex optimization. ICML 2019.

Jiang, R., Jin, Q., & Mokhtari, A. (2023). Online learning guided curvature approximation: A quasi-Newton method with global non-asymptotic superlinear convergence. COLT 2023.

**Theoretical Claims:**

Yes, I checked most of the proofs in the appendix and they appear correct, except for some typos (specified in "Other Comments Or Suggestions").

---

> ### Author Rebuttal · Authors · 2025-03-31
>
> Thank you for your time and insightful comments. Below are our responses to your two major concerns and other comments. If we have addressed your concerns, we kindly request a re-evaluation of the score, and please let us know if you have further questions.
>
> **Major concerns ("Questions")**
>
> 1. Numerical comparison
>
>    We added the suggested experiments to https://anonymous.4open.science/r/hypergrad-00CB, including line-search GD/AGD and [YK2020]. On the benchmark data, line-search improves the performance of GD/AGD, but neither suffices to achieve $10^{-4}$ gradient norm tolerance. Please see `add_exp.pdf` for more details.
>
> 2. htFv worries our theory is not particularly strong and shows no clear advantage over GD with line search
>
>    We politely disagree that there is no clear advantage w.r.t. line search GD.
>
>    Consider an ill-conditioned diagonal quadratic objective. Even with exact line-search, GD can converge arbitrarily slowly on this problem [1]. HDM does not! Rather, its diagonal version converges superlinearly. In the revision, we will use this example to demonstrate the advantage of HDM over GD with line-search, and more generally, the strength of our theory, which provides
>
>    - the first rigorous guarantee for HDM with matching empirical performance.
>    - a proof HDM achieves superlinear convergence with only first-order information. The first family of first-order algorithms with superlinear convergence, discovered in 1959, is quasi-Newton, which uses secant equations to approximate second-order curvature. Our paper discovers the second such family!
>    - novel proof techniques that show acceleration either 1) when a convex Lyapunov function is available (like HB) or 2) when the descent lemma is used in the analysis (like AGD).
>    - momentum extensions that perform better than any baseline algorithm (fixing the same memory budget).
>
>    Our work lays a foundation for future research, and we believe these contributions together are strong enough for a conference submission.
>
> **Claims And Evidence**
>
> 1. Path-length argument and adaptivity to the local landscape
>
>    Thanks for your insightful comment! In a word, the path-length argument unifies HDM's behavior when far from/near the optimum without complicating the analysis. This is a novel best-of-both-worlds guarantee for an optimization algorithm.
>
>    Due to space limit, we kindly refer the reviewer to our response to Reviewer fbAc Other Strengths And Weaknesses at https://openreview.net/forum?id=NkVCB1Cpgl&noteId=2xn5Z6a9Xd.
>
> 2. Require $\nabla f(x^\star)^{-1}\in\mathcal{P}$
>
>    You are correct that our superlinear convergence result requires this assumption; we'll clarify it in the revision.
>    However, all other non-second-order methods achieving superlinear convergence (the quasi-Newton (QN) family) also store an n x n preconditioner.
>    Moreover, HDM can achieve linear convergence competitive with the best possible preconditioner even if $\nabla f(x^\star)^{-1}\not\in\mathcal{P}$, while QN has no such guarantee.
>
> **Experimental Designs/Analyses**
>
> Please see our response to Major concerns.
>
> **Essential References Not Discussed**
>
> Thanks for these refs! We have added them to the revision.
>
> **Other Strengths And Weaknesses**
>
> 1. Similarity to [Gao2024]
>
>    Our paper builds on [Gao2024], and we restate Thm 3.1 for completeness, although it's a slightly refined version. Our dynamic regret perspective (Thm 3.2) offers new insights beyond [Gao2024], bounding regret w.r.t the best preconditioner **sequence** (see Claims And Evidence). The analysis is simple, but it *captures what HDM is doing* without complicated analysis and provides new insights. We believe this type of result is valuable.
>
> 2. Weakness of HDM-HB/HDM-AGD
>
>    Since HDM works by preconditioning, it improves the *problem-dependent constant* instead of the *convergence rate*. Hence, HDM-HB/HDM-AGD inherits the same rate of HB/AGD but gives improved constants:
>
>    - HDM-HB's constant is determined by hindsight w.r.t $h_x(P,\beta)$, which can be much smaller than $h_x(P)$.
>
>    - HDM-AGD does not improve and requires a warmstart.
>
>      It's true; but HDM-AGD can improve the constant hidden by the big-O! e.g., run AGD for $K$ iterations. If $\sum_{k=1}^K h_{x^k}(P)/K\ll-1/(2L)$ (as always holds in our experiments), then K iters of HDM-AGD (with the AGD warmstart) can outperform $K$ more iters of AGD.
>
> **Other Comments**
>
> 1. Dimension dependence
>
>    We'll discuss it in the revision.
>
> 2. Could a small loss bound argument provide constant regret guarantee and so faster convergence?
>
>    Alas no: this argument requires a global lower bound on $h_x$ which is not known for general convex problems, though possibly a refinement of the argument might work.
>
> 3. Notation
>
>    Thanks! We have fixed these issues.
>
> We again thank you for your time in the review process! Please let us know if you have further questions.
>
> **Reference**
>
> [1] Saad, Y. (03). Iterative methods for sparse linear systems.

---

> > ### Comment · Reviewer_htFv · 2025-04-06
> >
> > I thank the authors for their detailed responses in addressing my concerns. I especially appreciate the additional experiments, which show that the proposed method still outperforms GD/AGD with line search and the adaptive gradient method of [MM 2020].
> >
> > That said, I remain unconvinced about the strength of the theoretical contributions. My takeaway from the experiment is that much of HDM's strong empirical performance can be attributed to the use of a diagonal preconditioner, which does make sense intuitively. However, the main theorems in this paper do not rigorously rigorously quantify this advantage.
> > - First, since $\mathcal{P}$ is the set of diagonal matrices for HDM-Best, the local superlinear convergence results in Section 3.3 do not apply here.
> > - Second, it seems unclear whether the upper bounds in Theorems 3.1, 3.2, 4.1 and 4.2 are provably better than those for GD or AGD in theory. Moreover, since $D = \mathrm{diam}(\mathcal{P}) = \Theta(\sqrt{d})$ for the set of diagonal matrices, we have $\rho_K = \Theta(d \sqrt{K})$. This implies that the guarantees only become meaningful when $K \geq d^2$ (since we require $K \geq \rho_K$), which can be prohibitively large in high-dimensional settings.
> >
> > After some deliberation, I have decided to maintain my current score. I believe it would strengthen the paper to better align the theoretical results with the observed empirical performance. That said, if the AC and other reviewers are in favor of accepting the paper, I will not oppose the decision.

---

> > > ### Author Response · Authors · 2025-04-07
> > >
> > > **Response to Reviewer htFv**
> > >
> > > We appreciate your active engagement in the review process. We respect your decision to maintain the score.
> > >
> > > But we would like to make two clarifications.
> > >
> > > 1. Performance of HDM-Best
> > >
> > >    Diagonal HDM is a (memory and computation) efficient version of full-matrix HDM, just as L-BFGS is an efficient version of BFGS. In both cases, the more efficient version can be weaker in theory but preserve outstanding performance. HDM-Best uses additional techniques (e.g., momentum) compared to diagonal HDM to perform even better on large-scale optimization problems. The diagonal preconditioner balances the power of the hindsight preconditioner with computational efficiency. Full-matrix HDM converges in fewer iterations, but its iteration requires an additional $\mathcal{O}(n^2)$-element-wise projection onto a bounded box (since we do not require $P$ to be positive definite) and $\mathcal{O}(n^2)$ memory; this cost is not worth the iteration-number improvement for large-scale applications. Depending on the problem size/memory budget, one can always switch to the full-matrix version of HDM-Best and recover superlinear convergence--our analysis is still compatible (taking hindsight $\beta=0, P = [\nabla^2 f(x^\star)]^{-1}$).
> > >
> > > 3. There is no improvement in upperbound
> > >
> > >     Our upperbound is a worst-case upperbound, while explaining the performance of an optimizer with good empirical performance generally requires *beyond* worst-case analysis. We provide a best-of-both-worlds type result: our strong problem-dependent convergence results maintain worst-case convergence guarantees through tools in online convex optimization (Theorem 3.2), which, to our knowledge, is a novel contribution. But we agree that our analysis introduces a dimension-dependent constant, which we will discuss more thoroughly in the revision and actively address in the future work.
> > >
> > > Last, despite a lack of theoretical guarantees, papers on HDM have recently been published at top conferences on the basis of *strong empirical performance alone* [1-5]. Our paper develops the first convergence theory for HDM and uses that theory to develop new algorithms with stronger empirical performance.
> > >
> > > Thank you again for your efforts in the review process!
> > >
> > > **References**
> > >
> > > [1] Baydin, A. G., Cornish, R., Rubio, D. M., Schmidt, M., & Wood, F. (2018, February). Online Learning Rate Adaptation with Hypergradient Descent. In *International Conference on Learning Representations*.
> > >
> > > [2] Micaelli, P., & Storkey, A. J. (2021). Gradient-based hyperparameter optimization over long horizons. *Advances in Neural Information Processing Systems*, *34*, 10798-10809.
> > >
> > > [3] Chandra, K., Xie, A., Ragan-Kelley, J., & Meijer, E. (2022). Gradient descent: The ultimate optimizer. *Advances in Neural Information Processing Systems*, *35*, 8214-8225.
> > >
> > > [4] Ozkara, K., Karakus, C., Raman, P., Hong, M., Sabach, S., Kveton, B., & Cevher, V. (2024, July). MADA: Meta-adaptive optimizers through hyper-gradient descent. In *International Conference on Machine Learning*.
> > >
> > > [5] Wang, Z., Wang, J., & Li, A. (2024) FedHyper: A Universal and Robust Learning Rate Scheduler for Federated Learning with Hypergradient Descent. In *International Conference on Learning Representations*.

---

### Official Review · Reviewer_DNfY · 2025-03-18

**Overall Recommendation:** 3

**Summary:**

This paper presents the first rigorous convergence analysis of the hypergradient descent method (HDM), a technique for adaptive stepsize selection in optimization that has been used heuristically for 25 years. The authors develop a theoretical framework based on online learning to analyze HDM's convergence properties and create improved variants with momentum. Their analysis shows that HDM exhibits several desirable properties: it automatically adapts to the local optimization landscape, achieves local superlinear convergence for strongly convex problems, and learns the inverse Hessian at the optimum. The authors identify reasons for HDM's previously reported instability and propose solutions including null steps and improved online learning algorithms. They develop two momentum-based variants: HDM with heavy-ball momentum (HDM-HB) and HDM with Nesterov momentum (HDM-AGD). Their best practical variant, HDM-Best, combines diagonal preconditioners, heavy-ball momentum, and AdaGrad updates. Experiments on deterministic convex problems show that HDM-Best outperforms other adaptive first-order methods and often matches L-BFGS with less memory usage.

**Claims And Evidence:**

The claims in the paper are generally well-supported by theoretical analysis and experimental evidence:

1. Convergence properties of HDM: The authors provide clear theoretical results analyzing both global convergence (Theorems 1 and 2) and local superlinear convergence (Theorem 3). The analysis explains how HDM adapts to the local landscape and why it can achieve superlinear convergence for strongly convex problems.
2. Understanding HDM's instability: The paper convincingly explains why vanilla HDM can appear unstable in early iterations and proposes effective solutions through null steps and improved online learning algorithms. Figure 1 provides visual evidence of this instability and how their solutions address it.
3. HDM learns the inverse Hessian: Theorem 5 and Lemma 3 provide solid theoretical support for the claim that HDM converges to the inverse Hessian at the optimum, connecting it to quasi-Newton methods.
4. Momentum-based variants: The authors develop and analyze HDM-HB (Algorithm 2) and HDM-AGD (Algorithm 3), providing convergence guarantees for both in Theorems 4 and 6.
5. Practical performance: The experimental results in Section 6 support the claim that HDM-Best outperforms other adaptive first-order methods and competes with L-BFGS. Table 1 shows HDM-Best solves 32/33 SVM problems and 21/33 logistic regression problems, similar to L-BFGS-M10 and BFGS.

The evidence is convincing for convex optimization problems. The authors are appropriately cautious about generalizing their results beyond this context.

**Essential References Not Discussed:**

No

**Experimental Designs Or Analyses:**

The experimental design is sound and the analyses are valid:

1. The experiments use a diverse set of 33 benchmark datasets from LIBSVM for both SVM and logistic regression problems, providing a robust test of algorithm performance.
2. The algorithm comparison is comprehensive, including multiple variants of gradient descent, momentum methods, adaptive methods, and quasi-Newton methods.
3. The hyperparameter selection process is clearly described, with reasonable ranges for each algorithm.
4. The evaluation metrics (function value gap and gradient norm) are plotted against the number of gradient evaluations, providing a fair comparison of computational efficiency.
5. Table 1 provides a summary of algorithm performance across all test problems, while Figures 3 and 4 show detailed convergence plots for representative examples.

The experimental results align with the theoretical findings, showing that HDM-Best achieves faster convergence than other adaptive first-order methods and often matches L-BFGS. The authors' claim that HDM-Best uses less memory than comparable L-BFGS variants is supported by their description of the algorithm.

**Methods And Evaluation Criteria:**

The methods and evaluation criteria are appropriate and well-designed for the problem:

1. The theoretical analysis uses online learning tools to provide sharp convergence bounds for HDM and its variants. This approach is novel for analyzing HDM and yields insights about its behavior.
2. The experimental evaluation uses standard benchmark datasets from LIBSVM for two common convex optimization problems: support vector machines and logistic regression.
3. The comparison algorithms represent a comprehensive set of optimization methods, including standard gradient descent, momentum methods, adaptive methods (Adam, AdaGrad), and quasi-Newton methods (BFGS, L-BFGS).
4. The evaluation metrics (number of solved problems, function value gap, gradient norm) are standard and appropriate for assessing optimization algorithm performance.
5. The experimental setup is clearly described with appropriate details about algorithm configuration, stopping criteria, and maximum oracle access. This enables reproducibility and fair comparison.

The only potential limitation is that the evaluation focuses solely on deterministic convex problems, whereas the original motivation for HDM included stochastic optimization. However, the authors make this focus clear, and the insights gained should still be valuable for broader applications.

**Other Comments Or Suggestions:**

No

**Other Strengths And Weaknesses:**

No

**Questions For Authors:**

1. Your analysis focuses on deterministic convex optimization, but HDM was originally proposed for stochastic optimization. Do you expect your theoretical results to extend to the stochastic setting, and what modifications would be necessary? Would the instability issues be more or less severe with stochastic gradients?
2. You mention that HDM-Best uses approximately the same memory as L-BFGS-M1 but often matches the performance of L-BFGS-M5 or L-BFGS-M10. Could you elaborate on the memory requirements of HDM-Best compared to L-BFGS variants more precisely, and clarify the trade-offs?
3. The null step approach requires an additional function evaluation at each iteration. Have you explored alternative stability-enhancing approaches that avoid this extra cost? For instance, would trust-region approaches be applicable here?
4. Your theoretical analysis reveals connections between HDM and quasi-Newton methods. Have you explored hybrid approaches that combine insights from both methods? For example, could one incorporate secant conditions into the preconditioner updates of HDM?
5. The experimental results show that HDM-Best performs better on SVM problems (32/33 solved) than on logistic regression problems (21/33 solved). Do you have insights into why this performance difference exists? Are there particular problem characteristics that make HDM more effective?

**Relation To Broader Scientific Literature:**

The paper effectively situates its contributions within the broader scientific literature:

1. It connects to the extensive literature on adaptive first-order methods, including AdaGrad, Adam, and parameter-free stepsizes, providing a theoretical foundation for a different approach to adaptation.
2. It builds on the history of hypergradient descent, tracing its origins to Almeida (1999) and its rediscovery by Gunes (2018), while advancing beyond previous analyses by Rubio (2017), Kunstner (2024), and Gao (2024).
3. It establishes connections between HDM and quasi-Newton methods, showing both theoretical similarities (learning the inverse Hessian) and practical differences (how they measure and update the preconditioner).
4. It applies ideas from online learning theory to analyze offline optimization algorithms, following recent work that has demonstrated the power of this approach.

The authors appropriately credit prior work and clearly delineate their novel contributions, particularly the rigorous convergence analysis and the development of momentum-based variants.

**Theoretical Claims:**

The theoretical claims in the paper appear correct and are supported by detailed proofs:

1. Static adaptivity (Theorem 1): This result builds on the online learning framework and shows how HDM's convergence rate depends on the trajectory-dependent term γ_K^* reflecting the local landscape.
2. Dynamic adaptivity (Theorem 2): This extends the analysis to show HDM can adapt to different regions of the landscape, using dynamic regret concepts from online learning.
3. Local superlinear convergence (Theorem 3): This result establishes the superlinear convergence of HDM near the optimum for strongly convex problems with Lipschitz Hessian, explaining the fast convergence observed in practice.
4. Convergence of the preconditioner (Theorem 5): This result shows HDM learns the inverse Hessian at the optimum under appropriate conditions.
5. Convergence guarantees for HDM-HB (Theorem 4) and HDM-AGD (Theorem 6): These extend the analysis to the momentum-based variants.

The proofs use techniques from online learning theory, convex optimization, and analysis of first-order methods. The assumptions are standard in the optimization literature (smoothness, convexity, Lipschitz Hessian) and appropriately specified throughout the paper.

---

> ### Author Rebuttal · Authors · 2025-03-30
>
> Thank you for your feedback on our paper.
>
> **Questions**
>
> 1. Extension to the stochastic setting
>
>    This is a very good question. One major bottleneck in extending HDM to the stochastic case is that the hypergradient feedback can be corrupted by noise, especially the gradient norm denominator of the feedback, which can only be estimated using random samples. However, in some easier settings, such as over-parametrized and finite-sum problems, we still expect HDM to work.
>
> 2. Memory usage of HDM-Best and more clarification
>
>    Thanks for raising this point. As we mentioned in the Experiment setup (Line 333) and in Appendix A, for an optimization problem with variable dimension $n$, HDM-Best uses $\mathcal{O}(7 n)$ memory while L-BFGS with memory size $m$ uses $\mathcal{O}((5 + 2m) n)$ memory. Hence for large-scale problems, HDM has comparable memory usage compared to L-BFGS with memory size 1, and in practice, HDM-Best often has competitive performance with L-BFGS-5 or L-BFGS-10. Therefore, we expect HDM to be useful in memory-intensive nonlinear optimization tasks.
>
> 3. Stability-enhancing techniques other than null step
>
>    Our current theoretical analysis requires a null step to work, but in practice, we observe that after a good preconditioner is learned, the null step is rarely triggered. It is generally safe to turn it off after monotonicity is observed for many iterations. Trust-region methods might be useful to extend HDM to nonconvex optimization, but the trust region methods are generally not worthwhile for large-scale convex optimization, as the trust region subproblem is too expensive to solve the subproblem.
>
> 4. Hybrid approach of HDM and quasi-Newton
>
>    This is an interesting idea. Yes, theoretically, when the iterate $x^k$ is far from optimality and Hessian does not behave stably, it's more favorable to use HDM since it still has guarantees with respect to a high-quality preconditioner. When $x^k$ is near $x^\star$, using secant equations more efficiently captures the local curvature around the optimal solution, and quasi-Newton methods can be more suitable locally. As with incorporating the secant equation into HDM update, we feel it's an interesting future direction worth exploring.
>
> 5. Difference between performance on SVM and logistic regression
>
>    The smoothed SVM problems we use take the simple form $\\|[A x- b]_+\\|^2$ and is friendly to first-order methods. For the logistic regression problem, we observe that typically $x^\star$ can have a very large norm, and the Hessian around the optimal solution does not have strong curvature. Therefore, we find first-order methods slow down on these problems.
>
> Thank you for the time spent in the review process.

---

### Official Review · Reviewer_fbAc · 2025-03-19

**Overall Recommendation:** 3

**Summary:**

The paper analyzes the local and global convergence behavior of the hypergradient descent method using online learning. Underlying the analysis is interpreting hypergradient descent as doing online gradient descent on “hypergradient feedback,” a proxy loss function that measures the quality of a preconditioner. Concretely, they provide a sharpened version of a prior result due to Gao et al. (2024) which ties the difference in function value of the method’s solution versus the optimal one to the average hypergradient feedback acumulated.

In addition, the authors propose improved variants of hypergradient descent which have much more appealing empirical behavior, and conduct extensive experiments to characterize the empirical convergence behavior of the proposed methods, compared to a comprehensive set of baselines/alternative optimization methods.

**Claims And Evidence:**

Yes.

**Essential References Not Discussed:**

I think more references should be added in the very beginning following the remark that the choice of stepsize affects the performance. One example could be the paper mentioned above, Disentangling adaptive gradient methods from learning rates by Agarwal et al. 2020.

**Experimental Designs Or Analyses:**

The experimental section is well designed (e.g. including a comprehensive list of baselines, as well as detailed loss curve plots), albeit only encompassing two classical tasks and, in that sense, unclear how generalizable the trends are to broader tasks.

**Methods And Evaluation Criteria:**

Yes.

**Other Comments Or Suggestions:**

Throughout, citations with parentheses are used incorrectly. For example in line 19 in the abstract rather than “the framework of (Gao et al., 2024)” it should be “the framework of Gao et al. (2024).” This can be adapted by a change of the citing command (e.g. from \cite to \citep or something of the sort). Further examples include line 30 right hand column,  line 76 right hand side (change to “due to Gao et al., (2024)), line 106 right hand side, and line 115 left hand side.

I think in line 48, right hand column, you mean to say “UNexperienced users”?

In Lemma 2.3 you use the big O notation, but nowhere else I think — for consistency maybe put in the constants here too?

**Other Strengths And Weaknesses:**

I appreciate the intuition given on exactly what the method is doing and how that plays into the analysis and the interpretation of the loss blow-up during the “warm-up” phase.

One area of improvement in my opinion is that I believe more discussion and nuance are needed for the analysis of the local behavior. Currently the authors simply say that the preconditioner is chosen “to adapt to the local region, at the price of an additional regret term” that corresponds to the path length. The reason I believe more nuance is needed is that a priori it is not clear how big one should expect this path length to be, and that would determine the quality of the adaptation. So, without further analysis of why or when we’d expect this to be small (which a priori I don’t fully see why it would necessarily be), it is not clear that the analysis fully satisfactorily characterizes the local behavior.

**Questions For Authors:**

Could you comment more on the local convergence guarantee and when/why we can expect the path length contribution to be innocuous?

**Relation To Broader Scientific Literature:**

The paper is well related to the online learning and optimization literatures, by employing online learning tools and applying them to study the convergence behavior of an optimization method that has nice intuition behind it and potential for empirical appeal as an option to other popular optimizers currently used (AdaGrad, Adam, BFGS, etc.). Possibly more discussion could be included on observed empirical behavior of these optimizers, as well as systems-related reasons for choosing one versus the other. (One somewhat relevant example of could be Disentangling adaptive gradient methods from learning rates by Agarwal et al. 2020.)

**Theoretical Claims:**

I did not have time to carefully review the proofs (as they all are in the supplementary material).

---

> ### Author Rebuttal · Authors · 2025-03-30
>
> Thank you for your valuable feedback on our paper!
>
> **Experimental Designs Or Analyses**
>
> 1. Unclear how generalizable the trends are to broader tasks
>
>    Thank you for the comment. Our paper is primarily theoretical, so the experiments are intended to test the theory; hence, we compare HDM-Best with current popular first-order adaptive methods and quasi-Newton methods. To address your curiosity about whether HDM generalizes to other tasks, note that in the past 25 years, HDM (and its variants) have been widely used (despite a lack of theoretical guarantees) and have empirically shown excellent performance in a range of tasks [1-10].
>
>    We believe that our methodological tweaks motivated by our theoretical analysis (such as the null step) are likely to benefit these other applications as well. We also expect that our theoretical framework could also be adapted to some of these other setups (stochastic, nonsmooth, nonconvex, ...), but to do so is beyond the scope of this paper.
>
> **Relation To Broader Scientific Literature**
>
> 1. More discussion on the empirical behavior of these optimizers and choice
>
>    Thanks for the comment. First, we would like to note that although we frame HDM as a standard-alone algorithm to analyze its performance, it is a plugin-type acceleration technique that is compatible with most existing optimizers. For example, HDM has demonstrated superior performance when integrated with Adam [3]. Using our analysis framework, one can develop HDM variants such as HDM-Adam.
>
>    Pertaining to HDM-Best (HDM + HB) alone, we find that for deterministic optimization problems, if $f(x)$ looks globally quadratic, then QN works well. If it does not look quadratic near the starting point, we expect HDM to be a better choice. Since smooth strongly convex functions locally look quadratic near the optimum, we might expect that the best method would switch from HDM to QN as it nears convergence.
>
> 2. More discussions on the related literature
>
>    Thank you for pointing out this missing reference! We will add a more comprehensive discussion in the revision.
>
> **Other Strengths And Weaknesses**
>
> 1. Dynamic regret and path-length analysis
>
>    This is a very good question. Indeed, the path-length analysis is very detailed -- not just problem-dependent but actually trajectory-dependent! -- and it is hard to fully quantify its benefits in the standard oracle access model with only first-order information.
>
>    Here is one way to understand the benefit: our path-length dependent upper bound holds for **any** path $\\{\hat{P}_k\\}$, and in particular, it holds for the path $\\{\hat{P}_k\\}$ that *maximizes the algorithm progress*. In other words, the algorithm achieves the performance guaranteed by the optimal path. Let's call this the "long path" regime.
>
>    As another example of the strength of our analysis, the bound is always at least as good as the static adaptivity result guaranteed by Theorem 3.1, since we can take $\hat{P}_k\equiv \hat{P}$ (e.g., $1/L\cdot I$). Let's call this the "short path" regime.
>
>    In practice, we observe that HDM's behavior can be divided into two stages:
>
>    1. When the algorithm iterates $x^k$ are far from $x^\star$, the change in landscape is more drastic and we expect the long path regime to capture the convergence behavior.
>
>    2. When $x^k$ is near convergence,  $f(x)$ locally behaves like a quadratic, $P_k$ remains more stable and the short path regime describes the behavior.
>
>    Overall, we believe that having a trajectory dependent guarantee with a non-trivial worse-case guarantee is a powerful best-of-both-worlds result.
>
> **Other Comments Or Suggestions**
>
> 1. Citation issues and typos
>
>    Thank you for pointing out the issues, we have fixed these issues.
>
> 2. Lemma 2.3
>
>    Thank you for pointing this out. Using Lemma B.2, the constant here is $\frac{1}{2}(LD+1)^2$ and we will add it in the revised version.
>
> **Questions**
>
> Please see "Other Strengths And Weaknesses".
>
> We again appreciate the constructive feedback of the reviewer and thank you for the time spent on our paper.
>
> ----
> **References**
>
> [1] Schraudolph, N.N, 1999. Local gain adaptation in stochastic gradient descent.
>
> [2] Almeida, L.B. et al, 1999. Parameter adaptation in stochastic optimization.
>
> [3] Baydin, A.G. et al, 2017. Online learning rate adaptation with hypergradient descent.
>
> [4] Itakura, K. et al, 2020. Adapting the learning rate of the learning rate in hypergradient descent.
>
> [5] Chandra, K. et al, 2022. Gradient descent: The ultimate optimizer.
>
> [6] Jie, R. et al, 2022. Adaptive hierarchical hyper-gradient descent.
>
> [7] Amid, E. et al, 2022. Step-size adaptation using exponentiated gradient updates.
>
> [8] Wang, Z. et al, 2023. Fedhyper: A universal and robust learning rate scheduler for federated learning with HDM.
>
> [9] Yang, Z. & Li, X, 2024. Powered stochastic optimization with HDM for large-scale learning systems.
>
> [10] Ozkara. et al, 2024. MADA: Meta-adaptive optimizers through HDM

---

### Decision · Program_Chairs · 2025-05-01

**Decision:**

Accept (poster)

**Comment:**

This paper presents a rigorous theoretical analysis of the hypergradient descent method (HDM) for optimization, including both global and local convergence guarantees. The work provides comprehensive theoretical foundation for HDM, which has been used heuristically for 25 years, and introduces novel momentum-based variants with improved practical performance.

The paper's primary strengths include its theoretical analysis, which establishes explicit non-asymptotic bounds for both global and local convergence. The authors develop an online learning perspective that yields insights into HDM's behavior and extends to other first-order methods with adaptive hyperparameter updates. The practical variant HDM-Best demonstrates competitive performance with L-BFGS while using less memory, supported by empirical evaluation.

During the review process, some concerns were raised about theoretical aspects and experimental validation. In response, the authors:

- Added experiments comparing against line-search GD/AGD and other adaptive methods.
- Clarified the theoretical contributions, particularly emphasizing HDM as only the second known family of first-order methods achieving superlinear convergence.
- Provided explanations of the path-length analysis and its unification of HDM's behavior both far from and near the optimum.
- Added comparisons with recent results on superlinear convergence rates of quasi-Newton methods.

While one reviewer remained concerned about the alignment between theoretical results (which assume full-matrix preconditioners) and practical implementation (using diagonal preconditioners), the overall consensus is that the paper makes valuable contributions to both theory and practice of optimization methods.

The paper would benefit from:

1. More explicit discussion of dimensional dependencies in the theoretical bounds
2. Clearer exposition of the tradeoffs between diagonal and full-matrix variants
3. Additional discussion of the practical implications of theoretical assumptions